# Four-century history of land transformation by humans in the United States (1630-2020): Annual and 1-km grid data for the HIStory of LAND changes (HISLAND-US)

Xiaoyong Li[1,2,3,4], Hanqin Tian[2], Chaoqun Lu[5], Shufen Pan[3,2]

[1]State Key Laboratory of Urban and Regional Ecology, Research Center for Eco-environmental Sciences, Chinese Academy of Sciences, Beijing 100085, China.
[2]Schiller Institute for Integrated Science and Society, Department of Earth and Environmental Sciences, Boston College, Chestnut Hill, MA 02467, USA.
[3]International Center for Climate and Global Change Research, College of Forestry, Wildlife and Environment, Auburn University, Auburn, AL 36849, USA.
[4]University of Chinese Academy of Sciences, Beijing 100049, China.
[5]Department of Ecology, Evolution, and Organismal Biology, Iowa State University, Ames, IA 50010, USA.

*Correspondence to*: Hanqin Tian (hanqin.tian@bc.edu)

**Abstract.** The land of the conterminous United States (CONUS) has been transformed dramatically by humans over the last four centuries through land clearing, agricultural expansion and intensification, and urban sprawl. High-resolution geospatial data on long-term historical changes in land use and land cover (LULC) across the CONUS is essential for predictive understanding of natural-human interactions as well as land-based climate solutions for the United States. A few efforts have reconstructed historical changes in cropland and urban extent in the United States since the mid-19[th] century. However, the long-term trajectories of multiple LULC types with high spatial and temporal resolutions since the Colonia Era (early 17[th] century) in the United States are not available yet. By integrating multi-source data, such as high-resolution remote sensing image-based LULC data, model-based LULC products, and historical census data, we reconstructed the history of land use and land cover for the conterminous United States (HISLAND-US) at an annual time scale and 1 km x 1 km spatial resolution in the past 390 years (1630–2020). The results show widespread expansion of cropland and urban land associated with rapid loss of natural vegetation. Croplands are mainly converted from forest, shrub, and grassland, especially in the Great Plains and North Central regions. Forest planting and regeneration accelerated the forest recovery in the Northeast and Southeast since the 1920s. The geospatial and long-term historical LULC data from this study provides critical information for assessing the LULC impacts on regional climate, hydrology, and biogeochemical cycles as well as achieving sustainable use of land in the nation. The datasets are available at https://doi.org/10.5281/zenodo.7055086 (Li et al., 2022).

## 1 Introduction

Land use and land cover (LULC) change is an essential component of global change, and humans have altered over one-third of the Earth's land surface (Foley et al., 2005; Winkler et al., 2021). The human-induced LULC changes, such as cropland

expansion, deforestation, and tree planting, have profound impacts on climate change, carbon and nitrogen cycles, and biodiversity (Houghton et al., 1999; Dangal et al., 2014; Domke et al., 2020; Lark et al., 2020; Tian et al., 2020). In particular, managing agriculture and forest-related land use activities have been recognized as a critical pathway to achieve climate

mitigation targets (Grassi et al., 2017; Griscom et al., 2017). Thus, a better understanding of historical LULC and its spatial-temporal dynamics is critical for quantifying the effects of LULC change on the ecosystem and climate.

In the past four centuries, the conterminous United States (CONUS) has experienced dramatic land use and land cover (LULC) changes associated with land clearing, cropland reclamation, and urban land expansion (Steyaert and Knox, 2008; Drummond and Loveland, 2010; Oswalt et al., 2014; Sohl et al., 2016). Before the arrival of Europeans, indigenous agriculture and crop

planting existed in the eastern woodlands, the Great Plains, and the South (Hurt, 2002). Since the first colony in Virginia was established in 1607, cropland and pasture began to expand by land clearing, which initially occurred in the eastern United States. During the Colonia Era, most people lived in the east of the Appalachian Mountains, and agriculture was the primary livelihood. In the 19th century, territorial expansion (e.g., Louisiana Purchases) opened up new areas for agriculture. Driven by the western movement, land clearing, agriculture expansion, and deforestation expanded across the Appalachian Mountains

into Ohio, the Mississippi River Basin, and the Great Lakes (Cole et al., 1998; Billington et al., 2001; Steyaert and Knox, 2008; Yu and Lu, 2018). In the Mississippi River Valley and Alabama, hardwood forests were cleared for cotton and grain production (Hanberry et al., 2012). The center of lumber production was shifted from the Northeast to the Great Lakes in the 1850s (Fickle et al., 2001). In California, agriculture and ranching expanded throughout the state and soon became an exporter of wheat as the gold mining waned (Olmsted and Rhode, 2017). Entered the 20th century, cropland and pasture in New

England, the Atlantic coast, and the Southeast were abandoned (Foster, 1992; Hall et al., 2002; Jeon et al., 2014). The environmental protection movement originated in the 1880s. Both tree planting and forest regeneration from abandoned agricultural land accelerated forest restoration (Stanturf et al., 2014). In the following 90 years, the national total plantation forest area increased to 27 million hectares (Mha) (Oswalt et al., 2014, 2019; Chen et al., 2017). While general trends in historical US landscape change are known, we still lack a long-term and spatial-explicit LULC dataset to characterize historical

LULC trajectories for the CONUS.

Several efforts have produced LULC data for the CONUS in the past several decades. For example, multiple contemporary and spatially explicit LULC products with a resolution from 30 m to 1 km are available, including Global Land Cover (GLC) 2000 (Bartholome and Belward, 2005), MODIS land cover (Friedl et al., 2010), GlobeLand30 (Chen et al., 2015), National Land Cover Database (NLCD) (Yang et al., 2018; Homer et al., 2020), and Cropland Data Layer (CDL) (Boryan et al., 2011;

Lark et al., 2017, 2021). However, these datasets were generated using remote sensing images and cannot be used to characterize the century-long land use dynamics. Global-scale and long-term coverage land use datasets (e.g., Land and Use Harmonization (LUH2), the History Database of Global Environment (HYDE)) are widely used in global climate simulations and carbon budget projects (Goldewijk et al., 2017; Hurtt et al., 2006, 2020). However, these datasets have a coarse resolution (from 5 arcmins to 0.25 degrees), which cannot present regional-scale details well (Li et al., 2016; Yu and Lu, 2018). Moreover,

the data uncertainties will significantly impact the quantification of LULC effects on the ecosystem (Peng et al., 2017; Yu et

al., 2019). Some studies focused on reconstructing historical single-type land use datasets (e.g., built-up area and cropland) for the US (Zumkehr and Campbell, 2013; Yu and Lu, 2018; Lerk et al., 2020). Nevertheless, the dynamics of pasture, forest, shrub, and grassland also profoundly impact ecosystem carbon dynamics (Chen et al., 2006; Tian et al., 2012). Therefore, developing a long-term and high-resolution LULC dataset with multiple types for the CONUS is essential for understanding the LULC change history and LULC impact on ecosystem dynamics, regional climate, hydrology, carbon and nitrogen cycles, and greenhouse gas emissions.

In this study, we aim to reconstruct the HIStory of LAND use and land cover for the conterminous United States (HISLAND-US) and analyze the spatial and temporal patterns of LULC changes during 1630–2020 by integrating high-resolution satellite-based LULC data, reliable inventory data, and model-based LULC data. This study consists of three parts: a description of input data and methods, an analysis of spatiotemporal characteristics of LULC in the past four centuries, and a comparison between our results and other studies. We also discussed the driving forces of LULC changes and the uncertainties of the newly developed dataset.

## 2 Materials and Method

This study reconstructed the LULC history (1630–2020) at annual time step and 1 km x 1 km spatial resolution for the CONUS (48 states) using remote sensing-based LULC data, model-based land use data, and historical census data. In addition, we aggregated the state-level data into eight subregions to analyze the regional divergence of LULC changes. These subregions include Northeast, Northeast, North Central, Southeast, South Central, Great Plains, Intermountain, Pacific Northwest, and Pacific Southwest (Oswalt et al., 2014, 2019) (Figure 1).

The reconstruction process of historical LULC data mainly included two parts: (1) reconstructing the historical urban land, cropland, pasture, and forest area at the state level (Section 2.2), (2) generating 1 km x 1 km spatial resolution gridded LULC data (Section 2.3). Figure 2 shows the general workflow for generating historical LULC data. The following sections provide a detailed description of the input data and how we process the data.

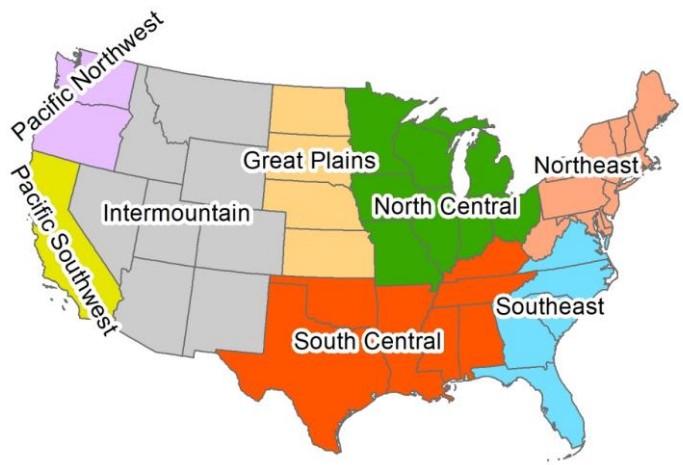

**Figure 1: The division of the conterminous United States into eight subregions for data synthesis and analysis in this study.**

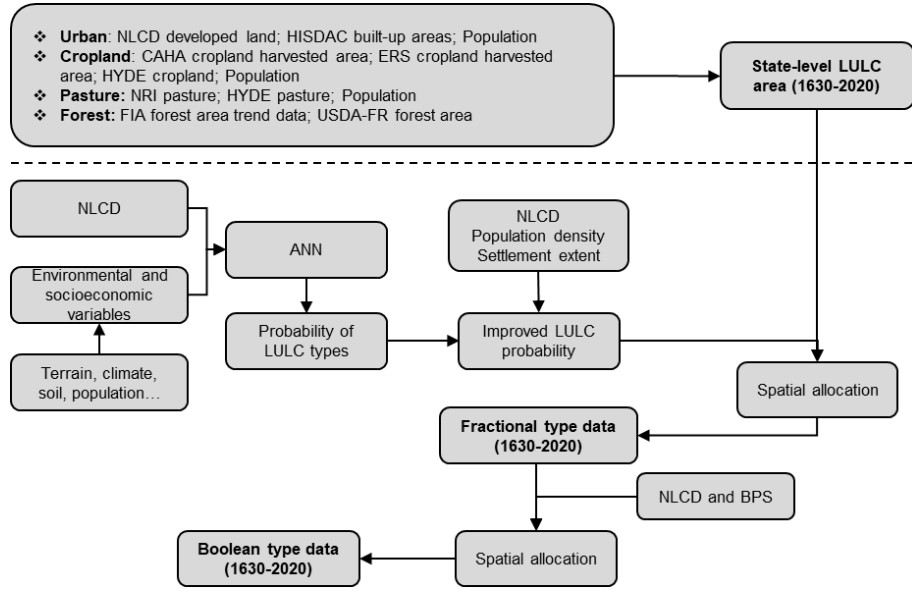

90

**Figure 2: Workflow for generating historical land use and land cover data for the conterminous United States. NLCD: National Land Cover Database; HISDAC: Historical Settlement Data Compilation; CAHA: Census of Agriculture Historical Archive; ERS: Economic Research Service; HYDE: History Database of the Global Environment; NRI: National Resource Inventory; FIA: Forest Inventory and Analysis; USDA-FR: USDA Forest Resources of the United States, 2017; ANN: Artificial Neural Network; BPS:**
95 **Biophysical Settings.**

## 2.1 Input datasets for land use and land cover reconstruction

The input datasets included satellite-based LULC data (National Land Cover Database, NLCD), model-based land use datasets (i.e., HYDE3.2 baseline), census and inventory data, and other auxiliary data (Table 1, Table S1-S4). The spatial data were resampled or aggregated to 1 km x 1 km resolution for further processing. We also collected some other LULC datasets to

100 validate the newly developed dataset, including cropland density (Yu and Lu, 2017), historical fractional cropland areas (Zumkehr and Campbell, 2013), Economic Research Service (ERS) Major Land Uses data (Bigelow and Borchers, 2017), Land Use Harmonization (LUH2) (Hurtt et al., 2020), CONUS historical land use and land cover (Sohl et al., 2016), county-level crops area (Crossley, 2020) and hay area (Haines et al., 2018) (Table A2).

**Table 1: Summary of the input datasets.**

| Data variables | Time period | Resolution | Data sources |
| --- | --- | --- | --- |
| National Land Cover Database | 2001, 2003, 2006, 2008, 2011, 2013, 2016, 2019 | 30 m | Multi-Resolution Land Characteristics Consortium https://www.mrlc.gov/ |
| Historical Settlement Data Compilation (HISDAC) | 1810-2015 | 250 m 5-year interval | https://dataverse.harvard.edu/dataverse/hisdacus |
| ERS Major Land Uses | 1910-2020 | Nation-level Annual | https://www.ers.usda.gov/data-products/major-land-uses/ |
| CAHA Cropland harvested area | 1879-2017 | State level 4 to 10-year interval | https://agcensus.mannlib.cornell.edu/AgCensus/homepage.do |
| HYDE3.2 cropland (Baseline version) | 1600-2017 | 5 arcmin 10-year interval | https://dataportaal.pbl.nl/downloads/HYDE/HYDE3.2/ |
| NRI Pasture area | 1982-2017 | State-level 5-year interval | National Resource Inventory Summary Report 2017 https://www.nrcs.usda.gov/wps/portal/nrcs/main/national/technical/nra/nri/results/ |
| HYDE3.2 pasture (Baseline version) | 1600-2017 | 5 arcmin 10-year interval | https://dataportaal.pbl.nl/downloads/HYDE/HYDE3.2/ |
| Forest area (USDA) | 1630-2017 | State level 5 to 18-year interval | Forest Resources of the United States, 2017. https://www.fs.usda.gov/treesearch/pubs/57903 |
| Forest area (FIA) | 1630-2000 | State level 10-year interval | Forest Inventory and Analysis https://www.fia.fs.fed.us/slides/Trend-data/Web%20Historic%20Spreadsheets/1630_2000_US_pop_and_forestarea.xls |
| Total population | 1630-1999 | State level Annual | Coulson and Joyce (2003). United States State-level Population Estimates: Colonization to 1999. |
| | 2000-2020 | State level Annual | https://www.census.gov/en.html |
| Population density | 1790-2010 | 1 km 10-year interval | Fang et al. (2018) https://doi.org/10.6084/m9.figshare.c.3890191 |
| HYDE3.2 population (Baseline version) | 1600-2017 | 5 arcmin 10-year interval | https://dataportaal.pbl.nl/downloads/HYDE/HYDE3.2/ |
| The extent of settled area | 1630-present | | https://maps.lib.utexas.edu/maps/histus.html |

Note: ERS: Economic Research Service, U.S. Department of Agriculture; CAHA: Census of Agriculture Historical Archive; HYDE: History Database of the Global Environment; USDA: United States Department of Agriculture; NRI: National Resource Inventory; FIA: Forest Inventory Analysis.

## 2.2 Historical land use and land cover area reconstruction

### 2.2.1 Urban land

In this study, we used the same definition for the developed land as NLCD for urban land. The developed land in NLCD includes four components: open space, low intensity developed land, medium intensity developed land, and high intensity developed land (Table 2). We used the NLCD developed land area during 2001–2019 as the urban land area baseline. Before

2001, we applied Historical Settlement Data Compilation for the United States (HISDAC-US) (Leyk et al., 2020; Uhl et al., 2021) as input to reconstruct the historical urban land area. The HISDAC-US built-up areas describes the built environment for most of the CONUS from 1810 to 2015 at 5-year temporal and 250 m spatial resolution using built-up property records, locations, and intensity data (Leyk and Uhl, 2018; Uhl et al., 2021). Here, we assumed that the HISDAC built-up areas data could capture the trend of urban land expansion. Then, the historical urban land can be estimated as follows:

$$HistUrban_{s,t} = HistUrban_{s,t+1} \times \frac{HISDAC_{s,t}}{HISDAC_{s,t+1}} \tag{1}$$

where $HistUrban_{s,t}$ and $HistUrban_{s,t+1}$ are the reconstructed urban land area of state $s$ in year $t$ and $t+1$; $HISDAC_{s,t}$ and $HISDAC_{s,t+1}$ are the HISDAC built-up area of state $s$ in year $t$ and $t+1$.

There is no census data on urban land area before 1810. Following Liu et al. (2010), we used population to estimate the urban land area by assuming that urban land expanded at the same rate as total population during 1630–1810. The urban land area of each state can be calculated as follows:

$$HistUrban_{s,t} = HistUrban_{s,t+1} \times \frac{Pop_{s,t}}{Pop_{s,t+1}} \tag{2}$$

where $HistUrban_{s,t}$ and $HistUrban_{s,t+1}$ are the reconstructed urban land area of state $s$ in year $t$ and $t+1$; $Pop_{s,t}$ and $Pop_{s,t+1}$ are the total population of state $s$ in year $t$ and $t+1$.

### 2.2.2 Cropland

The definition of cropland varies in the existing literature and datasets (Zumkehr and Campbell, 2013; Bigelow and Borchers, 2017; Goldewijk et al., 2017; Homer et al., 2020, Table S5). Cropland, defined by the U.S. Department of Agriculture (USDA) Economic Research Service (ERS), includes five components: cropland harvested, crop failure, cultivated summer fallow, cropland pasture, and idle cropland (Table 2). In this study, we only count the cropland harvested area, which includes row crops and closely sown crops, hay and silage crops, tree fruits, small fruits, berries, and tree nuts, vegetables and melons, and miscellaneous other minor crops (https://www.ers.usda.gov/data-products/major-land-uses/glossary/#cropland). USDA Census of Agriculture Historical Archive (CAHA) recorded state-level cropland harvested areas at 4 to 10 years intervals (Table 1 and Table S5), which was used for historical cropland area reconstruction between 1879 and 2017. The CAHA cropland was interpolated into annual using the linear method first. To subtract the double-cropped area, we applied the annual national cropland harvested area without double-cropped area from ERS Major Land Uses data to adjust the interpolated cropland harvested area. The adjustment can be expressed as follows:

$$HistCrop_{s,t} = \frac{Cropland\ Harvested_{s,t}^{linear}}{Cropland\ Harvested_{conus,t}^{linear}} \times Cropland\ Harvested_{conus,t}^{ERS} \tag{3}$$

where $HistCrop_{s,t}$ is the reconstructed cropland area of state $s$ in year $t$; $Cropland\ Harvested_{s,t}^{linear}$ is the linearly interpolated cropland harvested area of state $s$ in year $t$ based on CAHA cropland harvested area; $Cropland\ Harvested_{conus,t}^{ERS}$ is the national total cropland harvested area without double-cropped area in year $t$. For 2018–2020, the state-level cropland area was calculated based on the state-level area weight in 2017.

For 1879–1910, there was no national-level cropland harvested area without double-cropped area. Therefore, we applied the trend of the CAHA cropland harvested area to reconstruct the historical cropland:

$$HistCrop_{s,t} = HistCrop_{s,t+1} \times \frac{CAHA\_CHA_{s,t}}{CAHA\_CHA_{s,t+1}} \qquad (4)$$

where $HistCrop_{s,t}$ and $HistCrop_{s,t+1}$ are the reconstructed cropland area of state $s$ in year $t$ and $t$+1; $CAHA\_CHA_{s,t}$ and $CAHA\_CHA_{s,t+1}$ are the cropland harvested area of state $s$ in year $t$ and $t$+1.

Because there was no available cropland census data at the state level before 1879, the HYDE cropland was used. We first estimated the cropland per capita by applying the trend of HYDE cropland per capita. Then, the total cropland area can be calculated by multiplying cropland per capita and total population. The data harmonization process can be expressed as follows:

$$HistCrop_{s,t} = (HistCrop\_p_{s,t+1} \times \frac{HYDE\_Crop\_p_{s,t}}{HYDE\_Crop\_p_{s,t+1}}) \times Pop_{s,t} \qquad (5)$$

where $HistCrop_{s,t}$ is the reconstructed cropland area of state $s$ in year $t$; $HistCrop\_p_{s,t+1}$ is the reconstructed cropland per capita of state $s$ in year $t$+1; $HYDE\_Crop\_p_{s,t}$ and $HYDE\_Crop\_p_{s,t+1}$ are HYDE cropland per capita of state $s$ in year $t$ and $t$+1.

### 2.2.3 Pasture

The definition of pasture also varies among multiple datasets (Goldewijk et al., 2017; U.S. Department of Agriculture, 2020; Table S6). In this study, we use the definition from the National Resource Inventory (NRI), in which pasture is the land that has a vegetation cover of grasses, legumes, and forbs, regardless of whether it is being grazed by livestock, planted for livestock grazing, or the production of seed or hay crops (Table 2). The NRI provides state-level pasture area with 5-year interval between 1982 and 2017, and we set the pasture area as the baseline for historical reconstruction. Because there was no available pasture census data at the state level before 1982, the HYDE pasture was applied. We first estimated the pasture per capita by applying the trend of HYDE pasture per capita. Then, the total pasture area can be calculated by multiplying pasture per capita and total population. The data harmonization process can be expressed as follows:

$$HistPasture_{s,t} = (HistPasture\_p_{s,t+1} \times \frac{HYDE\_Pasture\_p_{s,t}}{HYDE\_Pasture\_p_{s,t+1}}) \times Pop_{s,t} \qquad (6)$$

where $HistPasture_{s,t}$ is the reconstructed pasture area of state $s$ in year $t$; $HistPasture\_p_{s,t+1}$ is pasture per capita of state $s$ in year $t$+1; $HYDE\_Pasture\_p_{s,t}$ and $HYDE\_Pasture\_p_{s,t+1}$ are the HYDE pasture per capita of state $s$ in year $t$ and $t$+1.

### 2.2.4 Forest

In this study, we use the forest definition from Forest Inventory and Analysis (FIA), in which the forest is defined as land at least 10 percent stocked by forest trees of any size, or formerly having such tree cover, with a minimum area classification of 1 acre (Table 2). Two datasets were used for the historical forest area reconstruction. The first is the USDA Forest Resources (USDA-FR) of the United States 2017 (Oswalt et al., 2019). It provides state-level forest areas from 1630 to 2017 with twelve

175    snapshots (i.e., 1907, 1920, 1938, 1953, 1963, 1977, 1987, 1997, 2007, 2012, 2017) and a shot in 1630. Another is FIA's Forest area trend data (FATD), which includes state-level forest area from 1760 to 2000 at 10-year interval and a snapshot in 1630. The data was rebuilt by integrating FIA field data and reports (1950–2000), field inventories (1910–1940), Bureau of the Census land clearing statistics (1850–1900), clearing estimates proportional to population growth (1760–1840), and USDA forest report. For 1907–2017, the USDA-FR data was used without adjustments. Before 1907, to keep the raw data consistent,

180    we adopted USDA-FR in 1630 as the initial point and gap-fill the missing years by using the changes reflected by FATD data to reconstruct the forest area between 1630 and 1907. The following harmonization method was conducted to combine the two datasets:

$$HistForest_{s,t} = USDA\_FR_{s,1630} \times \frac{FATD_{s,t}}{FATD_{s,1630}} \tag{7}$$

where $HistForest_{s,t}$ is the reconstructed forest area of state $s$ in year $t$; $USDAFR_{s,1630}$ is the USDA–FR forest area of state s

185    in 1630; $FATD_{s,t}$ and $FATD_{s,1630}$ are the FATD forest area of state $s$ in year $t$ and 1630, respectively.

For 2018, 2019, and 2020, we first collected the latest forest area of each state. If a state did not publish the forest area of the latest year, we assumed that the area during these three years was the same as that in 2017. The latest forest area data can be accessed at https://fia-usfs.hub.arcgis.com/ (last accessed: Aug 30, 2022).

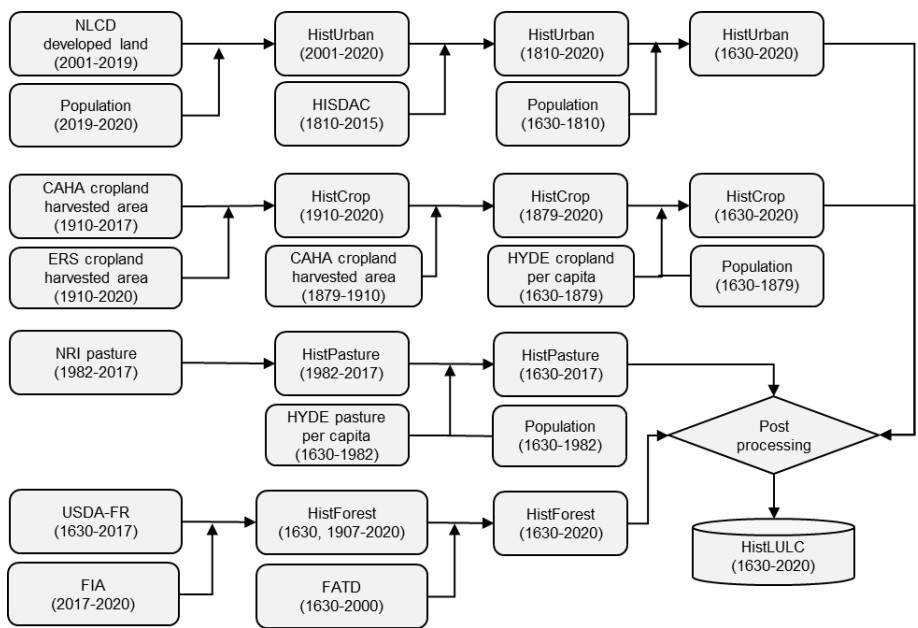

190    **Figure 3: Workflow for reconstructing historical land use and land cover area at the state level. NLCD: National Land Cover Database; HISDAC: Historical Settlement Data Compilation; ERS: Economic Research Service; CAHA: Census of Agriculture Historical Archive; NRI: National Resource Inventory; HYDE: History Database of the Global Environment; FIA: Forest Inventory and Analysis; USDA-FR: USDA Forest Resources of the United States, 2017; FATD: Forest Area Trend Data; HistUrban, HistCrop, HistPasture, HistForest, and HistLULC refer historical urban land, historical cropland, historical pasture, historical forest, and**

195    **historical land use and land cover.**

## 2.2.5 Post-processing of historical urban, cropland, pasture and forest area

Due to the difference in data sources in the reconstruction step, the total area of urban land, cropland, pasture, and forest may exceed the state's total land area (TLA). Therefore, we calibrated the reconstructed historical land use and land cover area using the following equations:

$$\begin{cases} A_{i,rc}^t(s) = A_{i,r}^t(s) & if\ TA_r^t(s) \leq TLA(s) \\ A_{i,rc}^t(s) = \frac{A_{i,r}^t(s)}{TA_r^t(s)} * TLA(s) & if\ TA_r^t(s) > TLA(s) \end{cases} \tag{8}$$

$$TA_{rc}^t = \sum_{i=1}^{n} A_{i,r}^t(s) \tag{9}$$

where $t$ is the current year; $A_{i,rc}^t(s)$ and $A_{i,r}^t(s)$ are re-calibrated area and reconstructed area for the land use class $i$ in the state $s$, respectively; $TA_{rc}^t$ is the total area of urban, cropland, pasture and forest; $n$ is total number of land use types; $s$ is the state index in the range from 1 to 48.

**Table 2: Definitions of urban, cropland, pasture, and forest in this study.**

| LULC | Definition |
|---|---|
| Urban land | Same as the definition of developed land in National Land Cover Database (NLCD). Developed land in NLCD include four components: open space, low intensity developed land; medium intensity developed land, and high intensity developed land (https://www.mrlc.gov/data/legends/national-land-cover-database-class-legend-and-description). |
| Cropland | Same as the definition of cropland in U.S. Department of Agriculture (USDA) Economic Research Service (ERS) major land use. Cropland defined by USDA ERS includes five components: cropland harvested, crop failure, cultivated summer fallow, cropland pasture, and idle cropland (https://www.ers.usda.gov/data-products/major-land-uses/glossary/#cropland). In this study, we only count the cropland harvested area subtracting the double-cropped area. |
| Pasture | Same as the definition of pasture in National Resource Inventory (NRI). Pasture is a land cover/use category of land managed primarily for the production of introduced forage plants for livestock grazing. |
| Forest | Same as the definition of forest from Forest Inventory Analysis (FIA). Forest is the land at least 10 percent stocked by forest trees of any size, or formerly having such tree cover, with a minimum area classification of 1 acre (https://www.fia.fs.fed.us/tools-data/maps/2007/descr/yfor_land.php). |

## 2.3 Approach for generating gridded land use and land cover data

### 2.3.1 Calculating the land use and land cover probability

Following previous studies, we applied the "Top-down" strategy to allocate the state-level LULC area to the grid level based on probability or suitability surfaces (Fuchs et al., 2013; West et al., 2014; He et al., 2015; Sohl et al., 2016). Previous spatially explicit land use and land cover simulation models, such as Conversion of Land Use and its Effects (CLUE) model and Forecasting Scenarios of Land use Change (FORE-SCE) model, used the logistic regression (LR) model to develop LULC

probability-of-occurrence (Verburg et al., 2009; Sohl et al., 2014, 2016; Li et al., 2016; Yang et al., 2020). However, it needs to train the LR model for the different units (e.g., county, grid) to calculate a good probability map due to the spatial heterogeneity of land conversion. In comparison, artificial neural networks (ANNs) can learn and fit complex relationships between input data and training targets and can be used to solve various non-linear geographical problems (Hagenauer and Helbich, 2022). Moreover, ANN performs better than LR in land use and land cover change simulation (Liu et al., 2016). Therefore, we used the ANN-based Probability of Occurrence Estimation tool in Future Land Use Simulation (FLUS) software to generate the LULC probability (Liu et al., 2017). The independent variables for the ANN model training and prediction include terrain (elevation and slope), climate (annual mean temperature, annual precipitation, annual maximum temperature (July), and annual minimum temperature (January)), crop productivity index, population density, distance to the city, distance to the road, distance to the railway, distance to the river, soil (soil organic carbon, soil sand, and soil clay) (Table A1). The Boolean type NLCD data in 2001 was used for ANN model training.

Over the past four centuries, the rules of LULC probability change a lot due to the interaction between natural environment and socioeconomic factors. The contemporary pattern of LULC probability is not representative for the early period (Sohl et al., 2016). Following Goldewijk et al. (2017), we improved the LULC probability by combining the biophysical probability and contemporary probability, as well as population density, human settlement extent, and satellite data. The total probability for each grid cell can be expressed as follows:

$$\begin{cases} TP_t = (S_{hist} \times w_1 + S_{satellite} \times w_2) \times (1.0 + r) & t \leq 2001 \\ TP_t = (S_{satellite} + Frac\_dt_{satellite}) \times (1.0 + r) & t > 2001 \end{cases} \quad (10)$$

$$\begin{cases} Prob_{hist} = Prob_{bio} * popd_t * SE_{weight,t} \\ Prob_{satellite} = Prob_{2001} * SE_{weight,t} \end{cases} \quad (11)$$

$$SE_{weight,t} = w_{t0} \times SE_{t0} + w_{t1} \times SE_{t1} \quad (12)$$

where $S_{hist}$ and $S_{satellite}$ is the LULC fraction generated by using the historical ($Prob_{hist}$) and satellite ($Prob_{satellite}$) probability; $w_1$ and $w_2$ is probability weight; $w_1$ is set to zero in 2001 and 100% in 1850 (and the pre-1850 period as well), while $w_2$ is set to 0 in 1850 (and the pre-1850 period as well) and 100% in 2001; $Frac\_dt_{satellite}$ is the NLCD LULC fraction dynamics between year $t$ and 2001; $SE_{weight,t}$ is settlement weight in year t, which is calculated based on the settlement in year t0 and year t1; $r$ is a random item with a range of [0, 0.5]. $Prob_{bio}$ is the LULC probability that only use biophysical variables (terrain, climate, and soil variables), $Prob_{2001}$ is the LULC probability that use all the variables; $popd_t$ is population density (Figure S5).

**2.3.2 Strategies to generate fractional and Boolean land use and land cover data**

Two types of gridded LULC data with 1 km x 1 km spatial resolution were generated. The first is fractional type, in which the dataset includes four fractional components: urban, cropland, pasture, and forest. Another is Boolean type with nine LULC types: urban, cropland, pasture, forest, shrub, grassland, wetland, water, and barren.

To generate the fractional gridded LULC data, we assumed that the fraction of each LULC type at the grid level was determined by the total probability (Fuchs et al., 2013; Tian et al., 2014; West et al., 2014; He et al., 2015). It means that a grid cell (LULC type k) with a high probability will have a high fraction. Based on this principle and the state-level LULC area, we generated the fractional LULC data at 1 km x 1 km resolution and annual time scale. The detailed information for generating fractional LULC data is shown in the following steps (Figure 4): (1) prepare the input data: state-level historical LULC area and probability; (2) calculate the state target LULC fraction for type k and initialize an empty LULC fraction map; (3) calculate a temporal fraction layer; (4) modify the temporal fraction, we assume that the fraction of water and barren is stable, and the sum of urban, crop, pasture, and forest fraction is lower than the maximum fraction in each grid cell; (4) add the temporal fraction data to the empty LULC fraction map; (5) judge whether the unallocated LULC area is smaller than 0.01 km$^2$, if yes, the iteration will stop and begin to allocate another LULC type, else the unallocated area will be assigned to target fraction and return to step (3). The allocation was processed until the unallocated area was less than the threshold (0.01 km$^2$). The above steps will be conducted for each state and output the fractional map.

Based on the LULC fraction map, we generated the Boolean type LULC data at 1 km x 1 km resolution. The detailed information is shown in the following steps (Figure 4): (1) prepare the input data: state-level historical LULC area and LULC fraction data; (2) generate a temporal LULC map (HistB) through identifying the dominate LULC type in each grid cell and initialize an empty LULC map (HisB$_E$); (3) calculate the area difference for LULC type $k$ between the HisB map and target area; (4) if the area difference is negative, we first sort the LULC fraction data where HisB equals to $k$, the top $m$ (equals to the target area) grid cells where HisB$_E$ not be assigned a value will be assigned as $k$, then if the available number of grid cell (type $k$) is less than the target area, we will sort the LULC fraction data where HisB map not equal to $k$, and the top $n$ (equals to the unallocated area) grid cells where HisB$_E$ not be assigned a value will be assigned as $k$; (5) if the area difference is positive, the grid cells where HisB data equals to $k$ and the will be assigned $k$ to HisB$_E$ not be assigned a value; then we will sort the LULC fraction data where HisB data not equals to $k$, and the top $n$ (equals to the unallocated area) grid cells where HisB$_E$ not be assigned a value will be assigned as $k$. If step (4) and (5) finish, the next LULC type will begin to allocate. After the four LULC types of allocation finish, the grid cell that is not assigned a type will be updated using the HisB data and LANDFIRE Biophysical Settings data (Figure A1; Rollins et al., 2009).

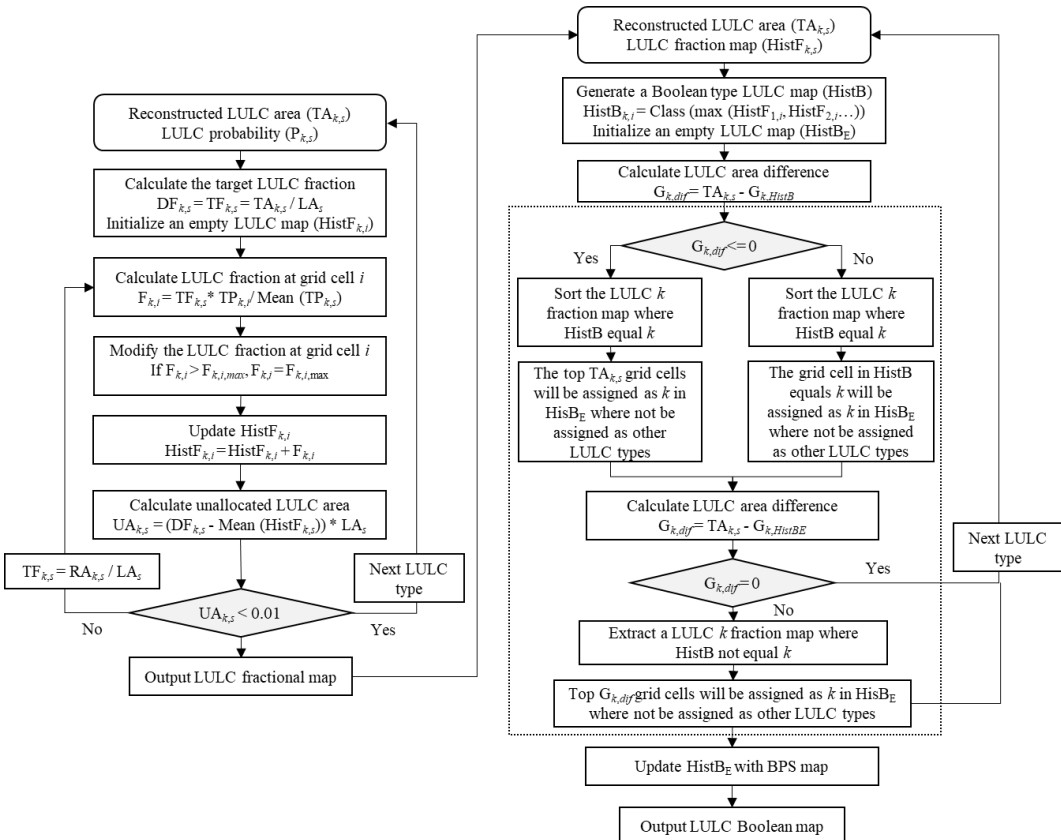

**Figure 4: Workflow for generating fractional (left) and Boolean (right) type LULC data.**

## 2.4 Data comparison

The lack of actual spatial explicit reference data made a complete formal validation impractical. Though the LULC definitions in this study are different from other LULC datasets, data comparison is a way to assess the accuracy of the reconstructed LULC area and spatial pattern. Thus, we conducted three data comparisons to increase the confidence of the newly developed LULC datasets. First, the state-level LULC area derived from the multisource datasets was used for comparison. Considering the differences in the cover period of multiple LULC datasets, we derived the average state-level statistics area for urban, cropland, pasture, and forest from 2000 to 2020 for comparison. Second, we collected the USDA county-level cropland area between 1840 and 2012 and compared the cropland proportion with that derived from our data in four selected years (1850, 1920, 1960, and 2002). Third, we compared urban, cropland, pasture, and forest from the newly developed LULC dataset with the NLCD during 2001–2019 at the grid level.

## 3 Results

### 3.1 Land use and land cover change during 1630–2020 in CONUS

The results showed that the LULC change from 1630 to 2020 was characterized by the expansion of cropland and urban land and the shrinking of natural land cover (e.g., forest, grassland, and shrub) (Figure 5, Figure A2-A5). In 1630, the primary landscape was the forest in the eastern US and Pacific Coast, grassland in the Great Plains, and shrub in the Rocky Mountains (Figure 5). Urban land, cropland, and pasture were mainly distributed in the east of US before 1850. Rapid cropland and pasture expansion occurred in the North Central (e.g., Iowa, Illinois, Minnesota), the Great Plains, and the Mississippi River Valley during 1850–1920 (Figure 5 and Figure A3). After 1920, the distribution of major LULC types became relatively stable (Figure 6). In the 2000s, the cropland in the Corn Belt regions, Central California, and Mississippi Alluvial Plain had the highest cropland density (Figure A3), and the highest pasture density was found in the east of Texas, Oklahoma, Missouri, and Kentucky (Figure A4).

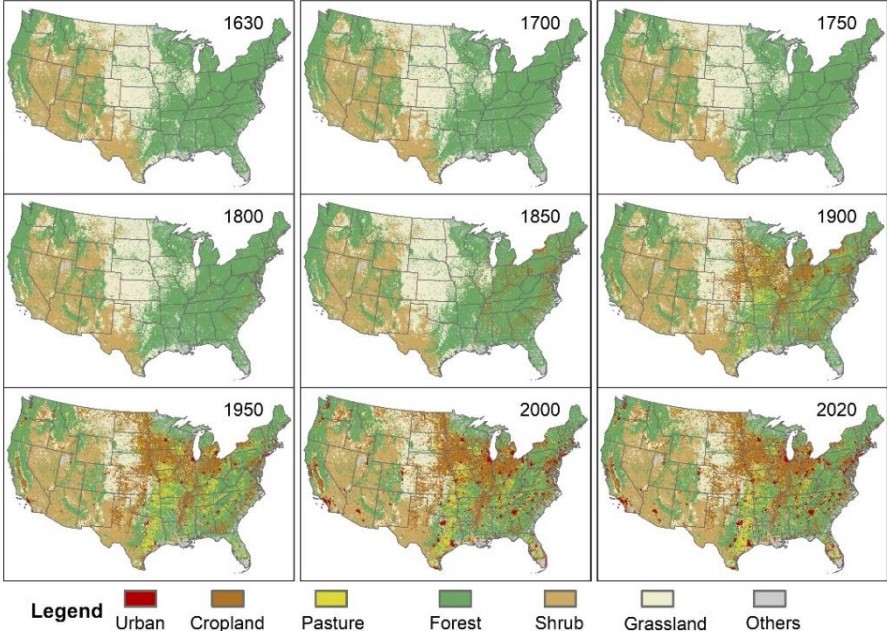

**Figure 5: Spatiotemporal patterns of land use and land cover in the conterminous United States during 1630-2020.**

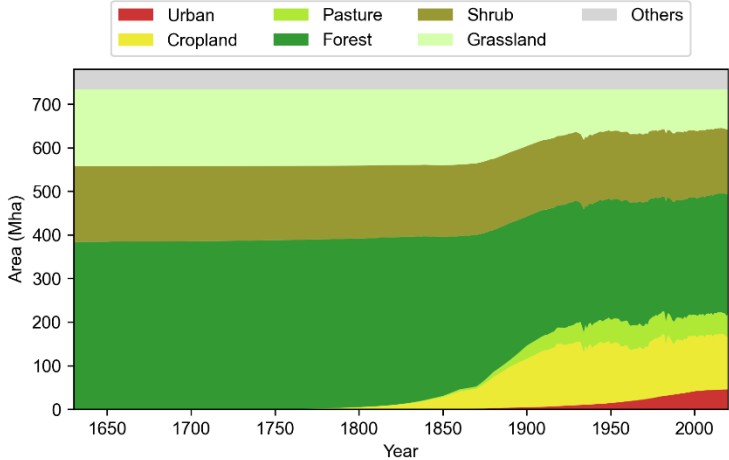

**Figure 6: Land use and land cover changes in the conterminous United States from 1630 to 2020.**

The US experienced the Colonial Era, the war of independence, and territorial expansion between 1630 and 1850. In this period, urban land increased by 0.80 Mha with a total population growth of 23 million (Figure 6). In the mid-1800s, the cheap land and the industrial revolution growth prospect attracted many European and Mexican immigrants, which accelerated urban development. In the second half of the 19th century, the population tripled and the total urban land increased to 4.3 Mha in 1900 (Figure 6). Entered the 20th century, both the rapid growth of population and urban land per capita accelerated the urban land expansion. Our result show that the urban land per capita increased from 0.02 ha/person in 1900 to 0.14 ha/person in 2020 (Figure S7). As a result, the national total urban land area increased to 45.46 Mha in 2020.

Cropland expanded slowly by 27.09 Mha from 1630 to 1850, and it increased substantially to 142.05 Mha in the following 70 years (Figure 6). Agriculture turned to be intensified after 1920, and but the total cropland area in the CONUS was relatively constant, with a peak area of 146.08 Mha in 1932 (Figure 6). Due to the competition with the high production land in the Midwest, cropland abandonment occurred in the Northeast, South, and Southeast (Bigelow and Borchers, 2017; Yu and Lu, 2018). During 1950–1975, the rise of the manufacturing and service industry resulted in agricultural labour and cropland area reduction. As the demand for biofuel and bulk grain grew in the 2000s, cropland began to extend again, and the total cropland area in 2020 was 126.94 Mha (Figure 6). Pasture showed an increasing trend with a slowly increasing rate during 1630–1850. It expanded more than 20 times from 1850 to 1950 and reached the maximum historical area (56.94 Mha) in 1959. The total pasture area in the CONUS kept relatively stable and decreased slowly in the following 70 years (Figure 6).

Forest was the dominant LULC type in the CONUS before the colonial era, which accounted for about 47% of the total land area. The trends in forest area were contrary to that of agricultural land before 1920. During 1630–1850, the national total forest loss was 40.83 Mha (Figure 6). Over the second period (1850–1920), forest area decreased by 83.02 Mha because of agricultural land occupation, lumber cut, and fuelwood consumption. In the third period (1920–2020), forest area has been relatively stable through forest management and planting (Figure 6).

## 3.2 Land use and land cover transitions during 1630-2020

The changes in the LULC area only reflected its quantitative changes. However, the LULC transition map further illustrates the spatial conversion distribution between two LULC types (Figure 7). Over the past 390 years, cropland expansion by occupying forest, shrub, and grassland was the primary LULC change characteristic (Figure 7). The natural land loss was mainly distributed in the North Central (e.g., Ohio, Indiana) and Southern states such as Tennessee, Texas, Alabama, and Georgia (Figure 7d). Cropland reclamation encroached 54.38 Mha (15.00% of total forest in 1630) of forest and 68.56 Mha (19.60% of total shrub and grassland in 1630) of grassland and shrub. Meanwhile, 37.76 Mha of forest and 11.15 Mha of shrub and grassland were converted to pasture. Moreover, urban land occupied more than 33.90 Mha of forest and 11.57 Mha of grassland and shrub (Table 3). In the early period (1630–1850), forest converted to cropland was the dominant LULC transition type, which was mainly distributed in the eastern US (Figure 7a). The US experienced the most dramatic LULC conversion with large forest and grassland loss in North Central and Great Plains during 1850–1920 (Figure 7b). Cropland expansion encroached 56.21 Mha of forest and 59.01 Mha of grassland, and pasture development also occupied more than 27.61 Mha of forest (Table 3). Furthermore, urban land expansion and abandoned cropland converted to forest (22.35 Mha) distributed in the Northeast and Southern states were the essential feature of LULC changes between 1920 and 2020 (Figure 7c).

Table 3: Net land use and land cover change during 1630-1850, 1850-1920, 1920-2020, and 1630-2020. Unit: Mha.

| LULC transition type | | 1630-1850 | 1850-1920 | 1920-2020 | 1630-2020 |
|---|---|---|---|---|---|
| Cropland to Others | Cropland to Urban | 0.00 | 0.46 | 10.92 | 0.00 |
| | Cropland to Pasture | 0.00 | 1.58 | 9.09 | 0.00 |
| | Cropland to Forest | 0.00 | 3.67 | 22.35 | 0.00 |
| | Sub-total | 0.00 | 5.71 | 42.36 | 0.00 |
| Others to Cropland | Pasture to Cropland | 0.00 | 0.27 | 0.97 | 0.00 |
| | Forest to Cropland | 25.91 | 56.21 | 11.40 | 54.38 |
| | Grassland to Cropland | 1.06 | 59.01 | 18.47 | 62.60 |
| | Shrub to Cropland | 0.02 | 5.43 | 3.21 | 5.96 |
| | Sub-total | 26.99 | 120.92 | 34.05 | 122.94 |
| Others to Pasture | Forest to Pasture | 2.44 | 27.61 | 15.24 | 37.76 |
| | Grassland to Pasture | 0.07 | 5.78 | 3.00 | 9.10 |
| | Shrub to Pasture | 0.00 | 0.68 | 1.56 | 2.05 |
| | Sub-total | 2.51 | 34.07 | 19.80 | 48.91 |
| Others to Urban | Forest to Urban | 0.76 | 4.07 | 18.01 | 33.90 |
| | Grassland to Urban | 0.03 | 1.60 | 2.81 | 7.56 |
| | Shrub to Urban | 0.00 | 4.07 | 2.71 | 4.01 |
| | Sub-total | 0.79 | 9.74 | 23.53 | 45.47 |

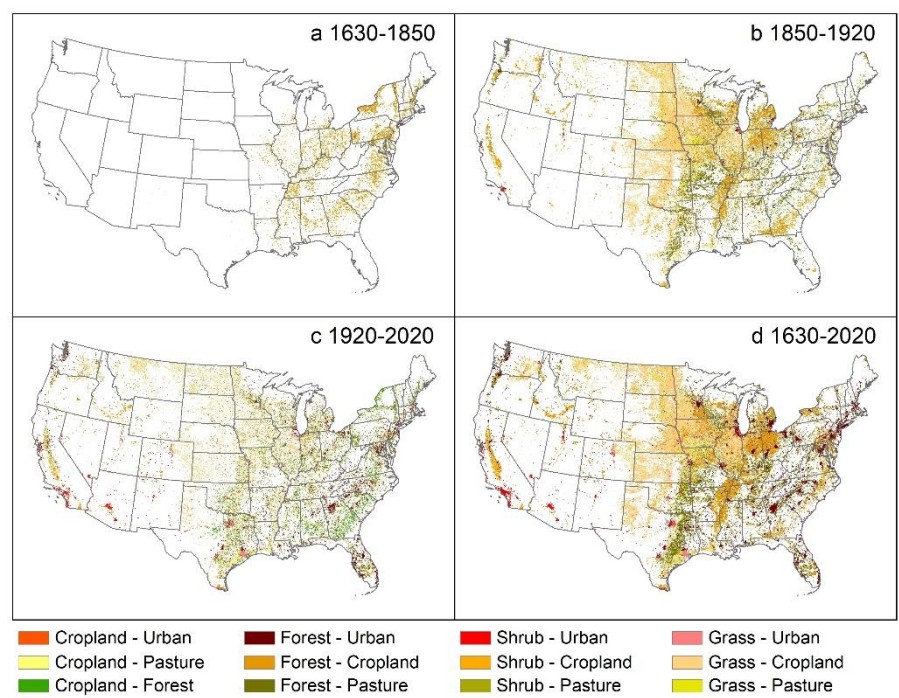

**Figure 7: Land transition (1 km x 1km spatial resolution) between 1630 and 1850 (a), 1850 and 1920 (b), 1920–2020 (c), 1630 and 2020 (d) in the conterminous United States.**

### 3.3 Land use and land cover changes during 1630–2020 at regional level

Given the differences in natural environmental conditions and social-economic development, land use and land cover changes showed spatial heterogeneity in the CONUS during 1630–2020. Since 1630, South Central experienced the most intensive urban land expansion (10.62 Mha), followed by the North Central (10.28 Mha), Southeast (7.38 Mha), and Northeast (6.00 Mha), respectively (Figure 8a). Rapid cropland expansion first occurred in the North Central, Northeast, South Central, and Southeast in the 1830s. Cropland in the Intermountain and the Great Plains began to develop after 1860. The trends of cropland

in eight regions except South Central and Southeast were consistent with the national total. Over the past four centuries, the North Central region had the largest cropland expansion area (46.01 Mha), followed by the Great Plains (31.41 Mha) and the South Central (20.10 Mha) (Figure 8b). Cropland in the South Central and Southeast had decreased by 4.91 Mha and 12.44 Mha since the 1930s due to the increasing urbanization pressures and low cropland profitability.

Similar to cropland, the Northeast was the first to develop pasture. The pasture experienced a rapid expansion during 1790–

1950 and reached the maximum historical area (4.56 Mha) in the 1950s, and then gradually decreased (Figure 8c). After the 1900s, the South Central had the largest pasture area. The maximum historical area was 21.07 Mha in 1950 and accounted for 37% of the national total. However, the pasture area in the North Central began to decrease in 1960, and 11.17 Mha of pasture was left in 2020 (Figure 8c).

Agricultural land encroachment, land clearing, and deforestation resulted in forest loss in eight regions (Oswalt et al., 2014,

2019). In the past four centuries, North Central lost the most forest area (36.12 Mha), followed by South Central (24.85 Mha). During 1850–1920, the forest area decreased rapidly in the North Central (24.96 Mha), South Central (29.39 Mha), Southeast (14.01 Mha), and Northeast regions (6.50 Mha). Most of the lost forest converted to cropland and pasture (Figure 8d). Since the 1920s, the regional forest area has been relatively stable with small fluctuations. Notably, the forest land recovered gradually, especially in Northeast, South Central, and Southeast. Compared with the 1920s, the total forest area in Northeast

increased by 6.87 Mha (Figure 8d).

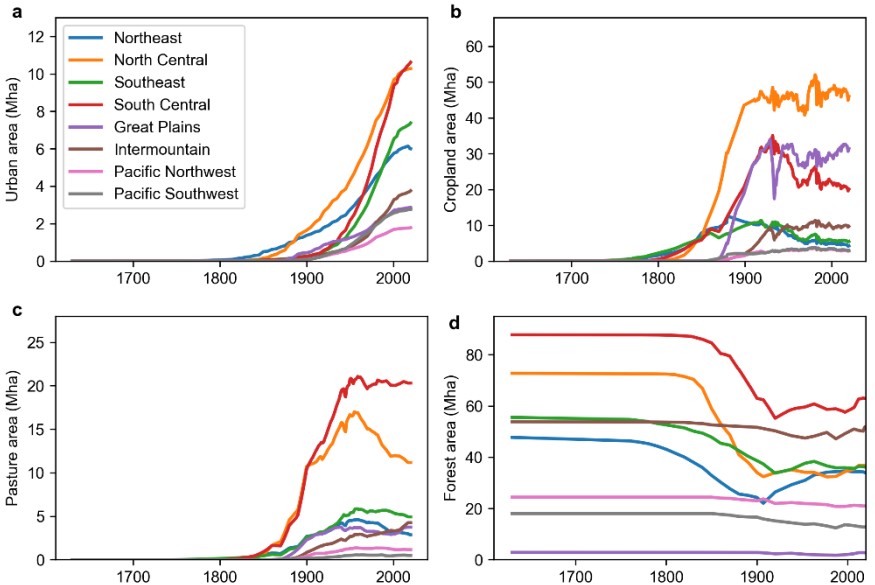

**Figure 8: Changes in areas of urban land (a), cropland (b), pasture (c), forest (d) in different geographic regions during 1630–2020.**

### 3.4 Comparison with other datasets

#### 3.4.1 State-level land use and land cover area comparison

We compared the state-level urban, cropland, pasture, and forest areas using data derived from ERS, HISDAC, HYDE, NLCD, LUH2, and YLmap with the newly developed LULC dataset. Generally, our data matches well with the data used for comparison (Figure 9). The urban land area from this study is higher than the ERS data (Figure 9a; $R^2$=0.93, Slope = 0.61) because ERS urban land only includes the densely-populated areas with at least 50000 people (urbanized areas) and densely-populated areas with 2500 to 50000 people (urban clusters). In contrast, HISDAC built-up areas data is higher than our data

(Figure 9a; $R^2$=0.88, Slope = 1.34), especially in Georgia, New York, North Carolina, Ohio, and Tennessee. It is because the HISDAC data is rebuilt using the detailed property records and have a relatively coarse resolution (Leyk et al., 2020). The cropland area derived from this study is consistent with NLCD (Figure 9b; $R^2 = 0.99$, Slope = 1.02) and YLmap (Figure 9b; $R^2 = 0.99$, Slope = 0.93). Nevertheless, the ERS cropland is higher than our data (Figure 9b; $R^2 = 0.96$; Slope = 1.26) because

the ERS cropland here includes the area the cropland harvested area, crop failure, cultivated summer fallow, cropland used for
pasture, and idle cropland. The coefficients of determination between our pasture acreages and NLCD (Figure 9c; $R^2 = 0.93$, Slope = 1.02) and HYDE (Figure 9c; $R^2 = 0.87$, Slope = 0.99) are higher than 0.87. For the forest, both NLCD and LUH2 data are lower than our data, especially in the Rocky Mountain states (Figure 9d; $Slope_{NLCD} = 0.72$, $Slope_{LUH2} = 0.66$). The differences in definition and data development method could result in LULC area differences for both pasture and forest (Table S1-S4), making it hard to compare. For example, the LUH2 forest area in Rocky Mountain states is lower than our data and
NLCD because they applied biomass density data to determine the forest extent. Though there still are some uncertainties, the comparison results show that the newly developed dataset can provide a relatively accurate LULC area at the state level.

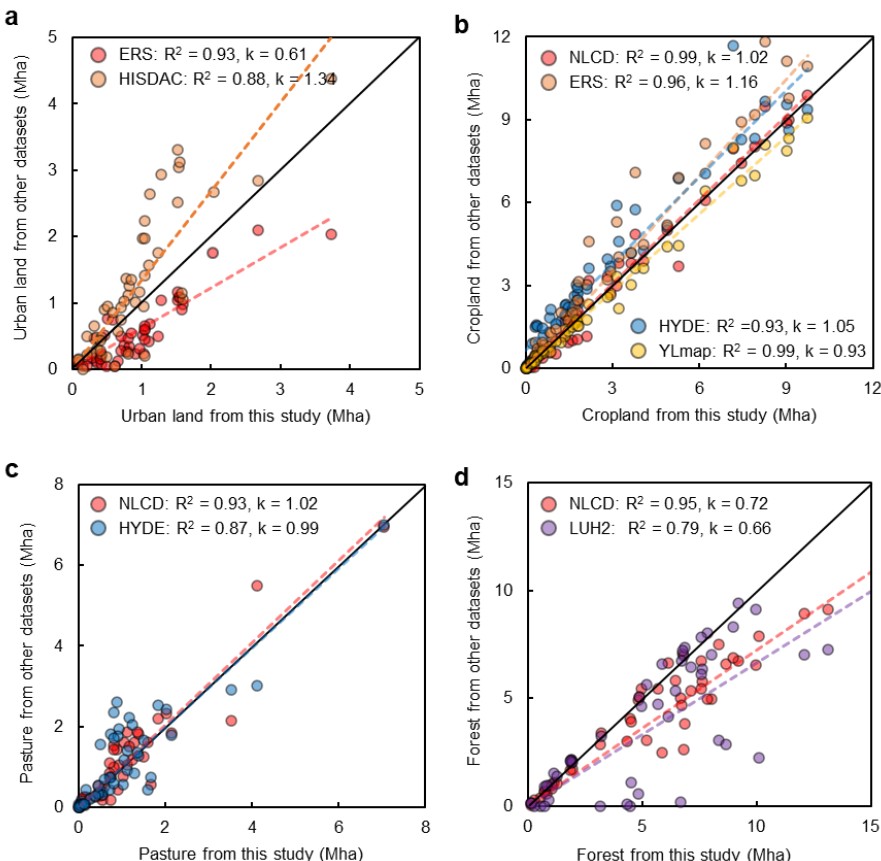

**Figure 9. Comparison of the average urban (a), cropland (b), pasture (c), and forest (d) area in each state among National Land**
**Cover Database (NLCD), Historical Settlement Data Compilation (HISDAC), Economic Research Service (ERS), Yu and Lu (2017) cropland (YLmap), History Database of the Global Environment (HYDE), Land Use Harmonization (LUH2) and this study. This study: 2000–2020; NLCD: 2001, 2003, 2006, 2008, 2011, 2013, 2016, and 2019; HISDAC: 2000, 2005, 2010, and 2015; ERS: 2002, 2007, and 2012; HYDE: 2000–2017; YLmap: 2000-2016; LUH2: 2000–2019.**

### 3.4.2 Comparison with cropland census data at county-level

An accurate cropland map is quite critical for historical LULC reconstruction. We compared our data with county-level census data to assess the accuracy. This study's spatial pattern of cropland proportion (i.e., cropland area/county area) is close to the census data in 1850, 1920, 1959, and 2002 (Figure 10). In 1850, both the newly developed cropland and census data showed high cropland density in the Black Belt, New England, and the North Central. In contrast, our data was higher in North Central, the east of Virginia and North Carolina, and the south of Georgia (Figure 10). Cropland derived from this study was higher

than the census data in the Atlantic coast, the Mississippi Alluvial Plain, the northwest of Texas, the west of Oklahoma, and California in 1920, 1959, and 2002. However, the cropland proportion in the Appalachian Mountains and the south of the Great Plains was lower than the census data (Figure 10). This underestimation may result from the low cropland fraction in satellite data because it is difficult for satellite data to identify the small area cropland patch in the mountain region and classify the pasture or grassland with cropland in the south of the Great Plains. Moreover, both datasets showed the cropland expansion

in the North Central, the Great Plains, the Mississippi Alluvial Plain, and California between 1850 and 2002. The cropland abandonment can also be found in the Appalachian Mountains between 1920 and 2002. The statistical comparison also shows that our data fits well with the census data in 1920 ($R^2 = 0.68$), 1960 ($R^2 = 0.89$), and ($R^2 = 0.91$) (Figure A6). Overall, the newly developed cropland has a relatively accurate spatial pattern and proportion.

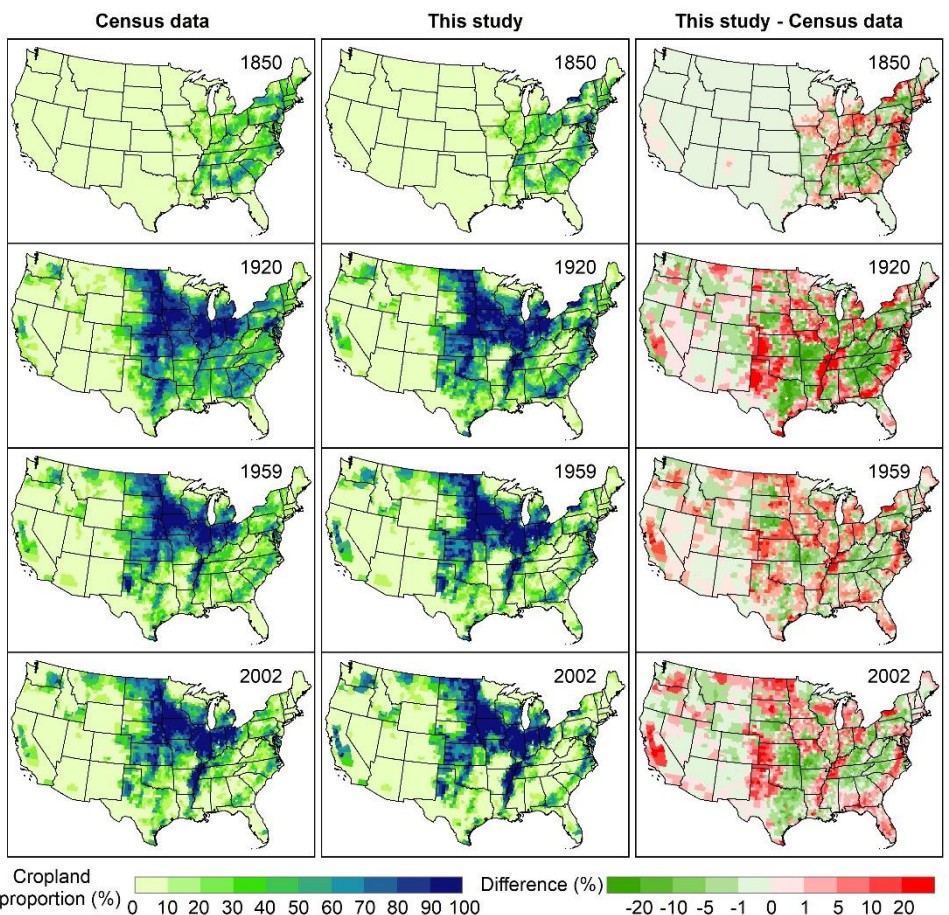

**Figure 10: Spatial comparison of county-level cropland proportion between our reconstruction and census data in 1850, 1920, 1959 and 2002. First column: cropland proportion from census data; Second column: cropland proportion derived from this study; Third column: cropland proportion difference between this study and census data.**

### 3.4.3 Comparison with NLCD at grid-level

The spatial patterns of urban, cropland, pasture, and forest in this study are close to the satellite-based data from NLCD, and most grid cells have a relatively small difference between 2001 and 2019 (Figure 11). Our results have a higher urban land fraction in the NLCD low urban density area, but the difference in 87% of urban grids is smaller than 10%. Cropland with a positive difference is mainly distributed in the Northeast, Alabama, and Missouri, in which 65.95% of grids have slight differences with less than 10% (Figure 11). 37.19% of grids have negative difference values and are mainly located in states with high cropland proportions. Moreover, most states in our data have a lower pasture fraction than NLCD data except in Oklahoma, Arkansas, Texas, and Georgia, and the grid cells with negative differences account for 39.82%. The reconstructed forest shows a higher density than NLCD in the South, Pacific coast, and Great Lakes. It underestimates the forest fraction in the central states, such as Missouri, Kentucky, and Ohio. There are 58.80% grids whose differences are relatively small and with a range from -10% to 20% (Figure 11).

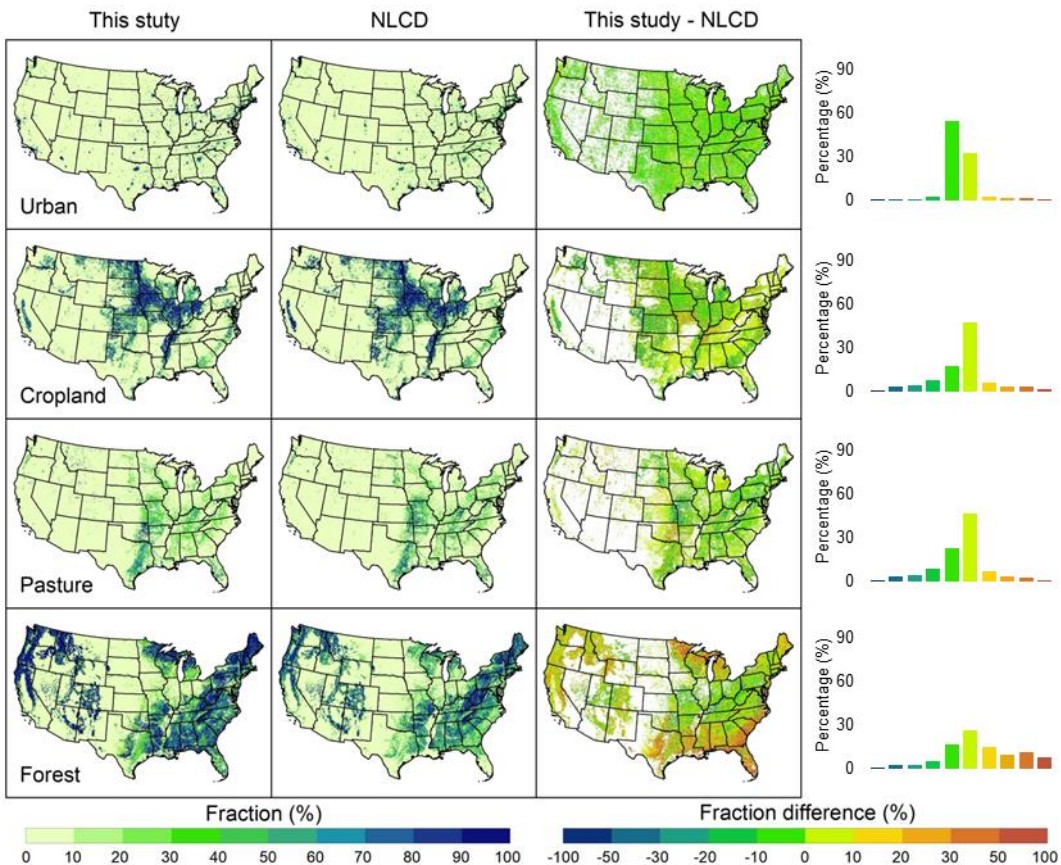

 **Figure 11: Spatial comparison between our reconstruction and satellite-based urban, cropland, pasture, and forest. First column: Reconstructed data in this study (average between 2001 and 2019); Second column: Satellite-based data (average between 2001 and 2019); Third column: Difference between first column and second column; Fourth column: Distributions of fraction difference between our reconstructed database and satellite-based data.**

## 4 Discussion

### 4.1 Comparison with the previous datasets

Compared with the ERS and HYDE data, the reconstructed urban land was higher (Figure 12a), attributed to the definition differences with NLCD. The ERS urban area includes the densely-populated areas with at least 50000 people (urbanized areas) and densely-populated areas with 2500 to 50000 people (urban clusters). The total urban land area from HISDAC data was higher than the newly developed data in the recent four decades (Figure 12a). Because the HISDAC built-up area dataset was developed by using the detailed property records data at a relatively coarse resolution (Lerk et al., 2020). Some small-scale built-up land cannot be identified using satellite images and NLCD may underestimate the total urban land area. Moreover, the HISDAC built-up areas underestimated the total urban area in the early years due to the high missing rate of property records (Lerk et al., 2020). Therefore, our data may also underestimate the total urban land area because we applied the trend

of HISDAC between 1810 and 2001. The spatial pattern of Boolean type urban land was consistent with the Sohl et al. (2016)
data and was mainly distributed in the area near the city, road, and railway (Figure 13). The spatial allocation rule determined
that the grid with a high probability of occurrence would be allocated first, which may underestimate the developed land in the
rural area (Verburg et al., 2009; Yang et al., 2020). Though some uncertainties in the urban data exist, we provided a long-
term description of urban land with higher resolution and consistency for the CONUS.

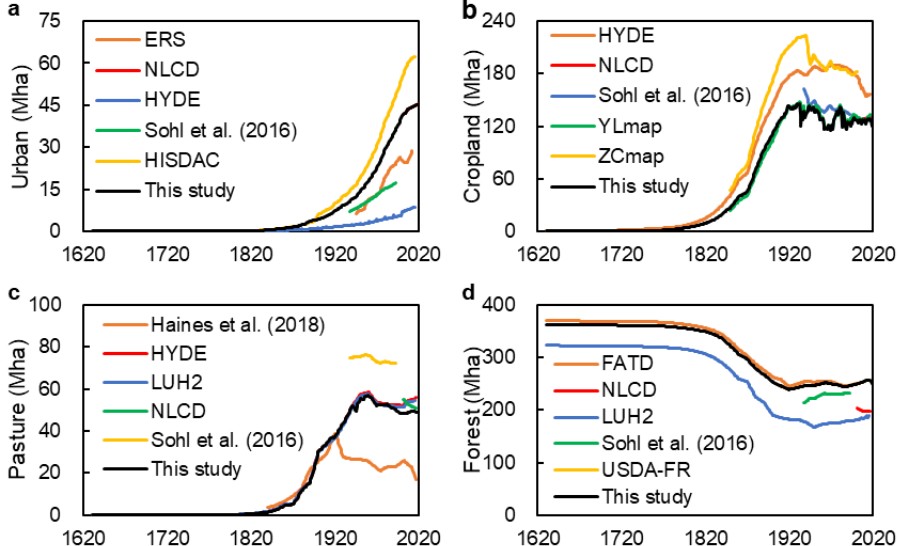

**Figure 12: Comparison with other datasets for the conterminous United States: urban land (a); cropland (b); pasture (c); forest (d).**
**NLCD: National Land Cover Database; HYDE: History Database of the Global Environment; HISDAC: Historical Settlement Data**
**Compilation; ERS: Economic Research Service; YLmap: Yu and Lu (2017) cropland density; ZCmap: Zumkehr and Campbell**
**(2013) historical fractional cropland areas; LUH2: Land Use Harmonization; FATD: Forest Area Trend Data; USDA-FR: USDA**
**Forest Resources of the United States of 2017.**

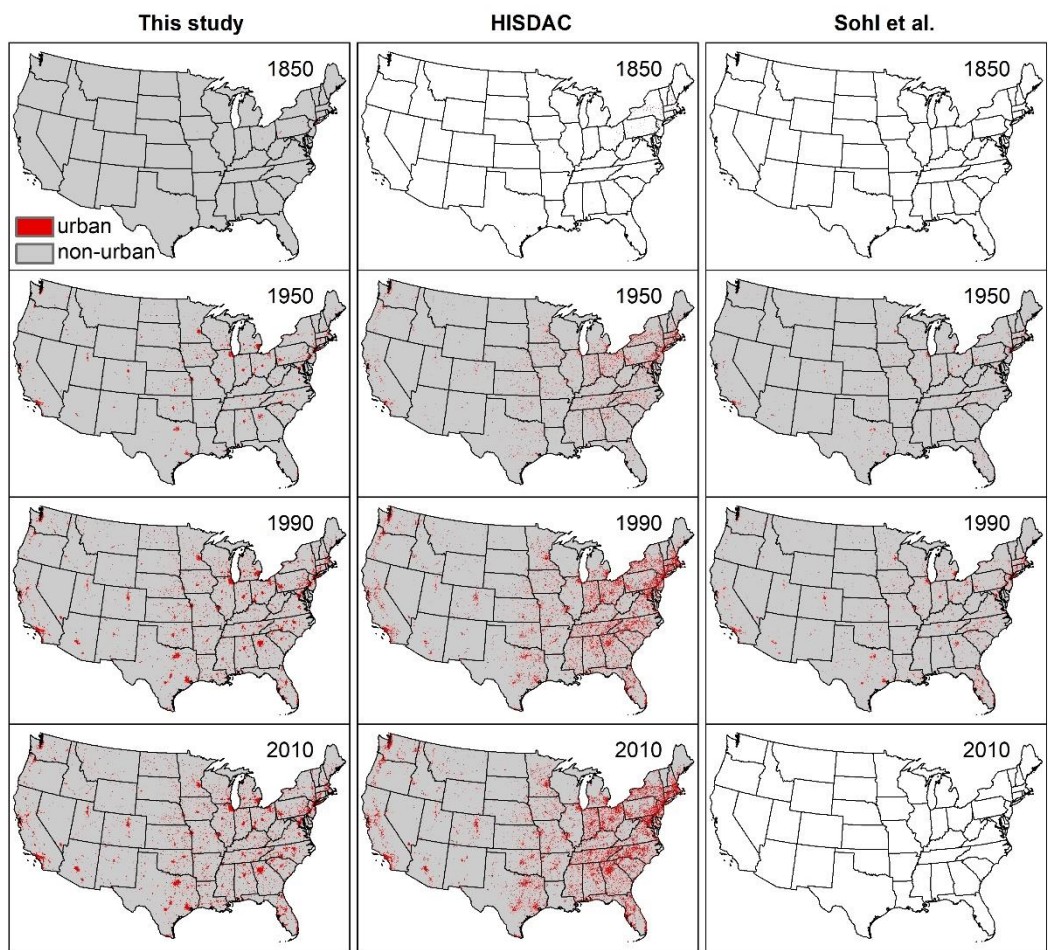


**Figure 13: Comparison of urban land maps among three data sets for the conterminous United States: this study (left column), Historical Settlement Data Compilation (HISDAC) map (central column), and Sohl et al. (2016) map (right column).**

The reconstructed cropland area was close to NLCD in the 2000s (Figure 12b). Our data and YLmap applied the cropland harvested area to estimate the historical cropland area and showed the same trend during 1850–2016 (Figure 12b). The cropland

area derived from ZCmap and ERS was higher than our data over the research period (Figure 12b) beacuse cropland harvested, crop failure, cultivated summer fallow, cropland use for pasture, idle cropland all counted (Zumkehr and Campbell, 2013; Bigelow and Borchers, 2017). The area trend between 1630 and 1879 was close to HYDE because we used its cropland per capita trend (Figure 12b). Spatially, four fractional cropland maps show the similar state and expansion patterns. The highest cropland density can be found in the Corn Belt, Central California, and Mississippi Alluvial Plain in the 2000s. Meanwhile,

cropland expansion initially occurred in the east of Mississippi River, then moved to the Midwest and the Great Plains between 1850 and 1920 (Figure 14). Our results can reflect the cropland abandonment in New England, the South, and the Southeast since the 1920s, which is consistent with in previous studies (Reuss et al., 1948; Land, 1974; Foster, 1992). Moreover, the newly developed cropland improved the spatial resolution compared with HYDE and ZCmap, making it possible to catch more

detailed information (Figure 15). In YLmap, there are some coarse grids in the early years (Figure 15) because they applied

HYDE data to reconstruct the cropland expansion and abandonment (Yu and Lu, 2018). Our data was processed at 1km
resolution and fixed this problem (Figure 15). Compared with the above cropland data, our product has higher spatial resolution
and more extended temporal coverage, making it capable of depicting the cropland dynamics better in CONUS over the past
four centuries.

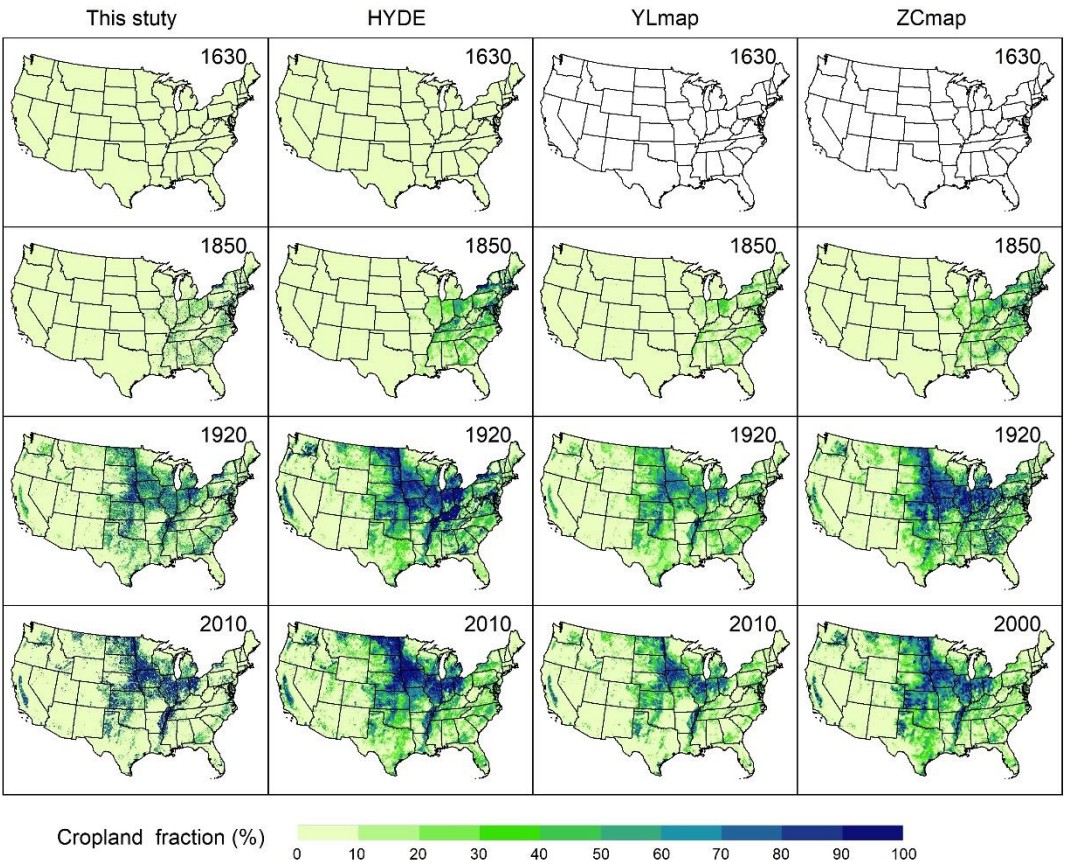

**Figure 14: Comparison of cropland maps among four datasets for the conterminous United States: this study (first column), the History Database of Global Environment (HYDE), Yu and Lu (2017) cropland density (YLmap), and Zumkehr and Campbell (2013) historical fractional cropland areas (ZCmap).**

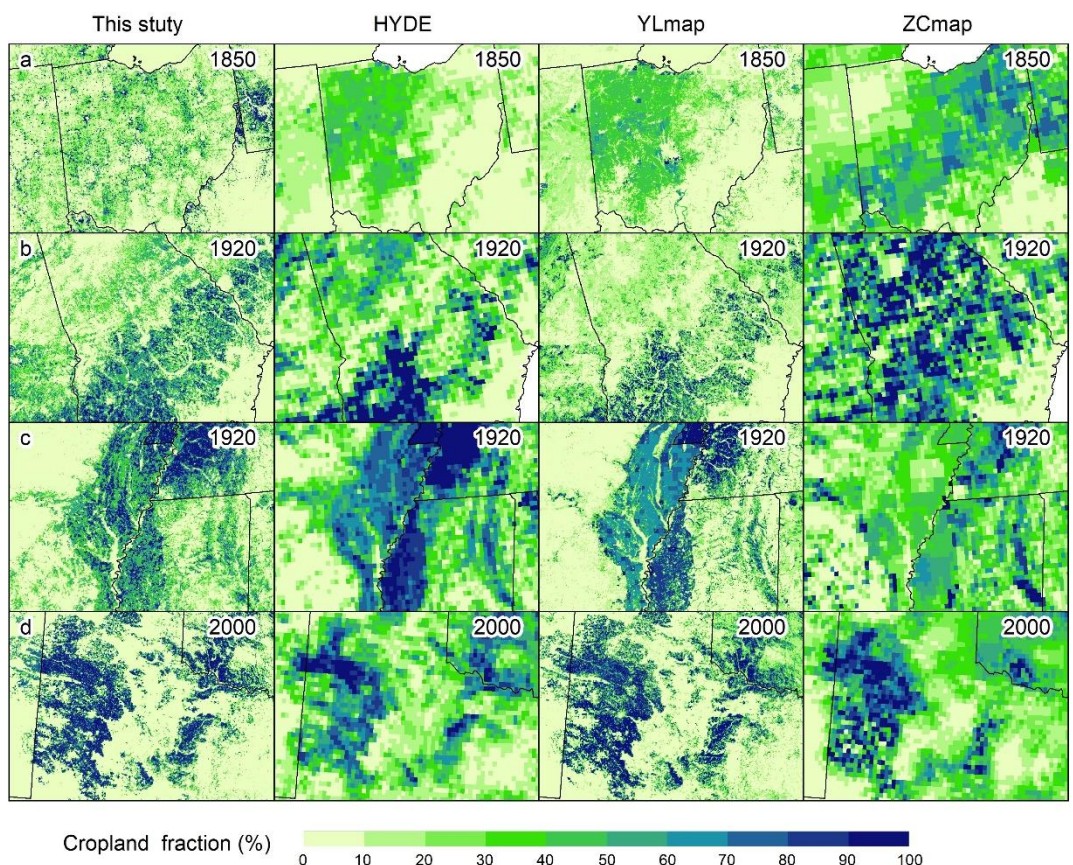

Cropland fraction (%)

0  10  20  30  40  50  60  70  80  90  100

**Figure 15: Visual comparison between our cropland data and the History Database of Global Environment (HYDE), Yu and Lu (2017) cropland density (YLmap), and Zumkehr and Campbell (2013) historical fractional cropland areas (ZCmap) in four different sites (a-d). The locations of image center points are as follows: a. Ohio (83.05 °W, 40.17 °N), b. Georgia (83.58 °W, 32.77 °N), c. Arkansas (90.56 °W, 34.76 °N), d. Texas (100.92 °W, 32.81°N).**

To the best of our knowledge, accurate temporal and spatially explicit data are still lacking to describe the pasture dynamics for the CONUS. This study set the state-level pasture area from the National Resource Inventory (NRI) as the baseline data for historical pasture reconstruction, which made our data more reliable than HYDE. During 2001–2020, the total national area of pasture located in non-federal land ranged from 48 to 53 Mha, which was close to the NLCD (53 Mha) and HYDE (52 Mha) (Figure 12c). We also found that NLCD pasture/hay decreased during 2001–2016, while NRI pasture kept relatively stable. The likely reasons for NLCD pasture/hay loss include normal crop cycling and more permanent conversion (Homer et al., 2020). The difference in pasture trends between NRI and NLCD may result from the definitional difference (Table S6). Nevertheless, Haines et al. (2018) pasture only includes hay, making it significantly lower than our result (Figure 12c). The ERS data also provided grazing land area, but the rangeland and pasture were not separated (Bigelow and Borchers, 2017). The application of the HYDE pasture per capita trend made our result close to it and reached the maximum historical value in the 1950s (Figure 12c). The three maps all show the highest pasture density in eastern Texas, Oklahoma, and Missouri on three

maps (Figure 16). At the regional scale, the spatial patterns of pasture from this study are close to the HYDE and LUH2 data,
but our data can characterize the historical changes of pasture with higher spatial resolution than current LULC products
(Figure 17).

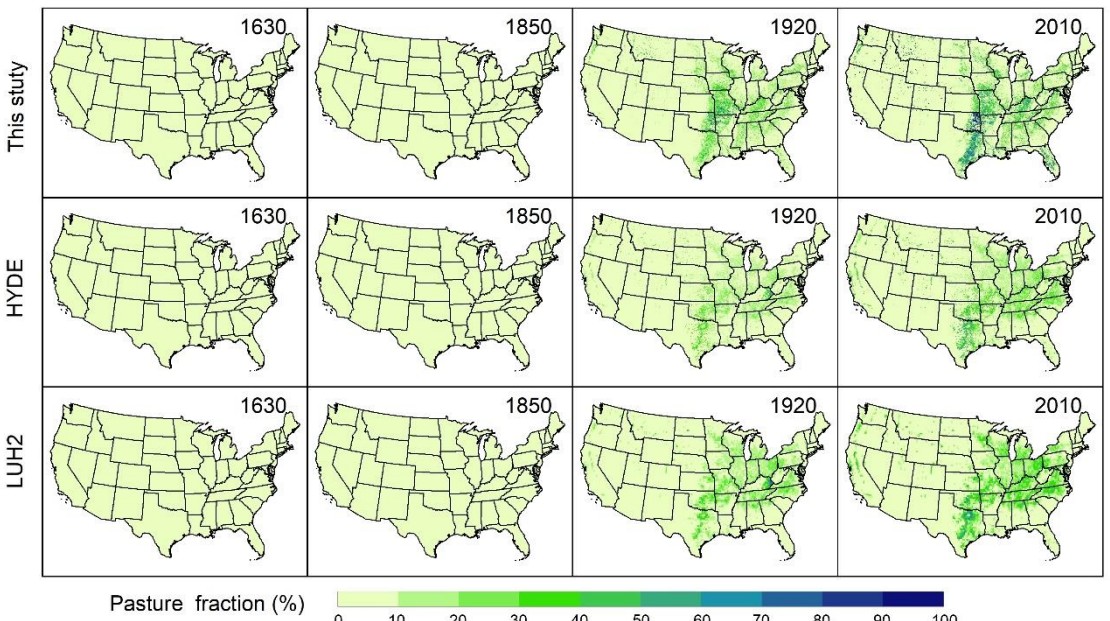

**Figure 16: Comparison of pasture patterns and changes among three data products for the conterminous United States: this study
(upper panel), the History Database of Global Environment (HYDE) (middle panel), and Land Use Harmonization (LUH2) (lower
panel).**

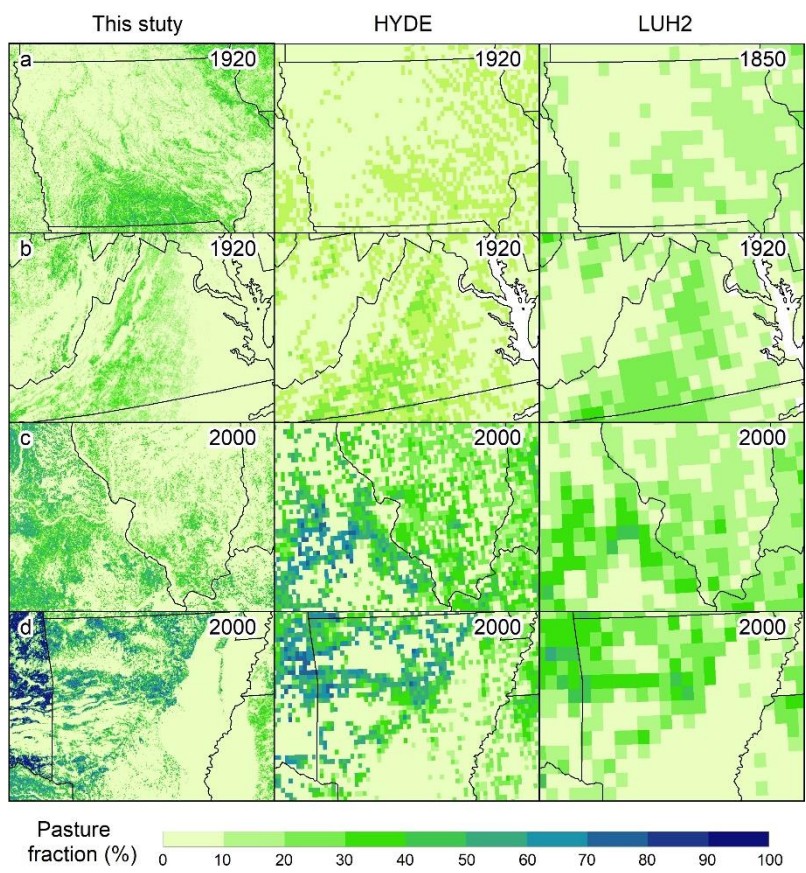

Figure 17: Visual comparison of our pasture data with History Database of Global Environment (HYDE), and Land Use Harmonization (LUH2) in four different sites (a-d). The locations of image center points are as follows: a. Iowa (93.64 °W, 42.03 °N), Virginia (78.72 °W, 37.96 °N), c. Illinois (90.07 °W, 38.68 °N), d. Arkansas (92.56 °W, 34.97 °N).

We used the inventory-based datasets (FATD and USDA-FR) to reconstruct the historical forest area. Compared with satellited-based forest (NLCD), Sohl et al. (2016), and LUH2 data, the total forest area in our data is higher. This area difference mainly resulted from the differences in forest definition. For example, NLCD and Sohl et al. (2016) define forest as the areas dominated by trees generally greater than 5 meters tall and greater than 20% of total vegetation cover, higher than that in our forest definition (forest cover greater than 10%) (Sohl et al., 2016; Homer et al., 2020; Table S7). Moreover, the forest in

LUH2 is determined by the vegetation biomass density and country-level forest area (Hurtt et al., 2020), underestimating the forest distribution in the area with low biomass density. Spatially, our data and LUH2 can describe the high density in the eastern US and Pacific Coast area, but LUH2 underestimates the forest fraction in Rock Mountain and Texas (Figure 18 and Figure 19). Our data fixed the above problem and improved the spatial resolution from 0.25 degrees to 1 km. Meanwhile, the newly developed forest data has good performance in capturing forest dynamics. For example, previous studies reported

deforestation in southern Michigan and forest cutting for agriculture and fuel in Virginia during the early settlement period (Carl, 2012; Mergener et al., 2014), also shown in our maps during the 19th century (Figure A5). Forest loss during the

westward expansion period can be captured in the Northeast, Midwest, and Great Plains (Figure 18, Figure A5). The LULC conversion map can reveal the forest regrowth on much cutover and abandoned agricultural land in the Northeast and Southeast since the 20th century (Foster et al., 1998; MacCleery, 2011) (Figure 10, Figure A5).

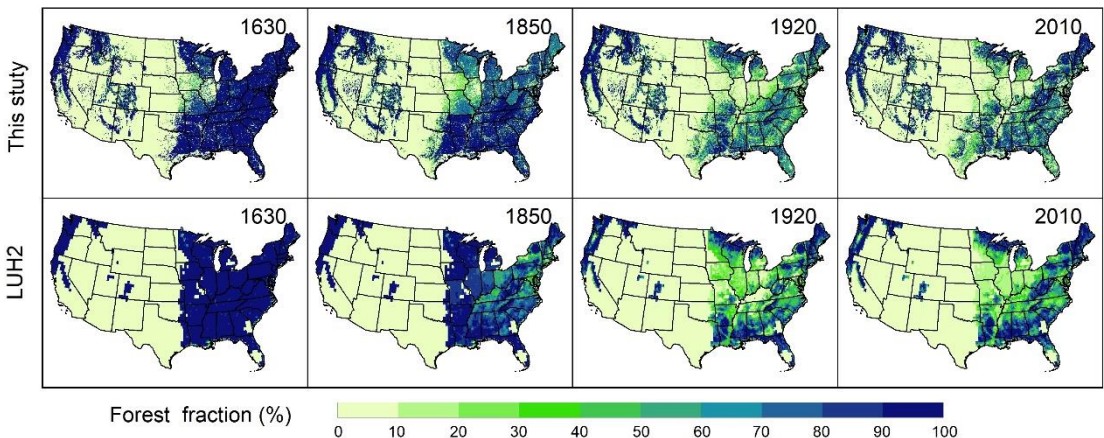


**Figure 18: Comparison of forest distribution between this study (upper panel) and Land Use Harmonization (LUH2) (lower panel) for the conterminous United States.**

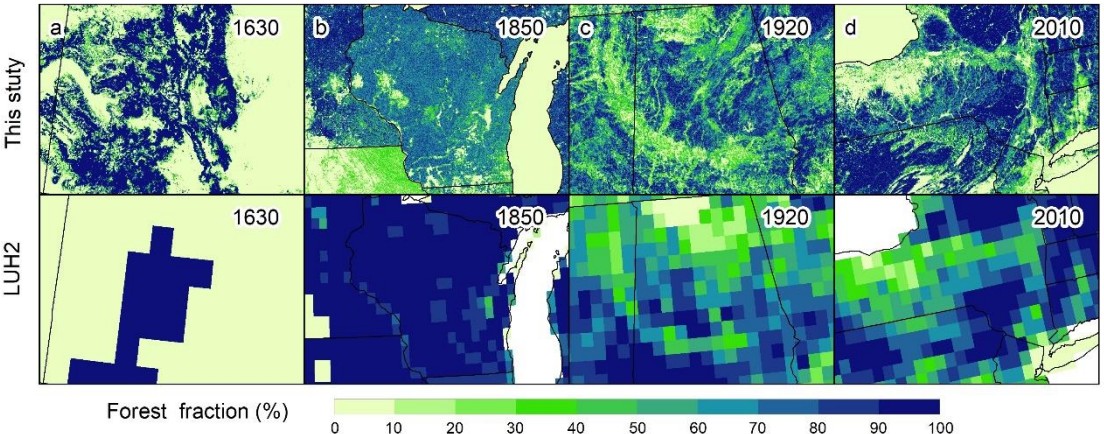

**Figure 19: Visual comparison between our forest data and Land Use Harmonization (LUH2) in four different sites (a-d). The**
**locations of image center points are as follows: a. Colorado (106.47 °W, 38.97 °N), Wisconsin (89.85 °W, 44.54 °N), c. Alabama (86.72 °W, 33.33 °N), d. New York (75.14 °W, 42.21 °N).**

## 4.2 Drivers of land use and land cover changes

Agricultural land expansion and natural vegetation loss (forest, grassland, and shrub) area is the primary characteristic of LULC change in the CONUS over the past four centuries. The complex interactions among land suitability, climate,
population, transportation, agricultural technologies, and policy shaped the contemporary LULC pattern. In the Colonial Era, the migration of Europeans into the Northeast and Mid-Atlantic converted the Eastern forests to cropland and pasture (Waisanen and Bliss, 2002). More than 90% of people lived in the east of the Appalachian Mountains, and most farms were

subsistence in this period. The forced migration of slaves contributed to the plantation agriculture expansion in Virginia, Maryland, South Carolina, and the Black Belt. After the new nation was established, numerous lands like Louisiana, Florida, Texas, Oregon, and New Mexico were acquired during 1800–1860 (Dahl and Allord, 1996; Fretwell, 1996). The westward movement opened new areas for agricultural development. With the building of canals and inland waterways, agricultural products from the cropland developed west of the Appalachians could be brought to the market (Meinig, 1993). In the second half of the 19th century, the rapid population growth and food demand resulted in the cropland expansion because farmers needed to reclaim another three to four acres to feed one person (MacCleey, 2011). After the 1920s, cropland, pasture, and forest area became relatively constant despite the growing population. The applications of hybrid crops and fertilizers and the increasing number of motor vehicles and farmer tractors improved agricultural productivity, which played an essential role in stabilizing cropland area (Waisanen and Bliss, 2002; MacCleey, 2011). Cropland abandonment in the east was affected by the fluctuations in crop prices, changes in labor markets, and competition from the high productivity in the Midwest (Hart, 1968; Williams, 1989; Bigelow and Borchers, 2017). The reversion of marginal cropland in the east and large-scale tree planting in the South contributed to the forest recovery (Clawson, 1979; Smith et al., 2001; Thompson et al., 2013). For example, many croplands in the South were abandoned following the disintegration of the post-bellum sharecropping system and later converted to plantations forest (Hart, 1968), and the plantation forest area increased from near zeros in the 1930s to 27 Mha in 2017 (Chen et al., 2017; Oswalt et al., 2019). Climate change also impacts the LULC change. For example, the Dust Bowl in the 1930s led to widespread crop failure in the Great Plains (Heimlich and Daugherty, 1991). Land marked by crop failure due to severe drought, extensive flooding, or wet weather has ranged between 5 and 22 million acres since 1949 (Bigelow and Borchers, 2017).

**4.3 Uncertainties and future perspectives**

This study provides a four-century LULC dataset at annual time step and 1 km x 1 km spatial resolution for the CONUS. However, some uncertainties may affect the accuracy of this dataset. For instance, both the reliability of input data and the harmonization method are critical for the historical LULC area reconstruction. Most census data used in this study was recorded at 4 to 10-year interval, making some interannual fluctuations impossible to capture. The rebuilt state-level LULC area is also coarse if there are significant spatial shifts (e.g., cropland abandonment in some counties but reclamation in others) for a LUCC type. Moreover, the definitional differences among datasets increased the difficulties and uncertainties in the harmonization process. Though we tried to gather the most reliable LULC datasets, the definitions of LULC vary (Table S5-S7). The definitions of four LULC types do not belong to a universal classification system, making it hard to process the total area, and a post-processing step needs to be conducted. For the urban land, HISDAC built-up area was higher than that from the NLCD dataset. However, the urban land area change rates were close during 2001–2015, indicating that there would be minor uncertainties in combining the HISDAC data and the newly developed urban land (Figure S8). We applied three datasets (i.e., ERS cropland harvested area, CAHA cropland harvested area, HYDE cropland) to generate the cropland area for the study period, but the definitions of cropland harvested area and cropland are different (Table S5). We conducted a data adjustment

to subtract the double-cropped cropland area and optimize the cropland harvested area interannual variations, but it only resulted in little change in the cropland area (2.23±1.81%) (Figure S9). The relative difference of cropland per capita change rate during 1910–2000 between HYDE3.2 cropland data and the newly developed cropland data is relatively higher than that during 2001–2017 and 1880–1909, but the values in most of the years were lower than 5% (Figure S10 and S11). For the

pasture, the state-level mean relative difference in pasture per capita change rate between HYDE3.2 pasture data and our data is 4.89±1.94% during 1982–2017 (Figure S13). Thus, though some uncertainties are introduced, it is acceptable to generate the historical LULC data.

More efforts are needed to generate accurate historical LULC maps for understanding the history of regional LULC changes. An accurate LULC probability or suitability surface is the key to generate spatial data. In this study, we assumed that the ANN-

based LULC probability was unchanged following previous LULC simulation models (Verburg et al., 2006, 2009; Sohl et al., 2014; Liu et al., 2016). However, we found that the contemporary probability surfaces could not represent the historical LULC pattern, especially for agricultural land (Sohl et al., 2016). To solve the problem, we modified the LULC probability by using population density, human settlement extent, and satellite observed LULC fraction, making it match the historical LULC pattern. The LULC pattern is highly related to that in the previous year, and the grid value is also affected by the fraction and

type in the neighbor grid cells. In our spatial allocation strategy, we generate the LULC map for each year based on the LULC probability or LULC fractional data, which ignores the LULC pattern interactions between the adjacent years. Some studies generated a LULC map by allocating the LULC net change area to a base map (West et al., 2014; Liu et al., 2016; Cao et al., 2021), but such an algorithm would underestimate the LULC gross change (Winkler et al., 2021). Therefore, an improved spatial allocation strategy should be developed to simulate LULC conversion better.

The newly developed LULC dataset reconstructed the LULC history with more LULC types than ZCmap and YLmap and has higher spatial resolution than HYDE and LUH2. Our LULC data emphasizes the accuracy of area change resulting from LULC conversion rather than the changes in LULC structure or attributes. For example, forest management (e.g., wood harvest and thinning) results in the forest cover decreases and ecosystem function change, but the LULC type is unchanged. HYDE and LUH2 not only have a more extended cover period, but also provide more sub-types and LULC attributes. HYDE classified

cropland into rain-fed rice, irrigated rice, rain-fed other crops, and irrigated other crops (Goldewijk et al., 2017). LUH2 divides cropland into C3 crops and C4 crops and includes the wood harvest (traditional fuelwood, commercial biofuels, and industrial roundwood) and primary/secondary forest age (Hurtt et al., 2020). In the future, the LULC sub-types (e.g., tree species, crop types) and attributes (e.g., forest age, management intensity) through collecting from agricultural census data and forest inventory data can be incorporated into our dataset (Thompson et al., 2013; Chen et al., 2017; Crossley et al., 2021).

**5 Data availability**

The land use and land cover datasets for the conterminous United States are available at https://doi.org/10.5281/zenodo.7055086 (Li et al., 2022). The annual gridded datasets (1km x 1km spatial resolution) with

GeoTiff format include fractional and Boolean types. An Excel table is used to organize the annual urban, cropland, pasture, and forest area at the state level. A detailed data description is also provided.

## 6 Conclusions

This study developed spatially-explicit LULC data at a spatial resolution of 1 km x 1 km and an annual time scale in the CONUS during 1630–2020 by integrating multisource datasets. The results showed that extensive cropland and pasture expansion and natural vegetation loss occurred from 1630 to 2020 in the CONUS. New reclaimed cropland was primarily converted from forest, shrub, and grassland. Tree planting and forest regeneration increased the forest cover in the Northeast and the South in the recent century. Compared to other LULC datasets, our data provided more accurate information with higher spatial and temporal resolution and better captured the characteristics of LULC changes. The LULC data can be used for regional studies on various topics, including LULC impacts on regional climate, ecosystems, biodiversity, water resource, carbon and nitrogen cycles, and greenhouse gas emissions.

## Appendices

**Table A1: Spatially explicit variables adopted for artificial neural network (ANN) modelling.**

| Variable | Description | Source | Resolution |
|---|---|---|---|
| Elevation | Digital elevation model (DEM) | Shuttle Radar Topography Mission (STRM) | 90 m |
| Slope | Slope calculated from DEM | (https://cgiarcsi.community/data/srtm-90m-digital-elevation-database-v4-1/) | |
| Pop | Population density | Fang and Jawitz (2018) (http://doi.org/10.6084/m9.figshare.c.3890191) | 1 km |
| City$_{dis}$ | Distance to city | https://www.sciencebase.gov/catalog/item/537d23fee4b00e1e1a484c82?community=Data+Basin | vector |
| Road$_{dis}$ | Distance to road | | vector |
| Railway$_{dis}$ | Distance to railway | | vector |
| River$_{dis}$ | Distance to river | North America River and Lakes (https://www.sciencebase.gov/catalog/item/4fb55df0e4b04cb937751e02) | vector |
| Soil clay | Soil texture clay fraction | Soil Grids 250 m v2.0 | 250 m |
| Soil sand | Soil texture sand fraction | https://soilgrids.org/ | 250 m |
| Soil SOC | Soil organic carbon | | 250 m |
| Crop PI | Crop productivity index | Soil Survey Geographic (SSURGO) Data | 250 m |
| PPT | Precipitation | | 800 m |
| TMP | Mean temperature | PRISM (https://prism.oregonstate.edu/recent/) | 800 m |
| Max TMP | July temperature | | 800 m |
| Min TMP | January temperature | | 800 m |

**Table A2: Land use and land cover datasets used for comparison.**

| Data variables | Time period | Resolution | Data sources |
| --- | --- | --- | --- |
| ERS Major land uses | 1945-2012 | State-level 4 to 5-year interval | Major Uses of Land in the United States, 2012. https://www.ers.usda.gov/data-products/major-land-uses/ |
| Land Use Harmonization (LUH2) | 1600-2020 | 0.25 degree Annual | https://luh.umd.edu/ |
| Cropland density (YLmap) | 1850-2016 | 1 km Annual | Yu and Lu (2017). https://doi.pangaea.de/10.1594/PANGAEA.881801 |
| Historical fractional cropland areas (ZCmap) | 1850-2000 | 5 arcmin Annual | Zumkehr and Campbell (2013). https://portal.nersc.gov/project/m2319/ |
| Hay area | 1840-2012 | County-level 10-year interval | Haines et al. (2018). https://www.icpsr.umich.edu/web/ICPSR/studies/35206# |
| Historical LULC dataset | 1938-1992 | 250 m Annual | Sohl et al. (2018) https://www.sciencebase.gov/catalog/item/59d3c73de4b05fe04cc3d1d1 |
| Crop area | 1840-2017 | County-level 10-year interval | Crossley and Michael (2020). https://www.openicpsr.org/openicpsr/project/115795/version/V3/view |

Note: ERS: Economic Research Service, U.S. Department of Agriculture; YLmap: cropland density (Yu and Lu, 2017); ZCmap: historical fractional cropland areas (Zumkehr and Campbell, 2013).


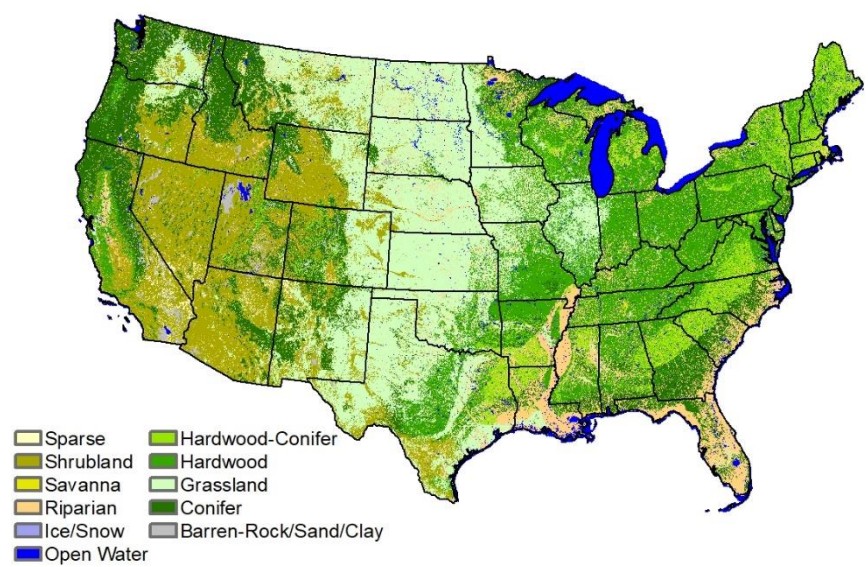

Sparse
Shrubland
Savanna
Riparian
Ice/Snow
Open Water
Hardwood-Conifer
Hardwood
Grassland
Conifer
Barren-Rock/Sand/Clay

**Figure A1: Vegetation type pre-Euro-American settlement.**


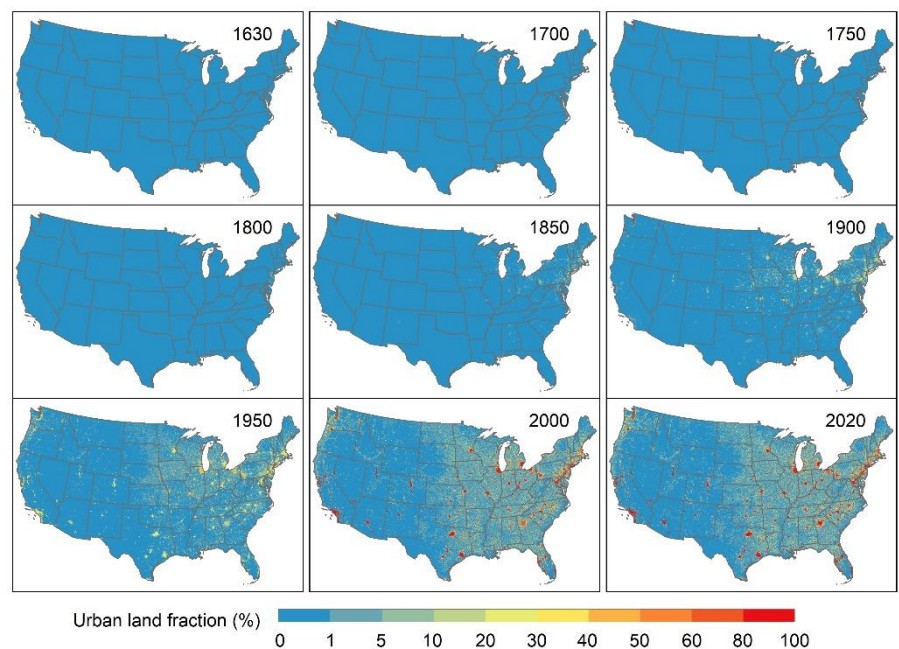

**Figure A2: Fractional urban land in the conterminous United States during 1630–2020.**

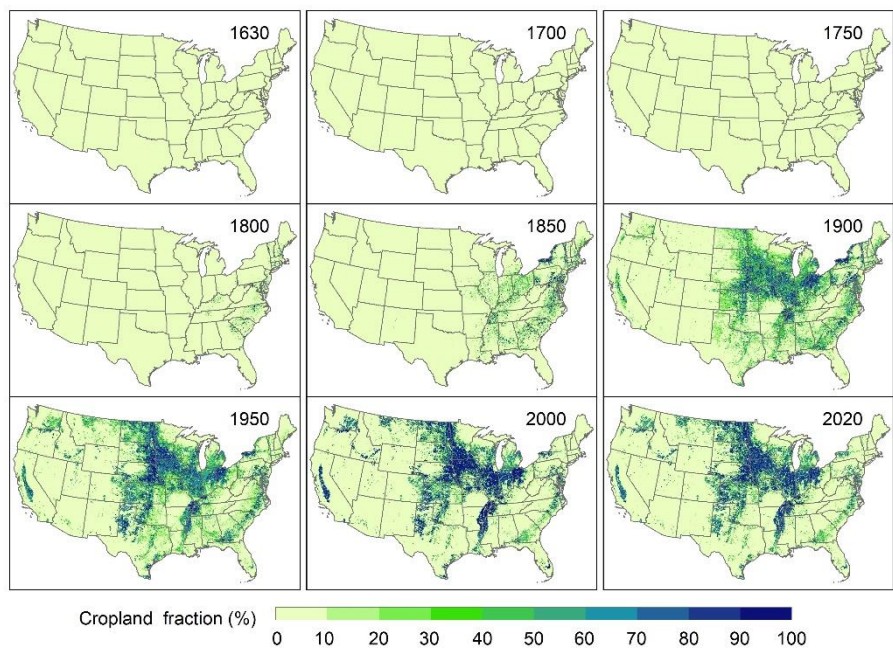

**Figure A3: Fractional cropland in the conterminous United States during 1630–2020.**

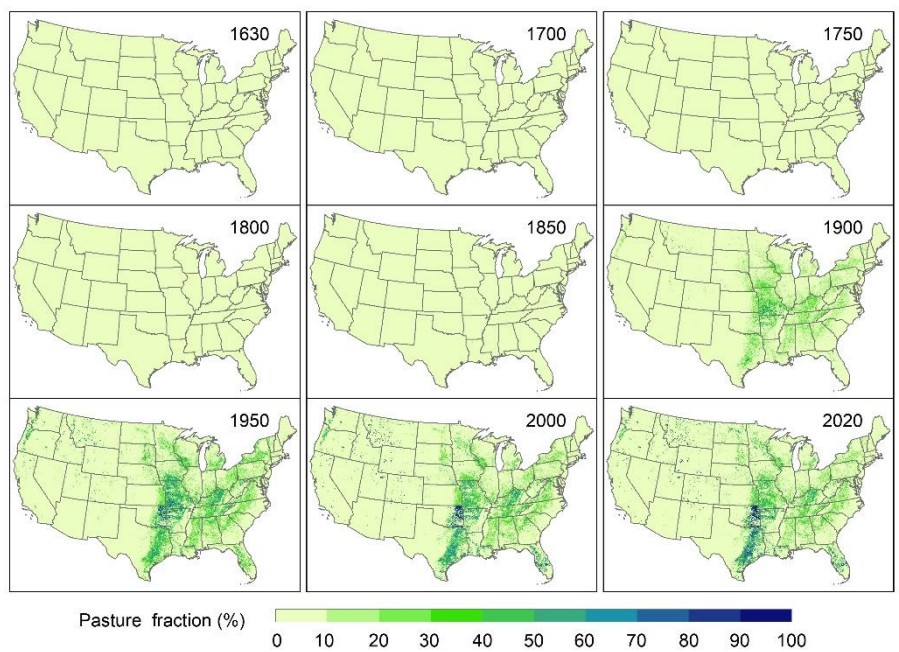

**Figure A4: Fractional pasture in the conterminous United States during 1630–2020.**

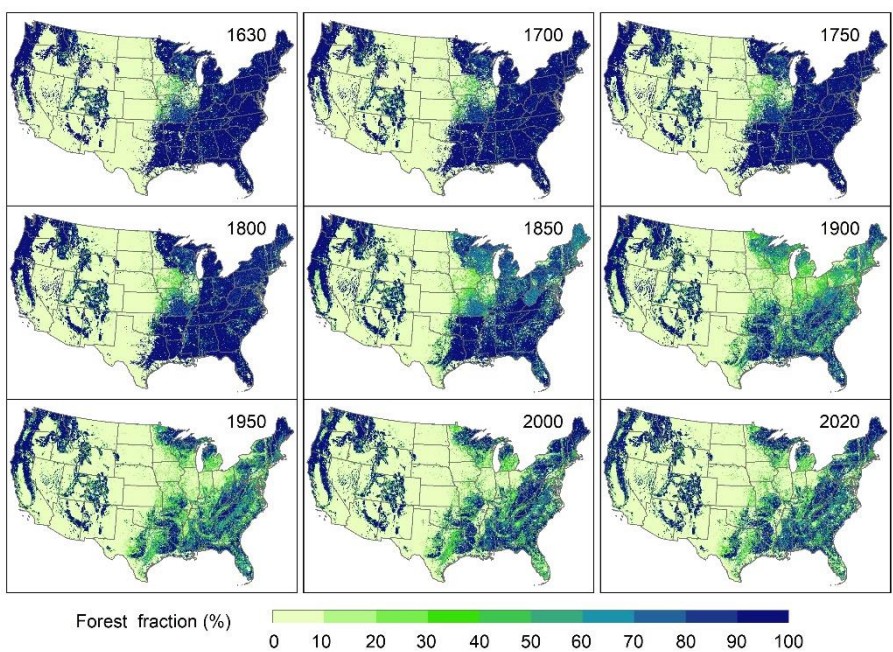

**Figure A5: Fractional forest in the conterminous United States during 1630–2020.**

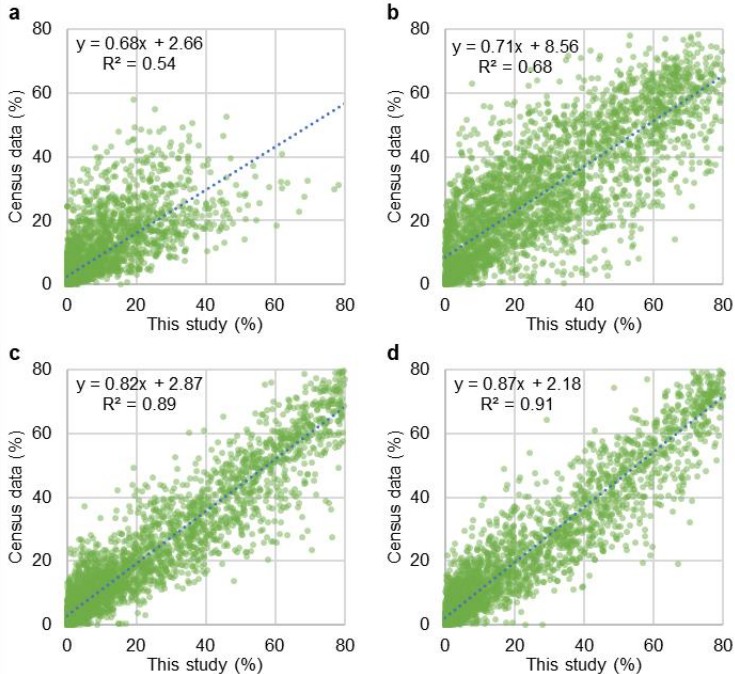

**Figure A6: Statistical comparison between cropland area from this study and census data in 1850, 1920, 1959, and 2002.**

**Author contributions**

HT designed the research; XL implemented the research and analyzed the results; XL, HT, SP, and CL wrote and revised the manuscript.

**Competing interests**

The authors declare that they have no conflict of interest.

**Acknowledgments**

We thank the anonymous reviewers for their very constructive comments that have helped us to significantly improve this work.

**Financial support**

This study has been supported in part by the National Science Foundation (grant nos. 1903722 and 1922687), the National Oceanic and Atmospheric Administration (grant nos. NA16NOS4780204 and NA16NOS4780207), the National Aeronautics

and Space Administration (grants nos. NNX12AP84G, NNX14AO73G, and NNX10AU06G) and the Department of the Treasury in cooperation with the State of Alabama Department of Conservation and Natural Resources (DISL-MESC-ALCOE-630 06). XL acknowledges a graduate fellowship from the University of Chinese Academy of Sciences for a collaborative training program at Auburn University.

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
