# Peer review of "Four-century history of land transformation by humans in the United States (1630-2020): Reconstructing annual and 1-km grid data for the HIStory of LAND changes (HISLAND-US)"

_Earth System Science Data, 2022_

## Referee Comment (RC1)

**Four-century history of land transformation by humans in the United States: 1630-2020**

**ESSD-2022-135**

The paper takes on the very substantial challenge of recreating historical land use for the United States. The authors are correct that outside of coarse-level reconstructions such as HYDE, that there is nothing of higher resolution that goes back to pre-settlement by European colonists. The authors pitch this methodology and dataset as the solution to that data gap.

I don't think the authors are completely successful in making that argument. After stating the need for higher-resolution data, and presenting a methodology that should allow for some higher-level of spatial detail than something like HYDE, the authors inexplicably leave out any kind of spatial analysis of how well their model result performs. It's difficult when spatially explicit data on land use is hard to come by (and hence the need for this work!), but some analysis of the full-resolution data compared to a data set such as NLCD (available 2001 to 2019) could have helped establish confidence in the model to capture spatial patterns well. Even a comparison to historical county-level data would help. However, the only "spatial" analysis of the data are some very coarse regional assessments that don't provide a reader much of a feel for the model's capability for generating realistic, high-resolution spatial patterns.

Validation overall is a weak point of the paper. On the one hand, I understand the difficulties in trying to "validate" results such as this, when consistent reference data is absent or scattered. However, it's not acceptable from a modeling perspective to use HYDE, NLCD, and other data to parameterize the model, and also use those data in what's labeled as "validation" of results. Given the lack of spatially explicit reference data, I have no issue with the authors doing "consistency checks" with other data sets, including missed opportunities such as county-level ag census data that could have been used to provide a better feel for the spatial patterns produced by the model. But don't try to sell it as a real "validation" of model results.

Overall, the authors continually note the "Uncertainties" associated with coarser data such as HYDE, but fail to conclusively demonstrate their results are superior in terms of those uncertainties.

A major need for the paper is recognition of the differences between source datasets, and the uncertainties it introduces into the modeling. For example, many parts of the model are parameterized by using multiple datasets that have inherent differences. Pasture, for example, is quantified at a state level by NRI, and then HYDE for older dates. Given how variable definitions are for "pasture" in the first place, it's asking for trouble to mix and match datasets such as that. The authors do seem to make some attempts to harmonize differences in datasets, but the methodology isn't well-defined enough to let me know if that's really being done. In short, historical land use reconstruction is difficult because 5 different datasets may give you 5 different answers for how much "forest", "urban", "cropland", or "pasture" is actually there for a given date!

The spatial allocation of "change" on the landscape is also quite simplistic, based solely on probability surfaces with no stochasticity. The authors do attempt to mitigate the 'static probability surface' problem present for past applications such as CLUE and FORE-SCE by some simple weighting with population. But the actual allocation is fully dependent on the probability surfaces at the end of the day,

and as a result, as you go back in time, you tend to see classes such as cropland and pasture become concentrated in the high-probability locations, with less fragmentation that's there in later dates.

Finally, note I did download and look at some of the output results. For brevity I'll keep my comments here to the "Boolean" land cover. Note that while results look reasonable at broad scales, the approach of parameterizing state-by-state does seem to cause some issues as you go back in time, as does the spatial allocation methodology. Going back in time reveals a number of obvious state and even what appear to be county boundaries, hard obvious lines where land use clearly differs on either side of a political boundary. Given the complete reliance on probability surfaces alone for the spatial allocation of change, land use looks more concentrated on the landscape for some classes going back in time. For example, on the 1850 map, cropland is concentrated in very large contiguous chunks in many areas, and very sharp and obvious political boundaries are present.

Individual comments follow.

**Specific Comments**

- Lines 28 – Would add one word…"In particular, _managing_ agriculture and forest-related activities…"
- Line 34 – Would add words…"…arrival of Europeans, indigenous communities practiced agriculture and crop planting in the…"
- Line 36 – Would change "mainly occurred" to "initially occurred" to indicate these activities first started here, but expanded elsewhere later (as noted by the next sentences)
- Line 37 – "Driven" not "driving".
- Page 2 – Overall the paragraph at the top of page 2 could use some work. It's a rather disjointed history of US land change. For one it doesn't really talk about land change west of the Mississippi River, it's focused solely on Eastern US change. The organization is also a bit odd and disjointed. The sentence on line 44, for example, seems like a very abrupt and odd ending to the final statement as to why a long-term land use dataset is needed. Perhaps a better organization by period (colonial, 19th century, 20th century), with a description of what occurred in each century? And perhaps a modification of the last sentence, adding "While general trends in historical US landscape change are known, we still lack a long-term dataset…"
- Line 55-56 – Be careful about highlighting "uncertainties" in datasets such as HYDE, as your historical landscape construction will also have substantial uncertainties. Your workflow itself uses HYDE data. There's limited spatially explicit data available from which to base a model-based landscape reconstruction, and many of the datasets you're using were also used by HYDE.
- Lines 87-88 – I wouldn't call the use of these other datasets "validation". It's a consistency check, not a validation, as these data sets too have uncertainties, and some are modeled just as you're modeling.
- Line 90-91 – How was resampling done to get to 1-km grid cells? Is it fractional LULC within a given 1-km cell for datasets with native resolution <1 km?
- Section 2.2.1 – This is an extremely simplistic methodology for calculating urban land area. To start, it's all based on one current dataset, NLCD. How was NLCD used? First of all, NLCD tends to underestimate low-density residential lands, which can bias your results. Secondly, NLCD "urban" classes also include extensive representation of road networks, which if counted as "urban", greatly overestimates urban land. For a rural state, for example, NLCD classes not only

major roads, but every small section road has a 1-pixel-wide "urban" class representing it. Unless measures were taken to account for NLCD's underrepresentation of low-density residential lands, and to account for all the "urban" pixels that are really roads, it biases the results.

The other problem is the very simplistic method for calculating land area. You're assuming the relationship between urban land per capita and total population is constant through time. Clearly it's not. Without accounting for that changing relationship, urban estimates can easily be biased.

- Section 2.2.2 – You're using (at least) three different data sources to help establish cropland area. For historical land use, estimates vary widely, dependent upon methodology, data source, thematic definitions of a land use, etc. As a result, when switching from USDA-based data, for example, after 1889, and using HYDE before 1889, you'd expect an obvious break in estimated "cropland" amounts. How were those inconsistencies among historical land use datasets harmonized?
- Paragraph starting on line 120 – You're assuming the relationship between harvested area and planted area from 1978 to 2017 is consistent decades and centuries before those data…a very dangerous assumption.
- Line 123 – Yet another dataset, Borchers et al. 2014, was used to establish double-cropping at a regional level.  Again, consistency among most of these datasets isn't great.
- Line 124-125 – Another basic assumption that likely isn't true through time.
- Line 125-126 – Because the 1879 number was different you assumed it was incorrect? But data >1889 were "correct"? Was the reconstructed cropland area in 1879 substantially lower or higher than 1889?
- Line 130-131 – Another basic assumption that likely doesn't hold region to region.
- Line 132-133 – Again…how did you account for differences in the HYDE data, and the (mostly) USDA-based data after 1889?  Is there an obvious break in cropland amount pre- and post-1889?
- Line 135-136 – One of the greatest difficulties in historical landscape reconstruction is the definition of "pasture", vs. "grassland", vs. "hay" vs. "rangeland", etc.  There is no one definition that's universally accepted. Your definition here states Pasture includes areas "for the production of seed or hay crops".  Many definitions of "cropland" include alfalfa, hay, and other crops in "planted area" or "cultivated crop" area.  For "Pasture", you're introducing yet another completely new dataset to establish pasture area, NRI.  Are the definitions of "pasture" for NRI the same for NLCD, HYDE, and the US Census of Ag?
- Line 139 – Note Wasianen and Bliss took great pains to harmonize those definitional differences across their harmonized dataset.
- Section 2.2.3 – Again…it's extremely simplistic to assume things such as "pasture per capita" and that that ratio is consistent over time, and space.
- Section 2.2.4 – Definitions of what is "Forest" vary greatly among data sets. You're introducing yet another data set in FIA that may have a definition of "forest" that differs from HYDE or from NLCD. How closely does the FATD data match with HYDE estimates, for example?
- Lines 156-157 – Yeah you lost me here with what you're trying to do, needs a better explanation.

- Section 2.2.5 – See main comments above related to how you balanced the four LULC classes.
- Line 195 – What was used to establish the "land use change boundary"? That is, what was the source of "settled area" data"?

  I don't mind the use of something like this to constrain the allocation of change, but do wonder about full-resolution results. Are there any hard border issues obvious in the data when change occurs at the edge of those defined boundary layers? Overall with the boundary and effect of population density, I appreciate you trying something other than assuming a static probability surface through time.
- Section 2.3.2 – There needs to be more explanation here. You've basically summarized the entire actual allocation to the pixel level in one sentence. I certainly get that higher probability areas will likely have a higher proportion of a given LULC class, but it's all deterministic and it's all based solely on the probability surface? There's no stochasticity? With such a sparse description of methodology, it's also hard to see how this simple description of the methodology ends up with the aggregate totals from the allocation stage matching the quantitative estimates you established for each of the LULC classes.
- As noted in the main comments, I have other concerns about the allocation strategy.
- Section 2.4 – Comparison to other LULC datasets isn't a validation, it's a consistency check. That's particularly true when every dataset has it's own production methodologies, data sources, and thematic definitions, all of which makes even direct comparison problematic.

  Beyond that, you're comparing your results to some of the same datasets from which you parameterized your modeling, as noted in the overarching comments. Also note there aren't any details as to what methodologies you're actually using for "validation" in this very short, one-paragraph section.
- Section 3.1 – As noted previously, this isn't very useful for inferring confidence in your results, when you're using the same datasets to parameterize the model as you are to "validate" model results.
- Line 273 (and throughout the results section) – If you're going to refer to a specific driving force of change, and, for example, point to a specific policy, you should name the policy and reference it (Immigration and Naturalization Act of 1965). While it certainly did change the nature of immigration to the country, you do give it too much focus as "the" causes of urban land increases after 1965. There's a lot more at play there than immigration policy.
- Lines 276-277 – You state "cropland area did not change significantly" from 1930 to present day. First, it's always problematic to use the term "significantly" in a journal paper, given the scientific meaning of the word. Secondly, I would argue there were "substantial" trends in agriculture after 1930, including some of those you mention (e.g., biofuel impacts).
- Figure 7 – On a national-scale map figure, it's difficult to see patterns of the individual land use transitions. Perhaps it would be augmented by a complementary confusion matrix of changes or some other tabular data approach that allows you to see (and easily quantify) transition types.
- Section 3.4 – This isn't the most effective section to me. As noted in the main comments, a major premise of the paper was to provide a "high resolution" historical landscape reconstruction for the US. Much of the "regional" information here is also discussed in the

overall results above.  I'd have much rather seen some real examples (and preferably validation) of landscape pattern at finer scales, given the focus on higher resolution with this paper.

- Lines 338-339 – Agreed about the "per capita" approach.
- Lines 339-341 – This doesn't serve as any kind of adequate validation or even consistency check between datasets. Showing a national-scale map and stating the patterns are "consistent" isn't valuable, and is very subjective at that scale.
- Line 357-358 – Exactly why it's not very valuable to compare your model results to HYDE…those data were used to help establish the model parameters themselves.
- Line 362 – Your product has higher spatial resolution than something like HYDE, but there's no quantitative analysis of that spatial pattern that proves the superior value of that higher native resolution.
- Figure 9 – It is difficult to compare all of these datasets given the definitional differences between them, particularly for pasture and cropland.
- Lines 382-383 – I'm not sure it's more "reliable", as sample-based, inventory approaches have flaws, just as satellite-based approaches have flaws. The bigger concern to me are the definitional differences, not the methodological differences.
- Section 4.2 doesn't add a lot to the paper for me, particularly since you've already tried to explain some driving forces in the previous paragraphs of the paper. I'd much rather have the drivers woven into the story of what's happening in your results, than as a separate section.
- Lines 435-436 – I think reconstruction of historical land use is limited more by reliable, consistent historical data than methodology. Machine learning methods aren't going to be that valuable for historical reconstruction given the paucity and inconsistency of historical data for training.
- Section 4.3 – Somewhere in here you absolutely need to highlight the difficulties with trying to harmonize data sets with different definitions, data sources, and methodologies.

---

## Author Comment (AC1)

Thank you for the comments and suggestions. These comments were very helpful for revising and improving our paper. We have responded to the comments point by point.

**General comments:**

The paper takes on the very substantial challenge of recreating historical land use for the United States. The authors are correct that outside of coarse-level reconstructions such as HYDE, that there is nothing of higher resolution that goes back to pre-settlement by European colonists. The authors pitch this methodology and dataset as the solution to that data gap. I don't think the authors are completely successful in making that argument. After stating the need for higher-resolution data, and presenting a methodology that should allow for some higher-level of spatial detail than something like HYDE, the authors inexplicably leave out any kind of spatial analysis of how well their model result performs. It's difficult when spatially explicit data on land use is hard to come by (and hence the need for this work!), but some analysis of the full-resolution data compared to a data set such as NLCD (available 2001 to 2019) could have helped establish confidence in the model to capture spatial patterns well. Even a comparison to historical county-level data would help. However, the only "spatial" analysis of the data are some very coarse regional assessments that don't provide a reader much of a feel for the model's capability for generating realistic, high-resolution spatial patterns. Validation overall is a weak point of the paper. On the one hand, I understand the difficulties in trying to "validate" results such as this, when consistent reference data is absent or scattered. However, it's not acceptable from a modeling perspective to use HYDE, NLCD, and other data to parameterize the model, and also use those data in what's labeled as "validation" of results. Given the lack of spatially explicit reference data, I have no issue with the authors doing "consistency checks" with other data sets, including missed opportunities such as county-level ag census data that could have been used to provide a better feel for the spatial patterns produced by the model. But don't try to sell it as a real "validation" of model results.

Overall, the authors continually note the "Uncertainties" associated with coarser data such as HYDE, but fail to conclusively demonstrate their results are superior in terms of those uncertainties. A major need for the paper is recognition of the differences

between source datasets, and the uncertainties it introduces into the modeling. For example, many parts of the model are parameterized by using multiple datasets that have inherent differences. Pasture, for example, is quantified at a state level by NRI, and then HYDE for older dates. Given how variable definitions are for "pasture" in the first place, it's asking for trouble to mix and match datasets such as that. The authors do seem to make some attempts to harmonize differences in datasets, but the methodology isn't well-defined enough to let me know if that's really being done. In short, historical land use reconstruction is difficult because 5 different datasets may give you 5 different answers for how much "forest", "urban", "cropland", or "pasture" is actually there for a given date! The spatial allocation of "change" on the landscape is also quite simplistic, based solely on probability surfaces with no stochasticity. The authors do attempt to mitigate the 'static probability surface' problem present for past applications such as CLUE and FORE-SCE by some simple weighting with population. But the actual allocation is fully dependent on the probability surfaces at the end of the day, and as a result, as you go back in time, you tend to see classes such as cropland and pasture become concentrated in the high-probability locations, with less fragmentation that's there in later dates.

Finally, note I did download and look at some of the output results. For brevity I'll keep my comments here to the "Boolean" land cover. Note that while results look reasonable at broad scales, the approach of parameterizing state-by-state does seem to cause some issues as you go back in time, as does the spatial allocation methodology. Going back in time reveals a number of obvious state and even what appear to be county boundaries, hard obvious lines where land use clearly differs on either side of a political boundary. Given the complete reliance on probability surfaces alone for the spatial allocation of change, land use looks more concentrated on the landscape for some classes going back in time. For example, on the 1850 map, cropland is concentrated in very large contiguous chunks in many areas, and very sharp and obvious political boundaries are present.

**Response**: Thank you for the comments and suggestions. According to your suggestions, we conducted another two data comparisons to increase the newly developed LULC data's confidence in section 3.1.

First, we did a comparison between our reconstruction with USDA county-level agricultural data for cropland in 1850, 1920, 1960, and 2002.

An accurate cropland distribution map is quite critical for historical LULC reconstruction. To prove the reliability of the newly developed cropland, we compared it with county-level census data. The spatial pattern of cropland proportion (cropland area/county area) from this study is close to the census data in the four years. Both two datasets show the cropland expansion in the North Central, the Great Plains, the Mississippi Alluvial Plain, and California. The cropland abandonment can also be found in the Appalachian Mountains. In the early period (1850), our results and census data show the cropland distribution in the central area of Alabama and Georgia, New England, and the North Central. But our reconstruction overestimated the cropland fraction in the North Central states, the east area of Virginia, North Carolina, South Carolina, and the south of Georgia. Because the census data only recorded the area of 18 major crops, our data is higher than the census data in the county with high cropland area. For example, cropland derived from this study was higher than the census data in the southeast coast, Atlantic coast, the Mississippi Alluvial Plain, northwest area of Texas, west of Oklahoma, and California in 1920, 1959, and 2002. While the cropland proportion in the Appalachian Mountains and the south area of the Great Plains was lower than the census data. This underestimation may result from the low cropland fraction in satellite data because it is hard for satellite data to extract the small area cropland patch in the mountain area and classify the pasture/grassland with cropland in the south of the Great Plains.

[Figure]

**Figure R1.** Spatial comparison of county-level cropland proportion between our reconstruction and census data in 1850, 1920, 1959, and 2002. First column: cropland proportion from census data; Second column: cropland proportion derived from this study; Third column: cropland proportion between this study and census data.

We also compared our data with NLCD data at the grid level and calculate the difference. The spatial patterns of urban, cropland, pasture, and forest in this study are close to the satellite-based LULC data from NLCD, and most grids have a relatively small difference. For urban land, our result overestimates the fraction in the low urban density area, but about 90% of urban land grids whose differences range from -10% to 10%. For cropland, the area with a positive difference is mainly distributed in the Northeast states, Alabama, and Missouri, in which 67% of grids have small differences values lower than 10%. 39% of grids have negative difference values and are mainly located in the states with high cropland area. Moreover, most states in our reconstruction have a lower pasture fraction than NLCD data except for Oklahoma, Arkansas, Texas, and Georgia. The grids with negative differences account for 49%. For forest, the reconstructed forest has a higher density than NLCD in the South, Pacific coast, and Great Lakes but underestimates the forest fraction in the central

states, such as Missouri, Kentucky, and Ohio. There are 73% grids whose differences are relatively small and range from -10% to 20%.

[Figure]

**Figure R2:** Spatial comparison between our reconstruction and satellite-based urban, cropland, pasture, and forest. First column: Reconstructed data in this study (average between 2001 and 2019); Second column: Satellite-based data (average between 2001 and 2019); Third column: Difference between first column and second column; Fourth column: Distributions of fraction difference between our reconstructed database and satellite-based data.

As you said, the definitional differences did increase the uncertainties of LULC modeling because we had to harmonize multiple datasets for each LULC type. Thus, we summarize and compare the definitions for cropland, pasture, and forest from the multisource dataset, please see Table R1, R2, R3. Moreover, to make the reader understand the urban, cropland, pasture, and forest reconstruction process easily, we revised the method described in section 2.2.

**Table R1:** Definition of cropland in different data sources.

| Data source | Definition |
|---|---|
| USDA-ERS | *Cropland*: Total cropland includes five components: cropland harvested, crop failure, cultivated summer fallow, cropland used only for pasture, and idle cropland (https://www.ers.usda.gov/data-products/major-land-uses/glossary/#croplandforcrops). |
| USDA-NRI | *Cropland*: A land cover/use category that includes areas used for the production of adapted crops for harvest. Two subcategories of cropland are recognized: cultivated and non-cultivated. Cultivated land comprises land in row crops or close-grown crops, as well as other cultivated cropland; for example, hayland or pastureland that is in a rotation with row or close-grown crops. Non-cultivated cropland includes permanent hayland and horticultural cropland (USDA, 2020). |
| NLCD | *Cultivated Crops*: areas used for the production of annual crops, such as corn, soybeans, vegetables, tobacco, and cotton, and also perennial woody crops such as orchards and vineyards. Crop vegetation accounts for greater than 20% of total vegetation. This class also includes all land being actively tilled (https://www.mrlc.gov/data/legends/national-land-cover-database-class-legend-and-description). |
| HYDE | FAO categories of "arable land and permanent crops" (Klein Goldewijk et al., 2017). |
| This study | Same as USDA-ERS, but we only count the cropland harvested area. |

**Table R2:** Definition of grazing land, pasture, and rangeland in different data sources.

| Data source | Definition |
|---|---|
| USDA-ERS | *Cropland pasture*: Cropland pasture includes acres of crops hogged or grazed but not harvested and some land used for pasture that could have been cropped without additional improvement.
*Grassland pasture and range*: Grassland pasture and range encompass all open land used primarily for pasture and grazing, including shrub and brush-land types of pasture, grazing land with sagebrush and scattered mesquite, and all tame and native grasses, legumes, and other forage used for pasture or grazing—regardless of ownership.
*Forest land grazed*: Forested pasture and range consisting mainly of forest, brush-grown pasture, arid woodlands, and other areas within forested areas that have grass or other forage growth. |
| USDA-NRI | *Pasture*: A land cover/use category of land managed primarily for the production of introduced forage plants for livestock grazing. Pastureland cover may consist of a single species in a pure stand, a grass mixture, or a grass-legume mixture. Management usually consists of cultural treatments: fertilization, weed control, reseeding, renovation, and control of grazing. For the NRI, includes land that has a vegetative cover of grasses, legumes, and/or forbs, regardless of whether or not it is being grazed by livestock (USDA, 2020).
*Rangeland*: A broad land cover/use category on which the climax or potential plant cover is composed principally of native grasses, grass-like plants, forbs or shrubs suitable for grazing and browsing, and introduced forage species that are managed like rangeland. This would include areas where introduced hardy and persistent grasses, such as crested wheatgrass, are planted and such practices as deferred grazing, burning, chaining, and rotational grazing are used, with little or no chemicals or fertilizer being applied. Grasslands, savannas, many wetlands, some deserts, and tundra are considered to be rangeland. Certain communities of low forbs and shrubs, such as mesquite, chaparral, mountain shrub, and pinyon-juniper, are also included as rangeland (USDA, 2020). |
| EPA | *Pastures*: Pastures are those lands that are primarily used for the production of adapted, domesticated forage plants for livestock.
*Rangelands*: Rangelands are those lands on which the native vegetation (climax or natural potential plant community) is predominantly grasses, grass-like plants, forbs, or shrubs suitable for grazing or browsing use. Rangelands include natural grassland, savannas, many wetlands, some deserts, tundra, and certain forb and shrub communities. |

| | |
|---|---|
| NLCD | ***Pasture/Hay***: Areas of grasses, legumes, or grass-legume mixtures planted for livestock grazing or the production of seed or hay crops, typically on a perennial cycle. Pasture/hay vegetation accounts for greater than 20% of total vegetation. |
| HYDE | ***Grazing land***:Land used for mowing or grazing livestock, based on the FAO category "permanent meadows and pastures". Grazing land can be a variety of ecosystems, ranging from managed irrigated grasslands to unmanaged open savannah woodlands to semi-shrub/scrub, almost desert, lands.
 ***Pasture***: Pasture is high-intensity grazing land, or low intensity grazing lands where a conversion of the natural vegetation has occurred.
 ***Rangeland***: rangeland is low-intensity grazing land where the natural vegetation has not been converted. |
| This study | Same as NRI. |

**Table R3.** Forest definitions from different data sources.

| Data source | Definitions |
|---|---|
| Forest Inventory Analysis (FIA) | Land at least 10 percent stocked by forest trees of any size, or formerly having such tree cover, and not currently developed for non-forest uses, with a minimum area classification of 1 acre (https://www.fia.fs.fed.us/tools-data/maps/2007/descr/yfor_land.php). |
| National Land Cover Database (NLCD) | Areas dominated by trees generally greater than 5 meters tall, and greater than 20% of total vegetation cover (https://www.mrlc.gov/data/legends/national-land-cover-database-class-legend-and-description). |
| Land Use Harmonization 2 (LUH2) | Forest was defined using a single tree canopy cover threshold to match the global forest extent provided by the FAO FRA report (Hurtt et al., 2020). |
| This study | Same as Forest Inventory Analysis (FIA). |

For the spatial allocation, we agree with you that the actual allocation is fully dependent on the probability surfaces and the landscape pattern at the end of the day. CLUE model and FORE-SCE model generate a LULC map at the predicted year by allocating the LULC demand (LULC area net change) to a LULC base map. This method works well for short-period studies because they can assume that the large-scale LULC pattern is not changed. To be honest, we also tried such a spatial allocation method for generating fractional and Boolean-type data. However, the contemporary LULC pattern is not representative for the historical LULC pattern even going back to the 1940s in CONUS (Sohl et al., 2016). Thus, we have to add some modifiers (such as population density) to improve the LULC probability. We know it is simple by adding a population weight, but it is effective. Because the distribution of human-related LULC types (e.g., urban, crop, and pasture) was always correlated with population density in the early period. In Figure R3, we can see that the county-level population and cropland proportion show the same spatial patterns in 1850.

[Figure]

**Figure R3:** County-level population density and cropland proportion in 1850.

We add a detailed description for fractional type data and Boolean type data allocation methods and a workflow (Figure R4). To generate the fractional grid data, we assumed that the fraction of each LULC type at the grid level was determined by the total probability, which means that a grid cell (LULC class $k$) with a high probability will have a high fraction. Based on this principle and the state-level LULC area, we generated the fractional land use data at 1 km x 1 km resolution and annual time scale. The detailed information for generating fractional LULC data is shown in Figure R4 and the following steps: (1) prepare the input data: reconstructed LULC area and probability; (2) calculate the state target LULC fraction for class $k$ and initialize an empty LULC fraction surface; (3) calculate a temporal fraction surface; (4) modify the temporal fraction surface, we assume that the fraction of water and barren is stable, and the sum of urban, crop, pasture, and forest fraction is lower than the maximum fraction in each grid cell; (4) add the temporal fraction data to the empty LULC fraction; (5) judge whether the unallocated LULC area is smaller than 0.01 km$^2$, if yes, the irritation will be stopped and begin to allocate another LULC class, else the unallocated area will assigned to target fraction and return back to step 3, the allocation was processed iteratively until unallocated area was less than the threshold (0.01 km$^2$) in all states.

Based on the fractional type LULC data, we further generated the Boolean type LULC data at 1 km x 1 km resolution. The detailed information for generating fractional LULC data is shown in Figure R4 and as following steps: (1) prepare the input data: reconstructed LULC area and fractional data; (2) generate a temporal LULC map (HistB) through identifying the dominate LULC type in each grid cell and initialize an

empty LULC map (HisB_E); (3) calculate the area difference for class *k* between the HistB map and target area; (4) if the area difference is negative, we first sort the LULC fraction map where HisB map equal to *k*, the top m (equals to target area) grid cells where HisB_E not be assigned a value will be assigned as *k*; then if the available number of grid cell (class *k*) is smaller than the target area, we will sort the LULC fraction map where HisB map not equal to *k*, and the top n (equals to unallocated area) grid cells where HisB_E not be assigned a value will be assigned as *k*; (5) if the area difference is positive, the grid cells where HisB map equal to *k* and will be assigned *k* to HisB_E not be assigned a value; then we will sort the LULC fraction map where HisB map not equal to *k*, and the top n (equals to unallocated area) grid cells where HisB_E not be assigned a value will be assigned as *k*. If step (4) or step 5 is finished, the next LULC type will begin to allocate. After the four types allocations are finished, the grid cell not being assigned a value will be updated using the HistB map and LANDFIRE Biophysical Settings data.

[Figure]

**Figure R4:** Workflow for generating fractional and Boolean type LULC data

(4) For the Boolean type data, you proposed several suggestions as follows:

- *Cropland and pasture become concentrated in the high-probability locations, with less fragmentation that's there in later dates.*

- *Going back in time reveals a number of obvious state and even what appear to be county boundaries, hard obvious lines where land use clearly differs on either side of a political boundary.*

- *Given the complete reliance on probability surfaces alone for the spatial allocation of change, land use looks more concentrated on the landscape for some classes going back in time.*

We re-check the Boolean type LULC data and analyze the reasons for these problems. The 'political boundary' issue resulted from the following two aspects. The first is that we conduct the spatial allocation at the state level. If the area of one LULC type has a large difference between two neighboring states, there would be a 'political boundary'. The second reason is that we use county-level population density (one value in a county) to modify the LULC probability in the early period, which will result in the hardlines between neighborhood counties. The LULC '*concentration*' problems also resulted from that we apply the population density data to modify the LULC probability surface. In our revised version, we optimized the population density weight and Boolean type spatial allocation method. A detailed description of the method can be found in section 2.4.1 and 2.4.2. Figure R5 shows the comparison of LULC at the local scale in 1850.

[Figure]

**Figure R5:** Comparison of Boolean type LULC map in 1850 between before and after optimization.

**Specific comments:**

**Comment 1:** Lines 28 – Would add one word…" In particular, managing agriculture and forest-related activities…"

**Response**: Thank you for this suggestion. We revise those sentences.

**Comment 2:** Line 34 – Would add words…"…arrival of Europeans, indigenous communities practiced agriculture and crop planting in the…"

**Response**: Thank you for this suggestion. We revise those sentences.

**Comment 3:** Line 36 – Would change "mainly occurred" to "initially occurred" to indicate these activities first started here, but expanded elsewhere later (as noted by the next sentences)

**Response**: Thank you for this suggestion. We revise those sentences.

**Comment 4:** Line 37 – "Driven" not "driving".

**Response**: Thank you for this suggestion. We revise those sentences.

**Comment 5:** Page 2 – Overall the paragraph at the top of page 2 could use some work. It's a rather disjointed history of US land change. For one it doesn't really talk about land change west of the Mississippi River, it's focused solely on Eastern US change. The organization is also a bit odd and disjointed.

The sentence on line 44, for example, seems like a very abrupt and odd ending to the final statement as to why a long-term land use dataset is needed. Perhaps a better organization by period (colonial, 19th century, 20th century), with a description of what occurred in each century? And perhaps a modification of the last sentence, adding "While general trends in historical US landscape change are known, we still lack a long-term dataset…"

**Response**: Thank you for this suggestion. We rewrite this paragraph. Please see Lines 32-49.

In the past four centuries, the conterminous United States (CONUS) has experienced dramatic land use and land cover (LULC) changes associated with land clearing, cropland, and urban land expansion (Steyaert and Knox, 2008; Drummond and Loveland, 2010; Oswalt et al., 2014; Sohl et al., 2016). Before the arrival of Europeans, indigenous agriculture and crop planting existed in the eastern woodlands, the Great

Plains, and the southwestern US (Hurt, 2002). Since the establishment of the first colony in Virginia in 1607, farms began to expand by land clearing, which initially occurred in the eastern United States (Steyaert and Knox, 2008). During the colonial era, most people lived in the east of the Appalachian Mountains and agriculture was the primary livelihood. In the late 18th and 19th centuries, territorial expansion (e.g., Louisiana Purchase) open up new areas for agriculture. Driven by westward movement, land clearing, agricultural land reclamation, and deforestation expanded across the Appalachian Mountains into Ohio, the Mississippi River basin, the Great Lakes region, and the western US (Cole et al., 1998; Billington et al., 2001; Steyaert and Knox, 2008; Yu and Lu, 2018). Hardwood forests in the Mississippi River Valley were cleared for cotton and grain production (Hanberry et al., 2012). The lumber production center also shifted from the Northwest to Great Lakes states in the 1850s (Fickle et al., 2001). In California, agriculture and ranching expanded throughout the state and soon became an exporter of wheat as gold mining waned (Olmstead et al., 2017). Entered the 20th century, the agricultural intensification resulted in cropland abandonment in New England, the Atlantic coast, and the southeast US (Foster, 1992; Hall et al., 2002; Jeon et al., 2014; Zumkehr and Campbell, 2013). Meanwhile, the environmental protection movement accelerated forest restoration, and the national plantation forest area increased to 27 Mha in 2017 (Oswalt et al., 2014; Stanturf et al., 2014; Chen et al., 2017). While general trends in historical US landscape change are known, we still lack a long-term spatial-explicit LULC dataset to characterize LULC trajectories for the CONUS.

**Comment 6:** Line 55-56 – Be careful about highlighting "uncertainties" in datasets such as HYDE, as your historical landscape construction will also have substantial uncertainties. Your workflow itself uses HYDE data. There's limited spatially explicit data available from which to base a model-based landscape reconstruction, and many of the datasets you're using were also used by HYDE.

**Response**: Thank you for this suggestion. We agree with you that the uncertainties of HYDE should not be highlighted because HYDE data is also one of the input data in our reconstruction. The historical LULC pattern in the CONUS has changed a lot compared with the contemporary pattern. This is also the reason that we used the

population density and human settlement extent to improve the LULC probability, even though it is a simple method.

**Comment 7:** Lines 87-88 – I wouldn't call the use of these other datasets "validation". It's a consistency check, not a validation, as these data sets too have uncertainties, and some are modeled just as you're modeling.

**Response**: Thank you for this suggestion. These data should be the input for consistency check rather than validation. We revise the description about the data comparison or consistency check. Please see section 2.4.

**Comment 8:** Line 90-91 – How was resampling done to get to 1-km grid cells? Is it fractional LULC within a given 1-km cell for datasets with native resolution <1 km?

**Response**: Thank you for this suggestion. The input data were resampled (nearest method) or aggregated to 1 km resolution. If the dataset with native resolution < 1 km, it will be fractional LULC within a given 1-km cell.

For the NLCD dataset, we generated the fractional type data to 1 km for each LULC type (urban, cropland, pasture, and forest) using the "aggregate" method.

**Comment 9:** Section 2.2.1 – This is an extremely simplistic methodology for calculating urban land area. To start, it's all based on one current dataset, NLCD. How was NLCD used? First of all, NLCD tends to underestimate low-density residential lands, which can bias your results. Secondly, NLCD "urban" classes also include extensive representation of road networks, which if counted as "urban", greatly overestimates urban land. For a rural state, for example, NLCD classes not only major roads, but every small section road has a 1-pixel-wide "urban" class representing it. Unless measures were taken to account for NLCD's underrepresentation of low-density residential lands, and to account for all the "urban" pixels that are really roads, it biases the results. The other problem is the very simplistic method for calculating land area. You're assuming the relationship between urban land per capita and total population is constant through time. Clearly it's not. Without accounting for that changing relationship, urban estimates can easily be biased.

**Response**: Thank you for this suggestion. In this study, we count the four components (Developed, Open Space; Developed, Low Density; Developed, Medium Density;

Developed, High Density) of NLCD developed land as urban land. And the developed land area between 2001 and 2019 is regarded as baseline data for historical urban land area reconstruction. Though NLCD has some shortcomings, we need to choose a dataset with a clear definition and spatial explicit map, which will be helpful for historical urban land reconstruction.

In the current method, we used the population and a stable urban land per capita to estimate the long-term urban area at the state level. We agree that it is a very simplistic method and also pointed out that it overestimates the total urban land area in the early period. To solve this problem, we apply the HISDAC data to reduce the bias between 1810 and 2000, and the method has been updated in *section 2.2.1*. Before 1810, there is no available data that can be used, and we assumed that urban land has the same change rate as the total population at the state level.

**Section 2.1.1**

In this study, we use the same definition as developed land in NLCD for urban land. Developed land in NLCD includes four components: open space, low intensity developed land; medium intensity developed land, and high intensity developed land. For 2001–2019, the NLCD developed land area was regarded as the urban land baseline. Before 2001, we applied the historical built-up area (HBUA) data from Historical Settlement Data Compilation for the United States (HISDAC-US) (Leyk et al., 2020; Uhl et al., 2021) as input to reconstruct the historical urban land area. The HISDAC-US describes the built environment of most of the CONUS from 1810 to 2010 at 5-year temporal and 250 m spatial resolution by using built-up property records, built-up property locations, and the built-up intensity (Leyk and Uhl, 2018; Uhl et al., 2021). Here, we assumed that the HISDAC data can capture the changing trend of urban land development. Then, the historical urban land can be estimated as follows:

$$HistUrban_{s,t} = HistUrban_{s,t+1} \times \frac{HISDAC_{s,t}}{HISDAC_{s,t+1}} \qquad (1)$$

where $HistUrban_{s,t}$ and $HistUrban_{s,t+1}$ are the reconstructed urban land area of state *s* in year t and t+1; $HISDAC_{s,t}$ and $HISDAC_{s,t+1}$ are the HISDAC built-up area of state *s* at year *t* and t+1. The step was first conducted in 2000 and went back to 1810.

There is no reliable data on urban development before 1810. Following Liu et al. (2010), we used population to estimate the urban land area by assuming that urban

land expanded at the same rate as total population during 1630–1810. The urban land area of each state can be calculated as follows:

$$HistUrban_{s,t} = HistUrban_{s,t+1} \times \frac{Pop_{s,t}}{Pop_{s,t+1}} \tag{2}$$

where $HistUrban_{s,t}$ and $HistUrban_{s,t+1}$ are the reconstructed urban land area of state $s$ in year t and t+1; $Pop_{s,t}$ and $Pop_{s,t+1}$ are the total population of state $s$ at year $t$ and t+1.

**Comment 10:** Section 2.2.2 – You're using (at least) three different data sources to help establish cropland area. For historical land use, estimates vary widely, dependent upon methodology, data source, thematic definitions of a land use, etc. As a result, when switching from USDA-based data, for example, after 1889, and using HYDE before 1889, you'd expect an obvious break in estimated "cropland" amounts. How were those inconsistencies among historical land use datasets harmonized?

**Response**: Thank you for this suggestion. In the current method, we first convert cropland harvest area to planted area by assuming a stable linear relationship between these two datasets, then we calculated the cropland area by subtracting the double-cropped area from the planted area. However, these two steps will result in some uncertainties as you said in Comment 11 and 12.

To reduce these uncertainties, we decided to change the input data for historical cropland reconstruction, including the ERS national cropland harvested area (without double-cropped area), the state level CAHA-cropland harvested area (1879-2017) and HYDE cropland (1600-1879) to reconstruct the historical cropland area. Please see the new method described in section 2.2.2.

The definition of cropland varies in the existing literature and datasets (Bigelow and Borchers, 2017; Yu and Lu, 2018; Homer et al., 2020). Cropland defined by the U.S. Department of Agriculture (USDA) Economic Research Service (ERS) includes five components: cropland harvested, crop failure, cultivated summer fallow, cropland pasture, and idle cropland (Table 2). In this study, we only counted the cropland harvested area, which includes row crops and closely sown crops, hay and silage crops, tree fruits, small fruits, berries, and tree nuts, vegetables and melons, and miscellaneous other minor crops (https://www.ers.usda.gov/data-products/major-land-uses/glossary/#cropland). The USDA Census of Agriculture Historical Archive

(CAHA) provides state-level cropland harvested area with 4 or 5 years interval, which was applied for historical cropland reconstruction between 1879-2020. The dataset was first interpolated linearly to annual. To subtract the double-cropped area in the dataset, we applied the national cropland harvested area from ERS major land uses data to adjust the interpolated data during 1910-2020. The adjustment can be expressed as follows:

$$HistCrop_{s,t} = \frac{Cropland\ Harvested_{s,t}^{linear}}{Cropland\ Harvested_{conus,t}^{linear}} \times Cropland\ Harvested_{conus,t}^{ERS} \qquad (3)$$

where $HistCrop_{s,t}$ is the reconstructed historical cropland area of state $s$ in year $t$; $Cropland\ Harvested_{s,t}^{linear}$ is the linear interpolated cropland harvested area of state $s$ in year $t$ based on CAHA data; $Cropland\ Harvested_{conus,t}^{ERS}$ is the national total cropland harvested area without double-cropped area. For 2018–2020, the state level cropland area was calculated based on the state level area weight in 2017.

For 1879–1910, there is no national level statistics for cropland harvested area without double-cropped area. We applied the changing trend the CAHA cropland harvested area to reconstruct the historical cropland in this period, which can be calculated as follows:

$$HistCrop_{s,t} = HistCrop_{s,t+1} \times \frac{CAHA\_CHA_{s,t}}{CAHA\_CHA_{s,t+1}} \qquad (4)$$

where $HistCrop_{s,t}$ and $HistCrop_{s,t+1}$ are the reconstructed historical cropland area of state $s$ in year $t$ and t+1; $CAHA\_CHA_{s,t}$ and $CAHA\_CHA_{s,t+1}$ are improved farmland area of state $s$ in year t and t+1.

Before 1879, there is no available census data, we used the HYDE cropland dataset to reconstruct historical cropland area. The harmonization method can be expressed as follows:

$$HistCrop_{s,t} = HistCrop_{s,t+1} \times \frac{HYDE\_Crop_{s,t}}{HYDE\_Crop_{s,t+1}} \qquad (5)$$

where $HistCrop_{s,t}$ and $HistCrop_{s,t+1}$ are the reconstructed historical cropland area of state $s$ in year t and t+1; $HYDE\_Crop_{s,t}$ and $HYDE\_Crop_{s,t+1}$ are HYDE cropland area of state $s$ in year t and t+1.

**Comment 11:** Paragraph starting on line 120 – You're assuming the relationship between harvested area and planted area from 1978 to 2017 is consistent decades and centuries before those data…a very dangerous assumption.

**Response**: Thank you for this suggestion. We agree that the relationship between cropland harvested area and the planted area is changed. Thus, we revise the

reconstruction method. This step doesn't need to conduct in the new method. Please see the new method description in the response for Comment 10.

**Comment 12:** Line 123 – Yet another dataset, Borchers et al. 2014, was used to establish double-cropping at a regional level. Again, consistency among most of these datasets isn't great.

**Response**: Thank you for this suggestion. To reduce such kind of uncertainties, we revise the reconstruction method, and the subtracting work doesn't need to conduct. Please see the new method described in the response to Comment 10.

**Comment 13:** Line 124-125 – Another basic assumption that likely isn't true through time.

**Response**: Thank you for this suggestion. We revise the cropland reconstruction method, and we used the changing trend of HYDE cropland rather than national cropland per capita. Please see the new method described in the response to Comment 10.

**Comment 14:** Line 125-126 – Because the 1879 number was different you assumed it was incorrect? But data >1889 were "correct"? Was the reconstructed cropland area in 1879 substantially lower or higher than 1889?

**Response**: Thank you for this suggestion. On the one hand, there was no record for South Dakota in 1879. On the other hand, we assumed that there should not be a large difference between the cropland harvest area between 1879 and 1889. However, the cropland harvested area in South Dakota, Nebraska, and Kansas did have large differences in 1879 and 1889 (Figure R6). Thus, we thought the cropland harvested area in 1879 is not correct before.

We rechecked the changes in cropland harvested area and total population (Figure R7) between 1879 and 1889, we think the cropland harvested area recorded by CAHA should be right and can be used for cropland reconstruction. So, we used the CAHA data in the revised version.

The reconstructed cropland area in 1879 was not substantially lower or higher than 1889 (Figure R8).

[Figure]

**Figure R6**: Comparison between cropland harvested area in 1889 and 1879.

[Figure]

**Figure R7**: Changes of total population in Nebraska, Kansas, and South Dakota between 1850 and 1890.

[Figure]

**Figure R8**: Changes of national total cropland area during 1850-1910.

**Comment 15:** Line 130-131 – Another basic assumption that likely doesn't hold region to region.

**Response**: Thank you for this suggestion. We change the harmonization method for integrating HYDE and reconstructed historical cropland. Please see the new method described in the response to Comment 10.

**Comment 16:** Line 132-133 – Again…how did you account for differences in the HYDE data, and the (mostly) USDA-based data after 1889? Is there an obvious break in cropland amount pre- and post-1889?

**Response**: Thank you for this suggestion. For the historical cropland reconstruction, we use the HYDE cropland area trends rather than absolute value. The harmonization method between HYDE and CAHA or reconstructed cropland area can be found in the response to Comment 10.

As our response in Comment 14, we found that cropland harvested area increased fast during 1879-1889, and this increase resulted from rapid population increase. The reconstructed data also show that cropland area in South Dakota, Nebraska, Kansa increased rapidly during 1879-1889. But there is no obvious break in cropland amount pre and post-1889 (Figure R8).

**Comment 17:** Line 135-136 – One of the greatest difficulties in historical landscape reconstruction is the definition of "pasture", vs. "grassland", vs. "hay" vs. "rangeland", etc. There is no one definition that's universally accepted. Your definition here states Pasture includes areas "for the production of seed or hay crops". Many definitions of "cropland" include alfalfa, hay, and other crops in "planted area" or "cultivated crop" area. For "Pasture", you're introducing yet another completely new dataset to establish pasture area, NRI. Are the definitions of "pasture" for NRI the same for NLCD, HYDE, and the US Census of Ag?

**Response**: Thank you for this suggestion. We summarize the definitions of grazing land, pasture, and rangeland from different sources (Table R2).

As you said, the definitions 'pasture' from NRI is not same as that in HYDE and NLCD or US Census of Ag. Therefore, it is difficult to reconstruct historical pasture area by harmonizing these datasets. In this study, we used the pasture definition from

NRI, and the state level pasture area between 1982 and 2017 was set as baseline data. Before 1982, we applied the changing trend of HYDE pasture data to reconstruct the historical pasture area. The following harmonization method was conducted to calculate historical pasture land area:

$$HistPasture_{s,t} = HistPasture_{s,t+1} \times \frac{HYDE\_Pasture_{s,t}}{HYDE\_Pasture_{s,t+1}} \tag{6}$$

where $HistPasture_{s,t}$ and $HistPasture_{s,t+1}$ are the reconstructed historical pasture areas of state $s$ in year t and t+1; $HYDE\_Pasture_{s,t}$ and $HYDE\_Pasture_{s,t+1}$ are the HYDE pasture area of state $s$ in year t and t+1.

**Comment 18:** Line 139 – Note Wasianen and Bliss took great pains to harmonize those definitional differences across their harmonized dataset.

**Response**: Thank you for this suggestion. We agree that they did make great efforts to harmonize those definitional differences across multisource dataset.

**Comment 19:** Section 2.2.3 – Again…it's extremely simplistic to assume things such as "pasture per capita" and that that ratio is consistent over time, and space.

**Response**: Thank you for this suggestion. We revise the historical pasture reconstruction method.

In this study, we used the pasture definition from NRI, and the state level pasture area between 1982 and 2017 was set as baseline data. Before 1982, we applied the changing trend of HYDE pasture to reconstruct the historical pasture area. The following harmonization method was conducted to calculate historical pasture area:

$$HistPasture_{s,t} = HistPasture_{s,t+1} \times \frac{HYDE\_Pasture_{s,t}}{HYDE\_Pasture_{s,t+1}} \tag{6}$$

where $HistPasture_{s,t}$ and $HistPasture_{s,t+1}$ are the reconstructed historical pasture areas of state $s$ in year t and t+1; $HYDE\_Pasture_{s,t}$ and $HYDE\_Pasture_{s,t+1}$ are the HYDE pasture area of state $s$ in year t and t+1.

**Comment 20:** Section 2.2.4 – Definitions of what is "Forest" vary greatly among data sets. You're introducing yet another data set in FIA that may have a definition of "forest" that differs from HYDE or from NLCD. How closely does the FATD data match with HYDE estimates, for example?

**Response**: Thank you for this suggestion. The definitional differences between FIA, NLCD, and LUH2 can be found in Table R3.

The HYDE data doesn't provide forest area estimation. So, the following figure shows the comparison between USDA-FR and NLCD, LUH2 between 2000-2020, between LUH2 and FATD in 1630. Both NLCD and LUH2 forest area are lower than USDA-FR. The forest area in Rocky Mountain states such as Nevada, Utah, New Mexico from NLCD and LUH2 is lower than that from USDA-FR. The forest area in 1630 derived from LUH2 and FATD does not match well.

[Figure]

**Figure R9**: Comparison between the average forest area (2000-2020) derived from USDA-FR and NLCD, LUH2.

**Comment 21:** Lines 156-157 – Yeah you lost me here with what you're trying to do, needs a better explanation.

**Response**: Thank you for the suggestions. We rewrite the method to harmonize the FATD and USDA-FR forest data.

**Comment 22:** Section 2.2.5 – See main comments above related to how you balanced the four LULC classes.

**Response**: Thank you for this suggestion.

**Comment 23:** Line 195 – What was used to establish the "land use change boundary"? That is, what was the source of "settled area" data"?

**Response**: Thank you for this suggestion. We assumed that in the area where human was not settled, there was no urban land, cropland, and pasture. The human settlement

boundary data were used to constrain the probability of urban, cropland, and pasture. The Exploration and Settlement maps were made by the U.S. Dept. of the Interior, Geological Survey and can be accessed at https://maps.lib.utexas.edu/maps/histus.html#exploration.html. We assumed that the LULC would not be changed as the pre-colonial era, though there were Native Indiana people.

**Comment 24:** I don't mind the use of something like this to constrain the allocation of change, but do wonder about full-resolution results. Are there any hard border issues obvious in the data when change occurs at the edge of those defined boundary layers? Overall with the boundary and effect of population density, I appreciate you trying something other than assuming a static probability surface through time.

**Response**: Thank you for this suggestion. We check the fractional and Boolean type gridded data, there are hard border issues in some years which resulted from the county-level population application to modify the LULC probability surfaces. We use a simple method to generate the historical gridded population by combing the gridded population data with 1-km resolution and county-level population, which can be expressed as:

$$Pop_{i,t} = \frac{Pop_{i,t}^{county}}{Pop_{i,2000}^{county}} \times Pop_{i,2000}^{grid}$$

where, $Pop_{i,t}$ is improved population density at grid cell $i$ and year $t$; $Pop_{i,2000}^{county}$ and $Pop_{i,t}^{county}$ is the county-level population density at grid cell $i$ in 2000 and year $t$; $Pop_{i,2000}^{grid}$ is the gridded population density at grid cell $i$ in 2000 and year $t$.

[Figure]

**Figure R10**: Comparison between county-level and gridded population density data in 1850.

**Comment 25:** Section 2.3.2 – There needs to be more explanation here. You've basically summarized the entire actual allocation to the pixel level in one sentence. I certainly get that higher probability areas will likely have a higher proportion of a given LULC class, but it's all deterministic and it's all based solely on the probability surface? There's no stochasticity? With such a sparse description of methodology, it's also hard to see how this simple description of the methodology ends up with the aggregate totals from the allocation stage matching the quantitative estimates you established for each of the LULC classes.

**Response**: Thank you for this suggestion. We rewrite the description of spatial allocation strategies for generating fractional and Boolean type LULC data and make it as detailed as possible (section 2.3.2). We also add a random item when calculating the total transition probabilities (section 2.3.1). Please see the spatial allocation method described in the response to the general comments.

**Comment 26:** As noted in the main comments, I have other concerns about the allocation strategy.

**Response**: Thank you for this suggestion. We rewrite the spatial allocation strategy and make it as detailed as possible. Please see the spatial allocation method described in the response to the general comments.

**Comment 27:** Section 2.4 – Comparison to other LULC datasets isn't a validation, it's a consistency check. That's particularly true when every dataset has it's own production methodologies, data sources, and thematic definitions, all of which makes even direct comparison problematic. Beyond that, you're comparing your results to some of the same datasets from which you parameterized your modeling, as noted in the overarching comments. Also note there aren't any details as to what methodologies you're actually using for "validation" in this very short, one-paragraph section.

**Response**: Thank you for this suggestion. We agree that the description or this step is a consistency check, rather than a data validation. Complete formal validation of model results was impractical because true, spatially explicit 'reference' data for historical LULC are difficult to obtain. In the historical LULC area reconstruction step, we assumed that the data used is reliable, which made it hard to conduct the

validation. If we apply a rule to reconstruct the historical LULC area, we can use the census data to validate the accuracy of the prediction. For example, Sohl et al. (2016) used the LULC change data from the Trends project to reconstruct historical LULC proportions (demand) between 1938 and 1992, and compared the model results with census data, but such comparison still was a consistency check. Moreover, the definitional differences make it difficult to compare the newly developed dataset with other LULC products. We keep the LULC area comparison at the state level. Two new comparisons were conducted: Comparison between the newly developed cropland and USDA historical cropland area at the county level; Comparison between our reconstruction and NLCD developed land, cropland, pasture, and forest. Please see section 3.1.

**Comment 28:** Section 3.1 – As noted previously, this isn't very useful for inferring confidence in your results, when you're using the same datasets to parameterize the model as you are to "validate" model results.

**Response**: Thank you for this suggestion. As our response in Comment 27, it is hard to validate the newly developed LULC data. We keep the LULC area comparison at the state level. Two new comparisons were conducted: Comparison between the newly developed cropland and USDA historical cropland area at the county level; Comparison between our reconstruction and NLCD developed land, cropland, pasture, and forest. Please see section 3.1.

**Comment 29:** Line 273 (and throughout the results section) – If you're going to refer to a specific driving force of change, and, for example, point to a specific policy, you should name the policy and reference it (Immigration and Naturalization Act of 1965). While it certainly did change the nature of immigration to the country, you do give it too much focus as "the" causes of urban land increases after 1965. There's a lot more at play there than immigration policy.

**Response**: Thank you for this suggestion. We agree that urban land expansion is driven by multiple factors, while it is largely determined by population and economy growth. We rewrite the related sentences about urban land expansion.

**Comment 30:** Lines 276-277 – You state "cropland area did not change significantly" from 1930 to present day. First, it's always problematic to use the term

"significantly" in a journal paper, given the scientific meaning of the word. Secondly, I would argue there were "substantial" trends in agriculture after 1930, including some of those you mention (e.g., biofuel impacts).

**Response**: Thank you for this specific suggestion. We rewrite the related sentences and this word should be used carefully. What I want to express is that the change magnitude of the national total cropland area is not like the period of 1850-1920 (Figure R11). In fact, cropland was abandoned in the southeast US and expanded in the Great Plains.

[Figure]

**Figure R11**: Changes of national total cropland area derived from the newly developed LULC dataset during 1630-2020.

**Comment 31:** Figure 7 – On a national-scale map figure, it's difficult to see patterns of the individual land use transitions. Perhaps it would be augmented by a complementary confusion matrix of changes or some other tabular data approach that allows you to see (and easily quantify) transition types.

**Response**: Thank you for this suggestion. We add a LULC conversion table including the information of major LULC conversions during 1630–1850, 1850–1920, 1920–2020, and 1630–2020. Please see Table 3.

**Comment 32:** Section 3.4 – This isn't the most effective section to me. As noted in the main comments, a major premise of the paper was to provide a "high resolution" historical landscape reconstruction for the US. Much of the "regional" information here is also discussed in the overall results above. I'd have much rather seen some real examples (and preferably validation) of landscape pattern at finer scales, given the focus on higher resolution with this paper.

**Response**: Thank you for this suggestion. Considering the differences in natural environmental conditions and social-economic development, land use and land cover change showed spatial heterogeneity in the CONUS during 1630–2020. The purpose of section 3.4 is to give a general description of how LULC changes among regions.

To show the improvement of newly developed LULC data, we add several figures to show LULC changes at a finer scale in the discussion section, please see Figure R12, 13, and 14.

[Figure]

**Figure R12: Visual comparison between our cropland and HYDE3.2, YLmap, and ZCmap in four different sites (a-d). The locations of image center points are as follows: a. Ohio in 1850 (83.05 °W, 40.17 °N), b. Georgia in 1920 (83.58 °W, 32.77 °N), c. Arkansas in 1920 (90.56 °W, 34.76 °N), d. Texas in 1920 (100.92 °W, 32.81°N).**

[Figure]

**Figure R13: Visual comparison our pasture and HYDE3.2, LUH2 in four different sites (a-d). The locations of image center points are as follows: a. Iowa in 1920 (93.64 °W, 42.03 °N), Virginia in 1920 (78.72 °W, 37.96 °N), c. Illinois in 2000 (90.07 °W, 38.68 °N), d. Arkansas in 2000 (92.56 °W, 34.97 °N).**

[Figure]

**Figure R14: Visual comparison between our forest and LUH2 in four different sites (a-d). The locations of image center points are as follows: a. Colorado in 1630 (106.47 °W, 38.97 °N), Wisconsin in 1850 (89.85 °W, 44.54 °N), c. Alabama in 1920 (86.72 °W, 33.33 °N), d. New York in 2010 (75.14 °W, 42.21 °N).**

**Comment 33:** Lines 338-339 – Agreed about the "per capita" approach.

**Response**: Thank you for this suggestion. We improve the urban land estimation method by using a changing urban land per capita. The HISDAC data is applied between 1810 and 2001 to reduce the bias in our estimation. The new method description can be found in the response to Comment 9.

**Comment 34:** Lines 339-341 – This doesn't serve as any kind of adequate validation or even consistency check between datasets. Showing a national-scale map and stating the patterns are "consistent" isn't valuable, and is very subjective at that scale.

**Response**: Thank you for this suggestion. The data validation/comparison or consistency check has been conducted in section 3.1 by comparing with NLCD data, agriculture census data, and state-level LULC area.

The national scale maps can give an overview of the spatial pattern of LULC in 1630, 1850, 1920, and 2010. We keep the national-scale map and add extra comparison figures at the regional scale to show our improvement. Please see Figure R12, R13, and R14.

**Comment 35:** Line 357-358 – Exactly why it's not very valuable to compare your model results to HYDE…those data were used to help establish the model parameters themselves.

**Response**: Thank you for this suggestion. The HYDE data was used to reconstruct historical cropland and pasture area. But not all the periods applied the HYDE data. Moreover, we used the trend or interannual variations of HYDE data rather than the absolute value of LULC area. Thus, we compared the reconstructed historical LULC area and spatial pattern with HYDE.

**Comment 36:** Line 362 – Your product has higher spatial resolution than something like HYDE, but there's no quantitative analysis of that spatial pattern that proves the superior value of that higher native resolution.

**Response**: Thank you for this suggestion. We add regional scale figures to show the data improvement than the dataset with coarse resolution (Figure R12, R13, and R14). Moreover, the HYDE or LUH2 have higher cropland acreage compared to US-specific datasets, like Yu and Lu, and USDA census data. We fixed this problem and went back to 400 years ago.

**Comment 37:**  Figure 9 – It is difficult to compare all of these datasets given the definitional differences between them, particularly for pasture and cropland.

**Response**: Thank you for this suggestion. We agree that it is really hard to say which data is more accurate or reliable because of the definitional differences among them. But we can know whether the area of the reconstructed historical LULC dataset is in a reasonable range through data comparison. Meanwhile, the previous spatial LULC datasets are also a good reference to judge whether the reconstructed data has a reasonable spatial pattern.

**Comment 38:**  Lines 382-383 – I'm not sure it's more "reliable", as sample-based, inventory approaches have flaws, just as satellite-based approaches have flaws. The bigger concern to me are the definitional differences, not the methodological differences.

**Response**: Thank you for the suggestions. We rewrite the sentence. The word 'reliable' may not suitable to describe the FIA data, I would say it has better consistency than other historical forest data for long-term study. But it is hard to say which forest data is more reliable because of the differences between forest definitions.

NLCD and Sohl et al. (2016) data define forest as the areas dominated by trees generally greater than 5 meters tall and greater than 20% of total vegetation cover, higher than that in our forest definition (forest cover greater than 10%). Thus, the forest area in this study was higher than the NLCD and Sohl et al. (2016) data. In LUH2, the biomass density (BD) map is used to identify the potential forest (BD > 2 kg C m$^{-2}$) and non-forest at $0.5 \times 0.5$-degree resolution, which underestimates the forest in Rock Mountain and Northwest. NLCD is produced by using Landsat images and a comprehensive method and provides nationwide data on land cover and land cover change at a 30 m resolution (Homer et al., 2020). Spatially, it can capture the forest distribution better than LUH2. The FIA data provides critical status and trend information through a system of annual resource inventory that covers both public and private forest lands across the United States (https://www.fs.usda.gov/research/inventory/FIA), and it can provide forest trend data back to 1630.

**Comment 39:** Section 4.2 doesn't add a lot to the paper for me, particularly since you've already tried to explain some driving forces in the previous paragraphs of the paper. I'd much rather have the drivers woven into the story of what's happening in your results, than as a separate section.

**Response**: Thank you for this suggestion. The driving forces of land use and land cover change are quite complex in the United States. Though some driving forces of LULC change have been mentioned in the *Results* part, we think a comprehensive analysis of the driving forces of LULC change is still needed. We reorganized this paragraph and add more discussions. Please see Section 4.2 in the revised manuscript.

**Comment 40:** Lines 435-436 – I think reconstruction of historical land use is limited more by reliable, consistent historical data than methodology. Machine learning methods aren't going to be that valuable for historical reconstruction given the paucity and inconsistency of historical data for training.

**Response**: Thank you for this suggestion. We agree that the most important step in the historical LULC reconstruction study is to collect reliable and consistent data. The spatial allocation algorithm also impacts the reconstructed landscape pattern.

**Comment 41:** Section 4.3 – Somewhere in here you absolutely need to highlight the difficulties with trying to harmonize data sets with different definitions, data sources, and methodologies.

**Response**: Thank you for this suggestion. In this section, we add the discussion about the difficulties of harmonizing data sets with different definitions, data sources, and methodologies. Please see Section 4.3 in the revised manuscript.

Collecting reliable and consistent historical data is critical to reconstruct the historical LULC area. In this study, the LULC area is reconstructed at the state level, and LULC change at the sub-state scale is uncertain. Most of the input datasets were recorded at 5–10 years intervals, which made some insignificant fluctuations cannot be captured. Another difficulty to reconstruct the historical LULC dataset is the way to harmonizing the input datasets. Though we tried to gather the most reliable LULC datasets, the definitions for the same LULC type vary among them. For example, we applied two datasets (CAHA cropland harvested area and HYDE cropland) to generate the historical cropland area for the study period, but the definitions of these

two datasets are different. Moreover, the definitions of four major LULC types do not belong to a universal classification system, making it difficult to process the total area and a post-processing step need to be conducted. The assumptions made in the reconstruction step take some uncertainties for the historical LULC area.

---

## Author Comment (AC2)

Thank you for the comments and suggestions. These comments were very helpful for revising and improving our paper. We have responded to the comments point by point.

**Reviewer #2:**

**General comment:**

The manuscript reconstructed land use and land cover history during 1630-2020 over the conterminous United States (CONUS). It gathered multiple sources of data including remote sensing-based land cover maps, national inventories and statistics data, meteorological fields, topographical data and others. The resulting dataset has an improved spatial resolution than other global datasets, potentially facilitating regional modeling work. However, the manuscript is not well organized. For example, in the method section, there are missing details about how various sources of input data were combined and adjusted to generate full time series of urban and crop land area. And also, the validation needs more justification on the choice of spatial scale and more discussion about what causes differences between the data from this work and other datasets. For example, this data shows a different trend in areas of urban and pasture in the past two decades compared to others. I have provided detailed comments below.

**Response**: Thank you for these suggestions. We rewrite the method about reconstructing historical urban and cropland area with more detail information, including the data used for difference period and harmonization method between different datasets. Please see the section 2.2.

*Urban land:*

In this study, we use the same definition as developed land in NLCD for urban land. Developed land in NLCD includes four components: open space, low intensity developed land; medium intensity developed land, and high intensity developed land. For 2001–2019, the NLCD developed land area was regarded as the urban land baseline. Before 2001, we applied the historical built-up area (HBUA) data from Historical Settlement Data Compilation for the United States (HISDAC-US) (Leyk et al., 2020; Uhl et al., 2021) as input to reconstruct the historical urban land area. The HISDAC-US describes the built environment of most of the CONUS from 1810 to 2010

at 5-year temporal and 250 m spatial resolution by using built-up property records, built-up property locations, and the built-up intensity (Leyk and Uhl, 2018; Uhl et al., 2021). Here, we assumed that the HISDAC data can capture the changing trend of urban land development. Then, the historical urban land can be estimated as follows:

$$HistUrban_{s,t} = HistUrban_{s,t+1} \times \frac{HISDAC_{s,t}}{HISDAC_{s,t+1}} \qquad (1)$$

where $HistUrban_{s,t}$ and $HistUrban_{s,t+1}$ are the reconstructed urban land area of state $s$ in year t and t+1; $HISDAC_{s,t}$ and $HISDAC_{s,t+1}$ are the HISDAC built-up area of state $s$ at year $t$ and t+1. The step was first conducted in 2000 and went back to 1810.

There is no reliable data on urban development before 1810. Following Liu et al. (2010), we used population to estimate the urban land area by assuming that urban land expanded at the same rate as total population during 1630–1810. The urban land area of each state can be calculated as follows:

$$HistUrban_{s,t} = HistUrban_{s,t+1} \times \frac{Pop_{s,t}}{Pop_{s,t+1}} \qquad (2)$$

where $HistUrban_{s,t}$ and $HistUrban_{s,t+1}$ are the reconstructed urban land area of state $s$ in year t and t+1; $Pop_{s,t}$ and $Pop_{s,t+1}$ are the total population of state $s$ at year $t$ and t+1.

*Cropland:*

The definition of cropland varies in the existing literature and datasets (Bigelow and Borchers, 2017; Yu and Lu, 2018; Homer et al., 2020). Cropland defined by the U.S. Department of Agriculture (USDA) Economic Research Service (ERS) includes five components: cropland harvested, crop failure, cultivated summer fallow, cropland pasture, and idle cropland (Table 2). In this study, we only counted the cropland harvested area, which includes row crops and closely sown crops, hay and silage crops, tree fruits, small fruits, berries, and tree nuts, vegetables and melons, and miscellaneous other minor crops (https://www.ers.usda.gov/data-products/major-land-uses/glossary/#cropland). The USDA Census of Agriculture Historical Archive (CAHA) provides state-level cropland harvested area with 4 or 5 years interval, which was applied for historical cropland reconstruction between 1879-2020. The dataset was first interpolated linearly to annual. To subtract the double-cropped area in the dataset,

we applied the national cropland harvested area from ERS major land uses data to adjust the interpolated data during 1910-2020. The adjustment can be expressed as follows:

$$HistCrop_{s,t} = \frac{Cropland\ Harvested_{s,t}^{linear}}{Cropland\ Harvested_{conus,t}^{linear}} \times Cropland\ Harvested_{conus,t}^{ERS} \qquad (3)$$

where $HistCrop_{s,t}$ is the reconstructed historical cropland area of state $s$ in year $t$; $Cropland\ Harvested_{s,t}^{linear}$ is the linear interpolated cropland harvested area of state $s$ in year $t$ based on CAHA data; $Cropland\ Harvested_{conus,t}^{ERS}$ is the national total cropland harvested area without double-cropped area. For 2018–2020, the state level cropland area was calculated based on the state level area weight in 2017.

For 1879–1910, there is no national level statistics for cropland harvested area without double-cropped area. We applied the changing trend the CAHA cropland harvested area to reconstruct the historical cropland in this period, which can be calculated as follows:

$$HistCrop_{s,t} = HistCrop_{s,t+1} \times \frac{CAHA\_CHA_{s,t}}{CAHA\_CHA_{s,t+1}} \qquad (4)$$

where $HistCrop_{s,t}$ and $HistCrop_{s,t+1}$ are the reconstructed historical cropland area of state $s$ in year $t$ and t+1; $CAHA\_CHA_{s,t}$ and $CAHA\_CHA_{s,t+1}$ are improved farmland area of state $s$ in year t and t+1.

Before 1879, there is no available census data, we used the HYDE cropland dataset to reconstruct historical cropland area. The harmonization method can be expressed as follows:

$$HistCrop_{s,t} = HistCrop_{s,t+1} \times \frac{HYDE\_Crop_{s,t}}{HYDE\_Crop_{s,t+1}} \qquad (5)$$

where $HistCrop_{s,t}$ and $HistCrop_{s,t+1}$ are the reconstructed historical cropland area of state $s$ in year t and t+1; $HYDE\_Crop_{s,t}$ and $HYDE\_Crop_{s,t+1}$ are HYDE cropland area of state $s$ in year t and t+1.

For the data validation/comparison, we add the causes analysis for the differences between the datasets (Please see section 3.1.1). Considering the importance of cropland and the census data availability, we add the county level cropland area comparison between our reconstruction and USDA census data (Please see section 3.1.2). Moreover, we also compare the NLCD and our data at grid level (Please see section 3.1.3).

**Specific comments**

**Comment 1:** P2, L55, LUH2 has a spatial resolution of 0.25-deg.

**Response**: Thank you for this suggestion. We revise it.

**Comment 2:** P2, l55, please provide references to the 'substantial uncertainties' statement.

**Response**: Thank you for this suggestion. We revise the word of "**substantial uncertainties**", and add references describing the spatial uncertainties of LULC data with coarse resolution.

Yu, Z. and Lu, C.: Historical cropland expansion and abandonment in the continental U.S. during 1850 to 2016, Glob. Ecol. Biogeogr., 27, 322-333, http://doi.org/10.1111/geb.12697, 2018.

Lin, S., Zheng, J., and He, F.: Gridding cropland data reconstruction over the agricultural region of China in 1820, J. Geogr. Sci., 19, 36–48, http://doi.org/10.1007/s11442-009-0036-x, 2009.

**Comment 3:** P5, L95, it would be clearer to split the datasets by the purposes in your model (e.g., input vs validation). And also, there is no temporal resolution for each dataset. Apparently, some of them are not annual.

**Response**: Thank you for this suggestion. We revise Table 1 and add the temporal resolution of each data set. Moreover, the data used for input and comparison/validation are also separate in Table 1.

**Comment 4:** P6, L100, please clarify if all land cover types of developed land (i.e., Developed, Open Space; Developed, Low Intensity; Developed, Medium Intensity; Developed High Intensity) were regarded as urban land.

**Response**: Thank you for this suggestion. We clarify the definition of urban land. In this study, the four components (i.e., open space, low intensity, medium intensity, high intensity developed land) in NLCD developed land is regarded as urban land.

**Comment 5:** P6, L100, did you take an average of urban land area during 2001-2016 and use it as the baseline? Or you did make use of the time series of urban land areas. I wonder about the temporal variation of urban land per capita and its impacts on

estimation of historical urban areas. It would be better to include this information in supplements.

**Response**: Thank you for this suggestion. We take an average of urban land area during 2001-2016 and use it as the baseline. In fact, the real urban land per capita is not stable especially for such long-term study period.

However, there is no census data for urban land area in the historical period. We calculated the built-up land area per capita during 1810–2010 using HISDAC data (Figure R1). The figure shows that the per capita increase gradually, indicating that our previous method overestimated the urban land area in the early period. Thus, we revise the urban land estimation method, please see the following descriptions.

[Figure]

**Figure R1**. Changes of historical built-up areas per capita during 1810–2010.

For 2001-2019, the NLCD developed land area was regarded as the baseline data. Before 2001, we applied built-up land data (Historical Settlement Data Compilation for the United States, HISDAC-US) from Leyk et al. (2020) as input to reconstruct the historical distribution of urban land. In HISDAC-US, the built environment of most of the CONUS from 1810 to 2010 was reconstructed using historical property records (Leyk and Uhl, 2018). We assume that HISDAC data can capture the interannual variations of urban land development. Thus, the historical urban land between 1810-2001 can be estimated as follows:

$$HistUrban_{s,t} = HistUrban_{s,t+1} \times \frac{HISDAC_{s,t}}{HISDAC_{s,t+1}} \tag{1}$$

where $HistUrban_{s,t}$ and $HistUrban_{s,t+1}$ are the reconstructed urban land area of state $s$ in year $t$ and $t+1$; $HISDAC_{s,t}$ and $HISDAC_{s,t+1}$ are the HISDAC built-up land area of state $s$ at year $t$ and $t+1$.

However, there is no reliable data on urban development before 1810. Following Liu et al. (2010), we use population to estimate the urban land expansion by assuming that urban areas expanded at the same rate as total population during 1630–1810. For this period, urban land area of each state can be calculated as follows:

$$HistUrban_{s,t} = HistUrban_{s,t+1} \times \frac{Pop_{s,t}}{Pop_{s,t+1}} \tag{2}$$

where $HistUrban_{s,t}$ and $HistUrban_{s,t+1}$ are the reconstructed urban land area of state $s$ in year $t$ and $t+1$; $Pop_{s,t}$ and $Pop_{s,t+1}$ are the total population of state $s$ at year $t$ and $t+1$.

**Comment 6:** P6, L106-108, which dataset were these criteria applied to? CPHR, CAHA, or other?

**Response**: Thank you for this suggestion. The cropland harvested area from CAHA is used for historical cropland reconstruction.

**Comment 7:** P6, L112, I could not find the CPHR dataset in Table 1. Please make sure the product name and time period is consistent between the table and manuscript.

**Response**: Thank you for this suggestion. The cropland planted area in Table 1 is the CPHR data. The cropland planted area was used to establish the relationship with cropland harvested area.

**Referee 1** thinks that converting cropland harvested area with an unchanged linear equation is not reasonable and dangerous. So, we change the cropland reconstruction method, and the cropland planted area is not used in the new method. Please see section 2.2.2.

**Comment 8:** P6, L117, 1975-2020, right? If not, please list this dataset clearly in Table 1. Please add more details about how the adjustment was done to make 'the inter-annual variation more reasonable'. For example, was this adjustment done at national scale then disaggregated to state scale?

**Response**: Thank you for this suggestion. The period of cropland planted area used before is 1975-2020. In the new method, the cropland planted area data is not used.

In the new method, we applied the national level cropland harvested area without double-cropped area to adjust the state level cropland harvested area during 1910-2020. The adjustment was conducted by dis-aggregating to the state level, which can be expressed as follows:

$$HistCrop_{s,t} = \frac{Cropland\ Harvested_{s,t}^{linear}}{Cropland\ Harvested_{conus,t}^{linear}} \times Cropland\ Harvested_{conus,t}^{ERS} \quad (5)$$

where $HistCrop_{s,t}$ is the reconstructed historical cropland area of state $s$ in year $t$; $Cropland\ Harvested_{s,t}^{linear}$ is the linear interpolated cropland harvested area of state $s$ in year $t$ based on CAHA data; $Cropland\ Harvested_{conus,t}^{ERS}$ is the national total cropland harvested area without double-cropped area.

**Comment 9:** P6, L125-126, what causes this difference? Due to the CAHA crop harvested area?

**Response**: Thank you for this suggestion. We assumed that there should be a large difference between the cropland harvest area between 1879 and 1889. However, we found that the cropland harvested area in South Dakota, Nebraska, and Kansas increased rapidly between 1879 and 1889 (Figure R2). Thus, we thought the cropland harvested area in 1879 is not correct before.

We rechecked the changes in cropland harvested area between 1879 and 1889 and total population (Figure R3) during 1850-1890. The rapidly increase of cropland area resulted from the population growth. we think the cropland harvested area should be right and can be used for cropland reconstruction.

[Figure]

**Figure R2**: Comparison between cropland harvested area in 1889 and 1879

[Figure]

**Figure R3**: Total population change in Nebraska, Kansas, and South Dakota between 1850 and 1890.

**Comment 10:** P6, L127, please justify why the cropland per capita was calculated at national level instead of state level like the urban area per capita. This could avoid any potential confusions about the following assumption you made.

**Response**: Thank you for this suggestion. Considering that HYDE is reconstructed based on the country level data, so the national level cropland per capita was used before. We agree that applying the state level cropland area per capita to estimate the total cropland area should be more reasonable.

In the new method, we change the harmonization method between the reconstructed historical cropland area and HYDE cropland. The trend of HYDE cropland area is used to reconstruct the historical cropland area at the state level as follows:

$$HistCrop_{s,t} = HistCrop_{s,t+1} \times \frac{HYDE\_Crop_{s,t}}{HYDE\_Crop_{s,t+1}} \tag{5}$$

where $HistCrop_{s,t}$ and $HistCrop_{s,t+1}$ are the reconstructed historical cropland area of state $s$ in year $t$ and t+1; $HYDE\_Crop_{s,t}$ and $HYDE\_Crop_{s,t+1}$ are HYDE cropland area of state $s$ in year t and t+1.

**Comment 11:** P7, L130-134, was there a harmonization process applied to connect cropland area during 1630-1880 derived from the HYDE and cropland area during 1889-2020 derived from the CAHA? If not, please add time-series plots of derived cropland areas that show such harmonization is not needed.

**Response**: Thank you for this suggestion. There was a harmonization process to connect cropland area derived from HYDE and reconstructed historical cropland area. The changing trend of HYDE cropland was used to reconstruct the historical cropland area at the state level, which can be expressed as follows:

$$HistCrop_{s,t} = HistCrop_{s,t+1} \times \frac{HYDE\_Crop_{s,t}}{HYDE\_Crop_{s,t+1}} \tag{5}$$

where $HistCrop_{s,t}$ and $HistCrop_{s,t+1}$ are the reconstructed historical cropland area of state $s$ in year $t$ and t+1; $HYDE\_Crop_{s,t}$ and $HYDE\_Crop_{s,t+1}$ are HYDE cropland area of state $s$ in year t and t+1.

**Comment 12:** P7, L141-L143, please clarify how the pasture per capita during 1630–1982 was estimated from pasture per capita at the state level (NRI data-based) in 1982 and the national level (HYDE data-based).

**Response**: Thank you for the suggestions. Same as the cropland area estimation, we agree that applying the state level pasture per capita to estimate the total pasture area should be more reasonable.

We change the pasture area estimation method by using the changing trend of HYDE pasture. The harmonization process can be expressed as follows:

$$HistPasture_{s,t} = HistPasture_{s,t+1} \times \frac{HYDE\_Pasture_{s,t}}{HYDE\_Pasture_{s,t+1}} \tag{6}$$

where $HistPasture_{s,t}$ and $HistPasture_{s,t+1}$ are the reconstructed historical pasture areas of state $s$ in year t and t+1; $HYDE\_Pasture_{s,t}$ and $HYDE\_Pasture_{s,t+1}$ are the HYDE pasture area of state $s$ in year t and t+1.

**Comment 13:** P7, L157, "... then multiplied the forest area from USDA-FR to generate …". What is the time period of USDA-FR forest area used here? 1630?

**Response**: Thank you for this suggestion. We revise it. …then multiplied the forest area from USDA-FR in 1630.

**Comment 14:** P8, L170, why is there no scaling factor for TA_t_r (s) < TLA(s) ? How was the difference/residual between TA_t_r (s) and TLA(s) dealt in your following analysis? And also, what is the source of the state's total land area (TLA)?

**Response**: Thank you for this suggestion. The purpose of this section is to check whether the total area of urban, cropland, pasture, and forest is larger than the state

land area. If the area is lower than the state land area, we will not modify the reconstructed area of each LULC class. Thus, there is no scaling factor for TA_t_r (s) < TLA(s). We don't do consider the difference/residual between TA_t_r (s) and TLA(s) dealt in the following analysis. In the updated version, we check the reconstructed area of each LULC class, the total area of urban, cropland, pasture, and forest at each state and each year stratify the condition "TA_t_r (s) < TLA(s)", so we remove this section.

For the state's total land area, we derive the area information from state boundary file. The file can be download in the following links:

https://www.nass.usda.gov/Publications/AgCensus/2017/index.php#highlights

**Comment 15:** P9, L182-184, please add references to how ANN can be used to solve the nonlinear geographical problems.

**Response**: Thank you for this suggestion. We add related references.

**Comment 16:** P9, L184-185, please add more details about how land use probability was generated from ANN and NLCD. And also, NLCD is supposed to be an independent variable used in the modeling, right? However, NLCD is a land cover data, how can it be useful for modeling of land use probability? Please keep in mind that land use and land cover are usually used interchangeably but are actually different terms. If you regard them as the same terms, why do you refer to it as land use probability instead of land use and land cover probability?

**Response**: Thank you for this suggestion. In this study, we used the ANN tool in Future Land Use Simulation (FLUS) software to estimate the probability of occurrence. The FLUS model is a land-use change simulation model that combines Cell Automa (CA) model and artificial neural networks (ANN) (Liu et al., 2017). The independent variables for the ANN model include elevation, slope, annual mean temperature, annual precipitation, annual maximum temperature (July), annual minimum temperature (January), crop productivity index, population density, distance to the city, distance to the road, distance to the railway, distance to the river, soil organic carbon, soil sand, soil clay. The purpose of ANN training is to establish the relationship between each LUCC type and the independent variables, and the NLCD Boolean type will be the dependent variable.

We agree that land use and land cover are usually used interchangeably but are different terms. In this study, we regard them as the same terms change the 'land use probability' as 'LULC probability'.

**Comment 17:** P9, L200, what are the difference between ES_weight_t and SE_weight_t in Eq. 4 and 6.

**Response**: Thank you for this suggestion. We revise it. Eq.4 should be SE_weight_t. In the revised method, the settlement extent is a binary map which is used to constrain the LULC change extent.

**Comment 18:** P9, L200, what are the values of t1 and t0 in Eq. 7? Are they the starting and ending year of each subperiod?

**Response**: Thank you for this suggestion. Yes, t0 and t1 are the starting and ending year of each subperiod. As our response for the Comment 17, the settlement extent is a binary map which is used to constrain the LULC change extent.

**Comment 19:** P10, L216, what does 'Boolean type' mean? Categorical type like NLCD that each grid (i.e, 30 m) has a single land use and land cover type?

**Response**: Thank you for this suggestion. Yes, categorical type.

**Comment 20:** P10, L217-218, please elaborate more on "the total number of potential pixels or the land use demand was determined based on the reconstruction results in Section 2.2. Then, the area difference of land use type k between the target and current map was calculated." The resulting data from section 2.2 is the state level total area of urban, crop, pasture and forest, right? So, such 'area difference' is at the state level, right?

**Response**: Thank you for this suggestion. We rewrite the spatial allocation process for both fractional LULC type and category type. We implement the spatial allocation algorithm at state level, so the resulting data from section 2.2 is the state level total area of urban, crop, pasture and forest and the 'area difference' is also at the state level. We rewrite the spatial allocation strategy and add a workflow, please see the response for the general comments.

**Comment 21:** P10, L228-236, the LUH2 used for validation, but it was not mentioned in section 2.1 and Table 1.

**Response**: Thank you for this suggestion. We revised the material and methods section and add the description of LUH2. Please see Table 1.

**Comment 22:** P10 L229, please justify why the validation against NLCD was at state-level. As you highlighted that your land use data is at 1 km, a finer spatial resolution than most other data, so the validation at 1 km will be more informative about the value of this dataset. A good agreement on state-level total land use area does not necessarily indicate the spatial allocation of total area to 1 km is good as well.

**Response**: Thank you for this suggestion. We agree with you that both data comparison or consistency check should be conducted at state-level and 1 km scale. Please see Figure R4 and R5.

**County-level cropland comparison:**

An accurate cropland distribution map is quite critical for historical LULC reconstruction. To prove the reliability of the newly developed cropland, we compared it with county-level census data. The spatial pattern of cropland proportion (cropland area/county area) from this study is close to the census data in the four years. Both two datasets show the cropland expansion in the North Central, the Great Plains, the Mississippi Alluvial Plain, and California. The cropland abandonment can also be found in the Appalachian Mountains. In the early period (1850), our results and census data show the cropland distribution in the central area of Alabama and Georgia, New England, and the North Central. But our reconstruction overestimated the cropland fraction in the North Central states, the east area of Virginia, North Carolina, South Carolina, and the south of Georgia. Because the census data only recorded the area of 18 major crops, our data is higher than the census data in the county with high cropland area. For example, cropland derived from this study was higher than the census data in the southeast coast, Atlantic coast, the Mississippi Alluvial Plain, northwest area of Texas, west of Oklahoma, and California in 1920, 1959, and 2002. While the cropland proportion in the Appalachian Mountains and the south area of the Great Plains was lower than the census data. This underestimation may result from the low cropland fraction in satellite data because it is hard for satellite data to extract the small area

cropland patch in the mountain area and classify the pasture/grassland with cropland in the south of the Great Plains.

[Figure]

**Figure R4.** Spatial comparison of county-level cropland proportion between our reconstruction and census data in 1850, 1920, 1959, and 2002. First column: cropland proportion from census data; Second column: cropland proportion derived from this study; Third column: cropland proportion between this study and census data.

**Grid-level LULC comparison:**
We compared our data with NLCD data at the grid level and calculate the difference. The spatial patterns of urban, cropland, pasture, and forest in this study are close to the satellite-based LULC data from NLCD, and most grids have a relatively small difference. For urban land, our result overestimates the fraction in the low urban density area, but about 90% of urban land grids whose differences range from -10% to 10%. For cropland, the area with a positive difference is mainly distributed in the Northeast states, Alabama, and Missouri, in which 67% of grids have small differences values lower than 10%. 39% of grids have negative difference values and are mainly located in the states with high cropland area. Moreover, most states in our reconstruction have a lower pasture fraction than NLCD data except for Oklahoma,

Arkansas, Texas, and Georgia. The grids with negative differences account for 49%. For forest, the reconstructed forest has a higher density than NLCD in the South, Pacific coast, and Great Lakes but underestimates the forest fraction in the central states, such as Missouri, Kentucky, and Ohio. There are 73% grids whose differences are relatively small and range from -10% to 20%.

[Figure]

**Figure R5:** Spatial comparison between our reconstruction and satellite-based urban, cropland, pasture, and forest. First column: Reconstructed data in this study (average between 2001 and 2019); Second column: Satellite-based data (average between 2001 and 2019); Third column: Difference between first column and second column; Fourth column: Distributions of fraction difference between our reconstructed database and satellite-based data.

**Comment 23:** P11, Figure 4. Such comparisons to different datasets are very important and informative. However, I would suggest more discussion on the 'outlier' states in figure 4a and 4b. For example, please dig into which states your estimates of urban land area is lower than HISDAC? And what are the potential causes?

**Response**: Thank you for this suggestion. We revised this section for the add more analysis about the differences between multiple datasets.

HISDAC area in Georgia, New York, North Carolina, Ohio, and Tennessee are significantly higher than NLCD, these can be explained that the property records for spatial reconstruction in these states have a low data missing rate. Meanwhile, the spatial resolution of HISDAC built-up area is 250 m, which is coarser than the NLCD.

**Comment 24:** P14, L290-L300, is the transition in figure 7 the gross transition or net transition? Please clarify this somewhere because their magnitude and impacts on climate and carbon modeling are quite different. For example, 30% deforestation and 30% reforestation have zero transition on forest, but the biophysical effects could not be ignored.

**Response**: Thank you for this suggestion. We agree that the effects of LULC gross transition and net transition are quite different. Figure 7 shows the LULC net transition, that is the LULC difference in 1630, 1850, 1920 and 2020.

**Comment 25:** P17, Figure 9d, the black line stops at near 1920.

**Response**: Thank you for this suggestion. We revise the Figure 9.

[Figure]

**Revised Figure 9**: Comparison with other datasets for the conterminous United States: urban land (a); cropland (b); pasture (c); forest (d). NLCD: National Land Cover Database; HYDE: History Database of the Global Environment; HISDAC: Historical Settlement Data Compilation; ERS: Economic Research Service; YLmap: Yu and Lu (2018) cropland density; ZCmap: Zumkehr and Campbell (2013) cropland fraction;

LUH2: Land Use Harmonization; FATD: Forest Area Trend Data; USDA-FR: USDA Forest Resources of the United States of 2017.

**Comment 26:** P17, Figure 9a and 9c, even though NLCD is the input of urban land area to your method, there is a difference in temporal trend in urban area between NLCD and your dataset. Please discuss more about what assumptions cause such trend difference, and how it affects the reconstruction before the 2000s. Same discussion is needed for the pasture as the trend is even opposite.

**Response**: Thank you for this suggestion. We estimated the historical urban land area by multiplying the urban land per capita (average between 2001 and 2016) and state total population. The area difference between NLCD and our estimation was induced by the increase rate difference between urban land and total population. In the revised method, the NLCD developed land area during 2001-2019 was regarded as the baseline. Considering the possible underestimation as you said in Comment 5, we revise the urban land estimation method by applying the HISDAC built-up area. The new estimation method can be found in the response of Comment 5.

For the pasture, we compared the state level NRI and NLCD data. The definitions of pasture in NRI and NLCD are different. In NLCD, pasture/hay areas is land of grasses, legumes, or grass-legume mixtures planted for livestock grazing or the production of seed or hay crops, typically on a perennial cycle. But pasture includes land that has a vegetative cover of grasses, legumes, and/or forbs, regardless of whether it is being grazed by livestock in NRI.

**Comment 27:** P20, L382-383, please explain why inventory-based data is more reliable than the satellite-based forest (NLCD) and biomass density-based forest (LUH2). Forests could have different definitions by various sources, and forest areas are subject to the definition, thus I could see which definition/data source is more reliable than others.

**Response**: Thank you for this suggestion. Exactly, it's hard to say which forest data is more reliable because of the differences between forest definitions (Please see Table R1).

In LUH2, the biomass density (BD) map is used to identify the potential forest (BD > 2 kg C m$^{-2}$) and non-forest at 0.5° × 0.5° resolution, which underestimates the forest in the Rocky Mountain region and Northwest. NLCD is produced by using Landsat images and a comprehensive method and provides nationwide data on land cover and land cover change at a 30 m resolution. So, it can provide a more accurate spatial pattern than the LUH2. While, the FIA data provides critical status and trend information through a system of annual resource inventory that covers both public and private forest lands across the United States (https://www.fs.usda.gov/research/inventory/FIA), and it can provide a forest trend data back to 1630. Thus, we think that the forest area from FIA is more credible and consistent for long-term LULC change study.

**Table R1.** Forest definitions from different data sources.

| Data source | Definitions |
| --- | --- |
| Forest Inventory Analysis (FIA) | Land at least 10 percent stocked by forest trees of any size, or formerly having such tree cover, and not currently developed for non-forest uses, with a minimum area classification of 1 acre (https://www.fia.fs.fed.us/tools-data/maps/2007/descr/yfor_land.php). |
| National Land Cover Database (NLCD) | Areas dominated by trees generally greater than 5 meters tall, and greater than 20% of total vegetation cover (https://www.mrlc.gov/data/legends/national-land-cover-database-class-legend-and-description). |
| Land Use Harmonization 2 (LUH2) | Forest was defined using a single tree canopy cover threshold to match the global forest extent provided by the FAO FRA report (Hurtt et al., 2020). |
| This study | Same as Forest Inventory Analysis (FIA). |

---

## Author Comment (AC3)

Thank you for the comments and suggestions. These comments were very helpful for revising and improving our paper. We have responded to the comments point by point.

**Reviewer #3:**

**General comments:**

This manuscript describes the development and details of a land-use dataset for the United States for the years 1630-2020. The dataset differs from other land-use datasets in that it is a high-resolution product, for almost 400 years of the historical period. The dataset combines multiple different input datasets, for different time-periods, and in different formats/spatial resolutions and reconstructs the historical areas of cropland, pasture, urban land, and forests annually at 1km x 1km spatial resolution, for the CONUS. The results show expansion of cropland and urban areas, with associated losses of natural vegetation. Comparison with other datasets show many areas of qualitative agreement, with some interesting differences for some time periods and land-use types.

Overall, the manuscript is mostly well-written and organized. It includes some useful information about the dataset development process, and an analysis of the resulting products. The dataset will be useful to modelers working in the areas of climate and ecosystems to better understand the high-resolution impacts of LCLUC in the CONUS over a long historical period. A few areas for improvement include:

**Specific comments:**

**Comment 1**: Although other alternative datasets are mentioned and compared with the new dataset, it would be helpful to know what advantages those other datasets might have (if any) over the new dataset (e.g. for HYDE an even longer time period is used, and for some datasets there could be additional data layers beyond the ones provided in this dataset, etc).

**Response**: Thank you for this suggestion. We add the related discussion in section 4.1 and 4.3. Compared with HYDE and LUH2, we improve the spatial resolution from 5 arcmin or 0.25 degree to 1 km. The newly developed dataset can show more detailed information than HYDE and LUH2. Please see Figure R1, R2, and R3.

[Figure]

**Figure R1:** Visual comparison between newly developed cropland and HYDE3.2, YLmap, ZCmap in four different sites (a-d). The locations of image center points are as follows: a. Ohio (83.05 °W, 40.17 °N), b. Georgia (83.58 °W, 32.77 °N), c. Arkansas (90.56 °W, 34.76 °N), d. Texas (100.92 °W, 32.81°N).

[Figure]

**Figure R2:** Visual comparison newly developed pasture and HYDE3.2, LUH2 in four different sites (a-d). The locations of image center points are as follows: a. Iowa (93.64 °W, 42.03 °N), Virginia (78.72 °W, 37.96 °N), c. Illinois (90.07 °W, 38.68 °N), d. Arkansas (92.56 °W, 34.97 °N).

[Figure]

**Figure R3**: Visual comparison between the newly developed forest and Land Use Harmonization (LUH2) in four different sites (a-d). The locations of image center points are as follows: a. Colorado (106.47 °W, 38.97 °N), Wisconsin (89.85 °W, 44.54 °N), c. Alabama (86.72 °W, 33.33 °N), d. New York (75.14 °W, 42.21 °N).

We collected the available state-level census data to reconstruct the LULC history for the CONUS, which can overcome some weaknesses in the previous datasets. For example, the HYDE and LUH2 overestimate the cropland acreages compared with YLmap and USDA census data, and we fixed this problem. But the HYDE and LUH2 data still are good LULC products to analyze LULC change and simulate its ecological impacts for global scale study.

**Comment 2**: There are different versions of HYDE3.2 – it would be good to know which one was used in this manuscript.

**Response**: Thank you for this suggestion. The HYDE3.2 baseline version was used, and we add the information in the Table 1.

**Comment 3**: Does the pasture category in the dataset include natural grasslands, as well as managed grasslands and rangelands?

**Response**: Thank you for this suggestion. In this study, pasture is defined as a land cover/use category of land managed primarily for the production of introduced forage plants for livestock grazing, consistent with the National Resource Inventory. So, it doesn't include natural grasslands and rangelands.

**Comment 4**: I also had a bit of confusion about the forest category in the dataset – is it primarily about land that is being used as a forest (regardless of the numbers or ages of trees)? Or is it based more on forest land cover? This distinction between land use and land cover could be discussed a bit more to help with this. There are several places in the manuscript where the authors state that forest area decreased due to wood harvest or fuelwood extraction, but if that did not result in a conversion to another land-use type, then the forest area would not be changed (even if the land cover changed).

**Response**: Thank you for this suggestion. In this study, we use the forest definition from FIA. Forest is the land at least 10 percent stocked by forest trees of any size, or formerly having such tree cover, with a minimum area classification of 1 acre (https://www.fia.fs.fed.us/tools-data/maps/2007/descr/yfor_land.php). The newly developed forest dataset doesn't include the tree numbers or age information, and it is land used as forest.

Moreover, we agree your opinion about the wood harvest or other management activities may not change the land use type. If the forest land is cleared, the forest land will be converted to another LULC type. We revise the related description in the manuscript.

**Comment 5:** I found the color scale on figures 5 and 7 quite difficult to read to distinguish between the various land-use colors.

**Response**: Thank you for this suggestion. For figure 5, the color of pasture and grassland makes people difficult to read. We change the colors and update the figure to make it easy to read. Please see Figure R4 (Figure 8 in the revised manuscript). For figure 7, there are 12 types of LULC conversion to show and it is hard to assign a suitable color, so we add a table (Table 3) to record the LULC conversion area in four periods (1630-1850, 1850-1920, 1920-2020, and 1630-2020).

[Figure]

**Figure R4:** Spatial and temporal patterns of land use and land cover in the conterminous United States during 1630-2020.

**Comment 6:** Overall, I think a discussion of the differences between land-use and land-cover and how that is represented in this dataset would be a helpful addition. Also, some more discussion of how this product differs from the technical details of other products and in what ways that is useful and in what ways other products might have some advantages, along with how those differences in underlying details are driving differences in the qualitative dataset results.

**Response**: Thank you for this suggestion. In *section 4.3*, we add the discussion about the uncertainties induced by the LULC definition differences. For the technical details, it should include the LULC area reconstruction strategy and spatial allocation strategy. We discussed the LULC probability calculation and spatial allocation algorithm difference between this study and other land use and land cover simulation model. In fact, the key to the spatial allocation algorithm is what ways you choose to allocate the LULC area. Some LULC simulation models allocate the LULC demand (net change) at a LULC base map and generate a new LULC map in the prediction year. But this algorithm will underestimate the gross LULC change area whatever it is used to generate fractional or Boolean type data. It is also not suitable for long-term LULC simulation, because they assumed the LULC probability or suitability surface is stable. In this study, because the contemporary LULC probability pattern is not representative for the early period, we need to modify the probability to make it close to the historical LULC pattern. Therefore, we used the spatial allocation algorithm in this study and generate a map for each year. But this strategy ignores the linkages of landscape in the neighboring two years.

---

## Author Comment (AC4)

Thank you for the comments and suggestions. These comments were very helpful for revising and improving our paper. We have responded to the comments point by point.

**Reviewer #1:**

**General comments:**

The paper takes on the very substantial challenge of recreating historical land use for the United States. The authors are correct that outside of coarse-level reconstructions such as HYDE, that there is nothing of higher resolution that goes back to pre-settlement by European colonists. The authors pitch this methodology and dataset as the solution to that data gap. I don't think the authors are completely successful in making that argument. After stating the need for higher-resolution data, and presenting a methodology that should allow for some higher-level of spatial detail than something like HYDE, the authors inexplicably leave out any kind of spatial analysis of how well their model result performs. It's difficult when spatially explicit data on land use is hard to come by (and hence the need for this work!), but some analysis of the full-resolution data compared to a data set such as NLCD (available 2001 to 2019) could have helped establish confidence in the model to capture spatial patterns well. Even a comparison to historical county-level data would help. However, the only "spatial" analysis of the data are some very coarse regional assessments that don't provide a reader much of a feel for the model's capability for generating realistic, high-resolution spatial patterns. Validation overall is a weak point of the paper. On the one hand, I understand the difficulties in trying to "validate" results such as this, when consistent reference data is absent or scattered. However, it's not acceptable from a modeling perspective to use HYDE, NLCD, and other data to parameterize the model, and also use those data in what's labeled as "validation" of results. Given the lack of spatially explicit reference data, I have no issue with the authors doing "consistency checks" with other data sets, including missed opportunities such as county-level ag census data that could have been used to provide a better feel for the spatial patterns produced by the model. But don't try to sell it as a real "validation" of model results.

Overall, the authors continually note the "Uncertainties" associated with coarser data such as HYDE, but fail to conclusively demonstrate their results are superior in terms of those uncertainties. A major need for the paper is recognition of the differences

between source datasets, and the uncertainties it introduces into the modeling. For example, many parts of the model are parameterized by using multiple datasets that have inherent differences. Pasture, for example, is quantified at a state level by NRI, and then HYDE for older dates. Given how variable definitions are for "pasture" in the first place, it's asking for trouble to mix and match datasets such as that. The authors do seem to make some attempts to harmonize differences in datasets, but the methodology isn't well-defined enough to let me know if that's really being done. In short, historical land use reconstruction is difficult because 5 different datasets may give you 5 different answers for how much "forest", "urban", "cropland", or "pasture" is actually there for a given date! The spatial allocation of "change" on the landscape is also quite simplistic, based solely on probability surfaces with no stochasticity. The authors do attempt to mitigate the 'static probability surface' problem present for past applications such as CLUE and FORE-SCE by some simple weighting with population. But the actual allocation is fully dependent on the probability surfaces at the end of the day, and as a result, as you go back in time, you tend to see classes such as cropland and pasture become concentrated in the high-probability locations, with less fragmentation that's there in later dates.

Finally, note I did download and look at some of the output results. For brevity I'll keep my comments here to the "Boolean" land cover. Note that while results look reasonable at broad scales, the approach of parameterizing state-by-state does seem to cause some issues as you go back in time, as does the spatial allocation methodology. Going back in time reveals a number of obvious state and even what appear to be county boundaries, hard obvious lines where land use clearly differs on either side of a political boundary. Given the complete reliance on probability surfaces alone for the spatial allocation of change, land use looks more concentrated on the landscape for some classes going back in time. For example, on the 1850 map, cropland is concentrated in very large contiguous chunks in many areas, and very sharp and obvious political boundaries are present.

**Response**: Thank you for the comments and suggestions.

(1) According to your suggestions, we conducted another two data comparisons to increase the newly developed LULC data's confidence in section 3.1. First, we did a comparison between our reconstruction with USDA county-level crops area in 1850, 1920, 1959, and 2002 (Figure R1). Second, we compared our data with NLCD at the grid level and calculate the difference (Figure R2). Please see Lines 304 to 338.

[Figure]

**Figure R1.** Spatial comparison of county-level cropland proportion between our reconstruction and census data in 1850, 1920, 1959, and 2002. First column: cropland proportion from census data; Second column: cropland proportion derived from this study; Third column: cropland proportion between this study and census data.

[Figure]

**Figure R2:** Spatial comparison between our reconstruction and satellite-based urban, cropland, pasture, and forest. First column: Reconstructed data in this study (average between 2001 and 2019); Second column: Satellite-based data (average between 2001 and 2019); Third column: Difference between first column and second column; Fourth column: Distributions of fraction difference between our reconstructed database and satellite-based data.

(2) As you said, the definitional differences did increase the uncertainties of LULC modeling because we had to harmonize multiple datasets for each LULC type. Thus, we compared the definitions for cropland, pasture, and forest from the multisource datasets, please see Table R1, R2, and R3. Moreover, to make readers understand the urban, cropland, pasture, and forest reconstruction process more easily, we add a table (Table 2) to introduce the LULC definitions used in this study.

**Table R1:** Definition of cropland in different data sources.

| Data source | Definition |
|---|---|
| USDA-ERS | *Cropland*: Total cropland includes five components: cropland harvested, crop failure, cultivated summer fallow, cropland used only for pasture, and idle cropland (https://www.ers.usda.gov/data-products/major-land-uses/glossary/#croplandforcrops). |
| USDA-NRI | *Cropland*: A land cover/use category that includes areas used for the production of adapted crops for harvest. Two subcategories of cropland are recognized: cultivated and non-cultivated. Cultivated land comprises land in row crops or close-grown crops, as well as other cultivated cropland; for example, hayland or pastureland that is in a rotation with row or close-grown crops. Non-cultivated cropland includes permanent hayland and horticultural cropland. |
| NLCD | *Cultivated Crops*: areas used for the production of annual crops, such as corn, soybeans, vegetables, tobacco, and cotton, and also perennial woody crops such as orchards and vineyards. Crop vegetation accounts for greater than 20% of total vegetation. This class also includes all land being actively tilled (https://www.mrlc.gov/data/legends/national-land-cover-database-class-legend-and-description). |
| HYDE | FAO categories of "arable land and permanent crops" (Klein Goldewijk et al., 2017). |

**Table R2:** Definition of grazing land, pasture, and rangeland in different data sources.

| Data source | Definition |
|---|---|
| USDA-ERS | *Cropland pasture*: Cropland pasture includes acres of crops hogged or grazed but not harvested and some land used for pasture that could have been cropped without additional improvement.
 *Grassland pasture and range*: Grassland pasture and range encompass all open land used primarily for pasture and grazing, including shrub and brush-land types of pasture, grazing land with sagebrush and scattered mesquite, and all tame and native grasses, legumes, and other forage used for pasture or grazing—regardless of ownership.
 *Forest land grazed*: Forested pasture and range consisting mainly of forest, brush-grown pasture, arid woodlands, and other areas within forested areas that have grass or other forage growth.
 https://www.ers.usda.gov/data-products/major-land-uses/glossary/ |
| USDA-NRI | *Pasture*: A land cover/use category of land managed primarily for the production of introduced forage plants for livestock grazing. Pastureland cover may consist of a single species in a pure stand, a grass mixture, or a grass-legume mixture. Management usually consists of cultural treatments: fertilization, weed control, reseeding, renovation, and control of grazing. For the NRI, includes land that has a vegetative cover of grasses, legumes, and/or forbs, regardless of whether or not it is being grazed by livestock (U.S. Department of Agriculture, 2020). |

*Rangeland*: A broad land cover/use category on which the climax or potential plant cover is composed principally of native grasses, grass-like plants, forbs or shrubs suitable for grazing and browsing, and introduced forage species that are managed like rangeland. This would include areas where introduced hardy and persistent grasses, such as crested wheatgrass, are planted and such practices as deferred grazing, burning, chaining, and rotational grazing are used, with little or no chemicals or fertilizer being applied. Grasslands, savannas, many wetlands, some deserts, and tundra are considered to be rangeland. Certain communities of low forbs and shrubs, such as mesquite, chaparral, mountain shrub, and pinyon-juniper, are also included as rangeland (U.S. Department of Agriculture, 2020).

| | |
|---|---|
| EPA | *Pastures***:** Pastures are those lands that are primarily used for the production of adapted, domesticated forage plants for livestock. |
| | *Rangelands***:** Rangelands are those lands on which the native vegetation (climax or natural potential plant community) is predominantly grasses, grass-like plants, forbs, or shrubs suitable for grazing or browsing use. Rangelands include natural grassland, savannas, many wetlands, some deserts, tundra, and certain forb and shrub communities. https://www.epa.gov/agriculture/agricultural-pasture-rangeland-and-grazing |
| NLCD | *Pasture/Hay***:** Areas of grasses, legumes, or grass-legume mixtures planted for livestock grazing or the production of seed or hay crops, typically on a perennial cycle. Pasture/hay vegetation accounts for greater than 20% of total vegetation. https://www.mrlc.gov/data/legends/national-land-cover-database-class-legend-and-description |
| HYDE | *Grazing land***:** Land used for mowing or grazing livestock, based on the FAO category "permanent meadows and pastures". Grazing land can be a variety of ecosystems, ranging from managed irrigated grasslands to unmanaged open savannah woodlands to semi-shrub/scrub, almost desert, lands (Klein Goldewijk et al., 2017). |
| | *Pasture***:** Pasture is high-intensity grazing land, or low intensity grazing lands where a conversion of the natural vegetation has occurred (Klein Goldewijk et al., 2017). |
| | *Rangeland***:** rangeland is low-intensity grazing land where the natural vegetation has not been converted (Goldewijk et al., 2017). |

**Table R3.** Forest definitions from different data sources.

| Data source | Definition |
|---|---|
| FIA | Land at least 10 percent stocked by forest trees of any size, or formerly having such tree cover, and not currently developed for non-forest uses, with a minimum area classification of 1 acre (https://www.fia.fs.fed.us/tools-data/maps/2007/descr/yfor_land.php). |

| NLCD | Areas dominated by trees generally greater than 5 meters tall, and greater than 20% of total vegetation cover (https://www.mrlc.gov/data/legends/national-land-cover-database-class-legend-and-description). |
| LUH2 | Forest was defined using a single tree canopy cover threshold to match the global forest extent provided by the FAO FRA report (Hurtt et al., 2020). |

(3) For the spatial allocation, we agree with you that the actual allocation is dependent on the probability surfaces and the landscape pattern at the end of the day. CLUE model and FORE-SCE model generate a LULC map at the predicted year by allocating the LULC demand (LULC area net change) to a LULC base map. This method works well for short-period studies because they can assume that the large-scale LULC pattern is stable. To be honest, we also tried such a spatial allocation method for generating fractional and Boolean-type data. However, the contemporary LULC pattern is not representative for the historical LULC pattern even going back to the 1940s in CONUS (Sohl et al., 2016). Thus, we need to add some modifiers (e.g., population density) to improve the LULC probability. We know it is simple by adding a population weight, but it is effective. Because the distribution of human-related LULC types (e.g., urban, crop, and pasture) was always correlated with population density in the early period. In Figure R3, we can see that the county-level population and cropland proportion show the same spatial patterns in 1850.

[Figure]

**Figure R3:** County-level cropland proportion (left) and population density (right) in 1850.

(4) To generate the fractional gridded LULC data, we assumed that the fraction of each LULC type at the grid level is determined by the total probability (Fuchs et al., 2013; Tian et al., 2014; West et al., 2014; He et al., 2015). It means that a grid cell (LULC type k) with a high probability will have a high fraction. Based on this principle and the state-level LULC area, we generated the fractional LULC data at 1 km x 1 km resolution and annual time scale. The detailed information for generating fractional LULC data is shown in the following steps: (1) prepare the input data: state-level historical LULC area and probability; (2) calculate the state target LULC fraction for type k and initialize an empty LULC fraction surface; (3) calculate a temporal fraction surface; (4) modify the temporal fraction, we assume that the fraction of water and barren is stable, and the sum of urban, crop, pasture, and forest fraction is lower than the maximum fraction in each grid cell; (4) add the temporal fraction data to the empty LULC fraction; (5) judge whether the unallocated LULC area is smaller than $0.01 \text{ km}^2$, if yes, the iteration will stop and begin to allocate another LULC type, else the unallocated area will be assigned to target fraction and return to step (3). The allocation was processed iteratively until the unallocated area was less than the threshold ($0.01 \text{ km}^2$). The above steps will be conducted for each state, and urban, cropland, pasture, and forest fractional map in the CONUS will be output.

Based on the LULC fraction map, we generated the Boolean type LULC data at 1 km x 1 km resolution. The detailed information for is shown the following steps: (1) prepare the input data: state-level historical LULC area and LULC fraction data; (2) generate a temporal LULC map (HistB) through identifying the dominate LULC type in each grid cell and initialize an empty LULC map (HisB$_E$); (3) calculate the area difference for LULC type k between the HisB map and target area; (4) if the area difference is negative, we first sort the LULC fraction data where HisB equals to k, the top m (equals to target area) grid cells where HisB$_E$ not be assigned a value will be assigned as k, then if the available number of grid cell (type k) is less than the target area, we will sort the LULC fraction data where HisB map not equal to k, and the top n (equals to unallocated area) grid cells where HisB$_E$ not be assigned a value will be assigned as k; (5) if the area difference is positive, the grid cells where HisB data equals to k and the will be assigned k to HisBE$_E$ not be assigned a value; then we will sort the LULC fraction data where HisB data not equals to k, and the top n (equals to unallocated area) grid cells where HisB$_E$ not be assigned a value will be assigned as k. If step (4) and (5) finish, the next LULC type will begin to allocate. After the four LULC types of allocation finish,

the grid cell not be assigned a type will be updated using the HisB data and LANDFIRE Biophysical Settings data.

[Figure]

**Figure R4:** Workflow for generating fractional (left) and Boolean (right) type LULC data.

(5) For the Boolean type data, you proposed several suggestions as follows:
- *Cropland and pasture become concentrated in the high-probability locations, with less fragmentation that's there in later dates.*
- *Going back in time reveals a number of obvious state and even what appear to be county boundaries, hard obvious lines where land use clearly differs on either side of a political boundary.*
- *Given the complete reliance on probability surfaces alone for the spatial allocation of change, land use looks more concentrated on the landscape for some classes going back in time.*

We re-check the Boolean type LULC data and analyze the reasons for these problems. The 'political boundary' issue resulted from the following two aspects. The first is that we conduct the spatial allocation at the state level. If the area of one LULC type has a large difference between two neighboring states, there would be a 'political

boundary'. The second reason is that we use county-level population density (one value in a county) to modify the LULC probability in the early period, which will result in the hardlines between neighborhood counties. The LULC '*concentration*' problems also resulted from that we apply the population density data to modify the LULC probability surface. In our revised version, we optimized the population density weight and Boolean type spatial allocation method. A detailed description of the method can be found in section 2.4.1 and 2.4.2. Figure R5 shows the comparison of LULC at the local scale in 1850.

[Figure]

**Figure R5:** Comparison of Boolean type LULC map in 1850 between before and after optimization.

**References**:

Goldewijk, K. K., Beusen, A., Doelman, J., and Stehfest, E.: Anthropogenic land use estimates for the Holocene - HYDE 3.2, Earth Syst. Sci. Data, 9, 927-953, https://doi.org/10.5194/essd-9-927-2017, 2017.

Hurtt, G. C., Chini, L., Sahajpal, R., Frolking, S., Bodirsky, B. L., Calvin, K., Doelman, J. C., Fisk, J., Fujimori, S., Goldewijk, K. K., Hasegawa, T., Havlik, P., Heinimann, A., Humpenoder, F., Jungclaus, J., Kaplan, J. O., Kennedy, J., Krisztin, T., Lawrence, D., Lawrence, P., Ma, L., Mertz, O., Pongratz, J., Popp, A., Poulter, B., Riahi, K., Shevliakova, E., Stehfest, E., Thornton, P., Tubiello, F. N., van Vuuren, D. P., and Zhang, X.: Harmonization of global land use change and management for the period 850-2100 (LUH2) for CMIP6, Geosci. Model Dev., 13, 5425-5464, https://doi.org/10.5194/gmd-13-5425-2020, 2020.

Sohl, T., Reker, R., Bouchard, M., Sayler, K., Dornbierer, J., Wika, S., Quenzer, R., and Friesz, A.: Modeled historical land use and land cover for the conterminous United

States, J. Land Use Sci., 11, 476-499, https://doi.org/10.1080/1747423X.2016.1147619, 2016.

U.S. Department of Agriculture.: Summary Report: 2017 National Resources Inventory, Natural Resources Conservation Service Washington, DC, and Center for Survey Statistics and Methodology, Iowa State University, Ames, Iowa, 2020.

**Specific comments:**

**Comment 1:** Lines 28 – Would add one word…" In particular, managing agriculture and forest-related activities…"

**Response**: Thank you for this suggestion. We revise those sentences. Please see Line 31.

**Comment 2:** Line 34 – Would add words…"…arrival of Europeans, indigenous communities practiced agriculture and crop planting in the…"

**Response**: Thank you for this suggestion. We revise those sentences. Please see Line 36.

**Comment 3:** Line 36 – Would change "mainly occurred" to "initially occurred" to indicate these activities first started here, but expanded elsewhere later (as noted by the next sentences)

**Response**: Thank you for this suggestion. We revise those sentences. Please see Line 28.

**Comment 4:** Line 37 – "Driven" not "driving".

**Response**: Thank you for this suggestion. We revise those sentences. Please see Line 40.

**Comment 5:** Page 2 – Overall the paragraph at the top of page 2 could use some work. It's a rather disjointed history of US land change. For one it doesn't really talk about land change west of the Mississippi River, it's focused solely on Eastern US change. The organization is also a bit odd and disjointed.
The sentence on line 44, for example, seems like a very abrupt and odd ending to the final statement as to why a long-term land use dataset is needed. Perhaps a better organization by period (colonial, 19th century, 20th century), with a description of

what occurred in each century? And perhaps a modification of the last sentence, adding "While general trends in historical US landscape change are known, we still lack a long-term dataset…"

Response: Thank you for this suggestion. We rewrite this paragraph. Please see Lines 34-52.

Comment 6: Line 55-56 – Be careful about highlighting "uncertainties" in datasets such as HYDE, as your historical landscape construction will also have substantial uncertainties. Your workflow itself uses HYDE data. There's limited spatially explicit data available from which to base a model-based landscape reconstruction, and many of the datasets you're using were also used by HYDE.

Response: Thank you for this suggestion. We revised this sentence. Please see Lines 59-60. We agree with you that the uncertainties of HYDE should not be highlighted because HYDE data is also one of the input data in our reconstruction. The historical LULC pattern in the CONUS has changed a lot compared with the contemporary LULC pattern. This is also the reason that we used the population density and human settlement extent to improve the LULC probability, even though it is a simple method.

Comment 7: Lines 87-88 – I wouldn't call the use of these other datasets "validation". It's a consistency check, not a validation, as these data sets too have uncertainties, and some are modeled just as you're modeling.

Response: Thank you for this suggestion. These data should be the input for consistency check rather than validation. We rewrite the description about the data comparison or consistency check. We rewrite the section 2.4. Please see Lines 266-275. Moreover, we also add another two data comparisons. Please see section 3.1.2 and 3.1.3 (Lines 304-338).

Comment 8: Line 90-91 – How was resampling done to get to 1-km grid cells? Is it fractional LULC within a given 1-km cell for datasets with native resolution <1 km?

Response: Thank you for this suggestion. The input data were resampled (nearest method) or aggregated to 1 km resolution. If the dataset with native resolution < 1 km, it will be fractional LULC within a given 1-km cell. For the NLCD dataset, we

generated the fractional type data to 1 km for each LULC type (urban, cropland, pasture, and forest) using the "aggregate" method.

**Comment 9:** Section 2.2.1 – This is an extremely simplistic methodology for calculating urban land area. To start, it's all based on one current dataset, NLCD. How was NLCD used? First of all, NLCD tends to underestimate low-density residential lands, which can bias your results. Secondly, NLCD "urban" classes also include extensive representation of road networks, which if counted as "urban", greatly overestimates urban land. For a rural state, for example, NLCD classes not only major roads, but every small section road has a 1-pixel-wide "urban" class representing it. Unless measures were taken to account for NLCD's underrepresentation of low-density residential lands, and to account for all the "urban" pixels that are really roads, it biases the results. The other problem is the very simplistic method for calculating land area. You're assuming the relationship between urban land per capita and total population is constant through time. Clearly it's not. Without accounting for that changing relationship, urban estimates can easily be biased.

**Response**: Thank you for this suggestion. In this study, we count the four components (Developed, Open Space; Developed, Low Density; Developed, Medium Density; Developed, High Density) of NLCD developed land as urban land. And the developed land area between 2001 and 2019 is regarded as baseline data for historical urban land area reconstruction. Though NLCD has some shortcomings, we need to choose a dataset with a clear definition and spatial explicit map, which will be helpful for historical urban land reconstruction.

In the current method, we used the population and a stable urban land per capita to estimate the long-term urban area at the state level. We agree that it is a very simplistic method and also pointed out that it overestimates the total urban land area in the early period. To solve this problem, we apply the HISDAC data to reduce the bias between 1810 and 2000, and the method has been updated in *section 2.2.1*. Before 1810, there is no available data that can be used, and we assumed that urban land has the same change rate as the total population at the state level. Please see Lines 105-122.

**Section 2.2.1**

In this study, we used the same definition for the developed land as NLCD for urban land. The developed land in NLCD includes four components: open space, low intensity developed land, medium intensity developed land, and high intensity developed land (Table 2). We used the NLCD developed land area during 2001–2019 as the urban land area baseline. Before 2001, we applied Historical Settlement Data Compilation for the United States (HISDAC-US) (Leyk et al., 2020; Uhl et al., 2021) as input to reconstruct the historical urban land area. The HISDAC-US built-up areas describes the built environment for most of the CONUS from 1810 to 2015 at 5-year temporal and 250 m spatial resolution using built-up property records, locations, and intensity data (Leyk and Uhl, 2018; Uhl et al., 2021). Here, we assumed that the HISDAC built-up areas data could capture the trend of urban land development. Then, the historical urban land can be estimated as follows:

$$HistUrban_{s,t} = HistUrban_{s,t+1} \times \frac{HISDAC_{s,t}}{HISDAC_{s,t+1}} \tag{1}$$

where $HistUrban_{s,t}$ and $HistUrban_{s,t+1}$ are the reconstructed urban land area of state $s$ in year $t$ and $t$+1; $HISDAC_{s,t}$ and $HISDAC_{s,t+1}$ are the HISDAC built-up area of state $s$ in year $t$ and $t$+1.

There is no census data on urban land area before 1810. Following Liu et al. (2010), we used population to estimate the urban land area by assuming that urban land expanded at the same rate as total population during 1630–1810. The urban land area of each state can be calculated as follows:

$$HistUrban_{s,t} = HistUrban_{s,t+1} \times \frac{Pop_{s,t}}{Pop_{s,t+1}} \tag{2}$$

where $HistUrban_{s,t}$ and $HistUrban_{s,t+1}$ are the reconstructed urban land area of state $s$ in year $t$ and $t$+1; $Pop_{s,t}$ and $Pop_{s,t+1}$ are the total population of state $s$ in year $t$ and $t$+1.

**References:**

Leyk, S. and Uhl, J. H.: HISDAC-US, historical settlement data compilation for the conterminous United States over 200 years, Sci. Data, 5, 1-14, https://doi.org/10.1038/sdata.2018.175, 2018.

Leyk, S., Uhl, J. H., Connor, D. S., Braswell, A. E., Mietkiewicz, N., Balch, J. K., and Gutmann, M.: Two centuries of settlement and urban development in the United States, Sci. Adv., 6, https://doi.org/10.1126/sciadv.aba2937, 2020.

Liu, M. and Tian, H.: China's land cover and land use change from 1700 to 2005: Estimations from high-resolution satellite data and historical archives, Global Biogeochem. Cy., 24, GB3003, https://doi.org/10.1029/2009gb003687, 2010.

Uhl, J. H., Leyk, S., McShane, C. M., Braswell, A. E., Connor, D. S., and Balk, D.: Fine-grained, spatiotemporal datasets measuring 200 years of land development in the United States, Earth Syst. Sci. Data, 13, 119-153, https://doi.org/10.5194/essd-2020-217, 2021.

**Comment 10:** Section 2.2.2 – You're using (at least) three different data sources to help establish cropland area. For historical land use, estimates vary widely, dependent upon methodology, data source, thematic definitions of a land use, etc. As a result, when switching from USDA-based data, for example, after 1889, and using HYDE before 1889, you'd expect an obvious break in estimated "cropland" amounts. How were those inconsistencies among historical land use datasets harmonized?

**Response**: Thank you for this suggestion. In the current method, we first convert cropland harvest area to planted area by assuming a stable linear relationship between these two datasets, then we calculated the cropland area by subtracting the double-cropped area from the planted area. However, these two steps will result in some uncertainties as you said in Comment 11 and 12.

To reduce these uncertainties, we decided to change the input data for historical cropland reconstruction, including the ERS national cropland harvested area (without double-cropped area), the state level CAHA-cropland harvested area (1879-2017) and HYDE cropland (1630-1879) to reconstruct the historical cropland area. Information about the updated method please see Lines 124-153 or the following description.

*Section 2.2.2*

The definition of cropland varies in the existing literature and datasets (Zumkehr and Campbell, 2013; Bigelow and Borchers, 2017; Goldewijk et al., 2017; Homer et al., 2020, Table R1). Cropland, defined by the U.S. Department of Agriculture (USDA) Economic Research Service (ERS), includes five components: cropland harvested, crop failure, cultivated summer fallow, cropland pasture, and idle cropland (Table 2). In this

study, we only count the cropland harvested area, which includes row crops and closely sown crops, hay and silage crops, tree fruits, small fruits, berries, and tree nuts, vegetables and melons, and miscellaneous other minor crops (https://www.ers.usda.gov/data-products/major-land-uses/glossary/#cropland). USDA Census of Agriculture Historical Archive (CAHA) recorded state-level cropland harvested areas at 4 to 10 years intervals (Table 1 and Table S5), which was used for historical cropland area reconstruction between 1879 and 2017. The CAHA cropland was interpolated into annual using the linear method first. To subtract the double-cropped area, we applied the annual national cropland harvested area without double-cropped area from ERS Major Land Uses data to adjust the interpolated cropland harvested area. The adjustment can be expressed as follows:

$$HistCrop_{s,t} = \frac{Cropland\ Harvested_{s,t}^{linear}}{Cropland\ Harvested_{conus,t}^{linear}} \times Cropland\ Harvested_{conus,t}^{ERS} \qquad (3)$$

where $HistCrop_{s,t}$ is the reconstructed cropland area of state $s$ in year $t$; $Cropland\ Harvested_{s,t}^{linear}$ is the linearly interpolated cropland harvested area of state $s$ in year $t$ based on CAHA cropland harvested area; $Cropland\ Harvested_{conus,t}^{ERS}$ is the national total cropland harvested area without double-cropped area in year $t$. For 2018–2020, the state-level cropland area was calculated based on the state-level area weight in 2017.

For 1879–1910, there was no national-level cropland harvested area without double-cropped area. Therefore, we applied the trend of the CAHA cropland harvested area to reconstruct the historical cropland:

$$HistCrop_{s,t} = HistCrop_{s,t+1} \times \frac{CAHA\_CHA_{s,t}}{CAHA\_CHA_{s,t+1}} \qquad (4)$$

where $HistCrop_{s,t}$ and $HistCrop_{s,t+1}$ are the reconstructed cropland area of state $s$ in year $t$ and $t+1$; $CAHA\_CHA_{s,t}$ and $CAHA\_CHA_{s,t+1}$ are the cropland harvested area of state $s$ in year $t$ and $t+1$.

Because there was no available cropland census data at the state level before 1879, the HYDE cropland was used. We first estimated the cropland per capita by applying the trend of HYDE cropland per capita. Then, the total cropland area can be calculated by multiplying cropland per capita and total population. The data harmonization process can be expressed as follows:

$$HistCrop_{s,t} = (HistCrop\_p_{s,t+1} \times \frac{HYDE\_Crop\_p_{s,t}}{HYDE\_Crop\_p_{s,t+1}}) \times Pop_{s,t} \qquad (5)$$

where $HistCrop_{s,t}$ is the reconstructed cropland area of state $s$ in year $t$; $HistCrop\_p_{s,t+1}$ is the reconstructed cropland per capita of state s in year $t+1$; $HYDE\_Crop\_p_{s,t}$ and $HYDE\_Crop\_p_{s,t+1}$ are HYDE cropland per capita of state $s$ in year $t$ and $t+1$.

**References:**

Bigelow, D. P. and Borchers, A.: Major Uses of Land in the United States 2012, U.S. Department of Agriculture, Economic Research Service, 2017.

Goldewijk, K. K., Beusen, A., Doelman, J., and Stehfest, E.: Anthropogenic land use estimates for the Holocene - HYDE 3.2, Earth Syst. Sci. Data, 9, 927-953, https://doi.org/10.5194/essd-9-927-2017, 2017.

Homer, C., Dewitz, J., Jin, S., Xian, G., Costello, C., Danielson, P., Gass, L., Funk, M., Wickham, J., Stehman, S., Auch, R., and Riitters, K.: Conterminous United States land cover change patterns 2001-2016 from the 2016 National Land Cover Database, ISPRS. J. Photogramm. Remote Sens., 162, 184-199, https://doi.org/10.1016/j.isprsjprs.2020.02.019, 2020.

Zumkehr, A. and Campbell, J. E.: Historical U.S. Cropland areas and the potential for bioenergy production on abandoned croplands, Environ. Sci. Technol., 47, 3840-3847, https://doi.org/10.1021/es3033132, 2013.

**Comment 11:** Paragraph starting on line 120 – You're assuming the relationship between harvested area and planted area from 1978 to 2017 is consistent decades and centuries before those data…a very dangerous assumption.

**Response**: Thank you for this suggestion. We agree that the relationship between cropland harvested area and the planted area is changed. Thus, we revise the reconstruction method. This step doesn't need to conduct in the new method. Please see the new method description in the response for Comment 10.

**Comment 12:** Line 123 – Yet another dataset, Borchers et al. 2014, was used to establish double-cropping at a regional level. Again, consistency among most of these datasets isn't great.

**Response**: Thank you for this suggestion. To reduce such kind of uncertainties, we revise the reconstruction method, and the subtracting work doesn't need to conduct. Please see the new method described in the response to Comment 10.

**Comment 13:** Line 124-125 – Another basic assumption that likely isn't true through time.

**Response**: Thank you for this suggestion. We revise the cropland reconstruction method, and we used the changing trend of HYDE cropland rather than national cropland per capita. Please see the new method described in the response to Comment 10.

**Comment 14:** Line 125-126 – Because the 1879 number was different you assumed it was incorrect? But data >1889 were "correct"? Was the reconstructed cropland area in 1879 substantially lower or higher than 1889?

**Response**: Thank you for this suggestion. On the one hand, there was no record for South Dakota in 1879. On the other hand, we assumed that there should not be a large difference between the cropland harvest area between 1879 and 1889. However, the cropland harvested area in South Dakota, Nebraska, and Kansas did have large differences in 1879 and 1889 (Figure R6). Thus, we thought the cropland harvested area in 1879 is not correct before. We rechecked the changes in cropland harvested area and total population (Figure R7) between 1879 and 1889, we think the cropland harvested area recorded by CAHA should be right and can be used for cropland reconstruction. So, we used the CAHA data in the revised version. The reconstructed cropland area in 1879 was not substantially lower or higher than 1889 (Figure R8).

[Figure]

**Figure R6**: Comparison between cropland harvested area in 1889 and 1879.

[Figure]

**Figure R7**: Changes of total population in Nebraska, Kansas, and South Dakota between 1850 and 1890.

[Figure]

**Figure R8**: Changes of national total cropland area during 1850-1910.

**Comment 15:** Line 130-131 – Another basic assumption that likely doesn't hold region to region.

**Response**: Thank you for this suggestion. We change the harmonization method for integrating HYDE and reconstructed historical cropland. Please see the new method described in the response to Comment 10.

**Comment 16:** Line 132-133 – Again…how did you account for differences in the HYDE data, and the (mostly) USDA-based data after 1889? Is there an obvious break in cropland amount pre- and post-1889?

**Response**: Thank you for this suggestion. For the historical cropland reconstruction, we use the HYDE cropland area trends rather than absolute value. The harmonization method between HYDE and CAHA or reconstructed cropland area can be found in the response to Comment 10.

As our response in Comment 14, we found that cropland harvested area increased fast during 1879-1889, and this increase resulted from rapid population increase. The reconstructed data also show that cropland area in South Dakota, Nebraska, Kansa increased rapidly during 1879-1889. But there is no obvious break in cropland amount pre and post-1889 (Figure R8).

**Comment 17:** Line 135-136 – One of the greatest difficulties in historical landscape reconstruction is the definition of "pasture", vs. "grassland", vs. "hay" vs. "rangeland", etc. There is no one definition that's universally accepted. Your definition here states Pasture includes areas "for the production of seed or hay crops". Many definitions of "cropland" include alfalfa, hay, and other crops in "planted area" or "cultivated crop" area. For "Pasture", you're introducing yet another completely new dataset to establish pasture area, NRI. Are the definitions of "pasture" for NRI the same for NLCD, HYDE, and the US Census of Ag?

**Response**: Thank you for this suggestion. We summarize the definitions of grazing land, pasture, and rangeland from different sources (Table R2).

As you said, the definitions 'pasture' from NRI is not same as that in HYDE and NLCD or US Census of Ag. Therefore, it is difficult to reconstruct historical pasture

area by harmonizing these datasets. We rewrite the pasture area reconstruction method, please see Lines 154-166.

In this study, we use the definition from the National Resource Inventory (NRI), in which pasture is the land that has a vegetation cover of grasses, legumes, and forbs, regardless of whether it is being grazed by livestock, planted for livestock grazing, or the production of seed or hay crops (Table 2). The NRI provides state-level pasture area with 5-year interval between 1982 and 2017, we set the pasture area as the baseline for historical reconstruction. Because there was no available pasture census data at the state level before 1982, the HYDE pasture was applied. We first estimated the pasture per capita by applying the trend of HYDE pasture per capita. Then, the total cropland area can be calculated by multiplying pasture per capita and total population. The data harmonization process can be expressed as follows:

$$HistPasture_{s,t} = (HistPasture\_p_{s,t+1} \times \frac{HYDE\_Pasture\_p_{s,t}}{HYDE\_Pasture\_p_{s,t+1}}) \times Pop_{s,t} \qquad (6)$$

where $HistPasture_{s,t}$ is the reconstructed pasture area of state $s$ in year $t$; $HistPasture\_p_{s,t+1}$ is pasture area per capita of state $s$ in year $t+1$; $HYDE\_Pasture\_p_{s,t}$ and $HYDE\_Pasture\_p_{s,t+1}$ are the HYDE pasture per capita of state $s$ in year $t$ and $t+1$.

**Comment 18:** Line 139 – Note Wasianen and Bliss took great pains to harmonize those definitional differences across their harmonized dataset.
**Response**: Thank you for this suggestion. We agree that they did make great efforts to harmonize those definitional differences across multisource dataset.

**Comment 19:** Section 2.2.3 – Again…it's extremely simplistic to assume things such as "pasture per capita" and that that ratio is consistent over time, and space.
**Response**: Thank you for this suggestion. We revise the historical pasture reconstruction method. We rewrite the pasture area reconstruction method, please see Lines 154-166. The method description can also be found in the response of Comment 17.

**Comment 20:** Section 2.2.4 – Definitions of what is "Forest" vary greatly among data sets. You're introducing yet another data set in FIA that may have a definition of

"forest" that differs from HYDE or from NLCD. How closely does the FATD data match with HYDE estimates, for example?

**Response**: Thank you for this suggestion. The definitional differences between FIA, NLCD, and LUH2 can be found in Table R3.

The HYDE data doesn't provide forest area estimation. So, the following figure shows the comparison between USDA-FR and NLCD, LUH2 between 2000-2020, between LUH2 and FATD in 1630. Both NLCD and LUH2 forest area are lower than USDA-FR. The forest area in Rocky Mountain states such as Nevada, Utah, New Mexico from NLCD and LUH2 is lower than that from USDA-FR. The forest area in 1630 derived from LUH2 and FATD does not match well.

[Figure]

**Figure R9**: a. Comparison between the average forest area (2000-2020) derived from USDA-FR and NLCD, LUH2. b. Comparison between the forest area derived from FATD and LUH2 in 1630.

**Comment 21:** Lines 156-157 – Yeah you lost me here with what you're trying to do, needs a better explanation.

**Response**: Thank you for the suggestions. We rewrite the forest reconstruction method. Please see Line 176-185.

**Comment 22:** Section 2.2.5 – See main comments above related to how you balanced the four LULC classes.

**Response**: Thank you for this suggestion.

**Comment 23:** Line 195 – What was used to establish the "land use change boundary"? That is, what was the source of "settled area" data"?

**Response**: Thank you for this suggestion. We assumed that in the area where human was not settled, there was no urban land, cropland, and pasture. The human settlement boundary data were used to constrain the probability of urban, cropland, and pasture. The Exploration and Settlement maps were made by the U.S. Dept. of the Interior, Geological Survey and can be accessed at https://maps.lib.utexas.edu/maps/histus.html#exploration.html. We assumed that the LULC would not be changed as the pre-colonial era, though there were Native Indiana people.

**Comment 24:** I don't mind the use of something like this to constrain the allocation of change, but do wonder about full-resolution results. Are there any hard border issues obvious in the data when change occurs at the edge of those defined boundary layers? Overall with the boundary and effect of population density, I appreciate you trying something other than assuming a static probability surface through time.

**Response**: Thank you for this suggestion. We check the fractional and Boolean type gridded data, there are hard border issues in some years which resulted from the county-level population application to modify the LULC probability surfaces. A simple method to generate the historical gridded population by combing the gridded population data with 1-km resolution and county-level population, which can be expressed as:

$$Pop_{i,t} = \frac{Pop_{i,t}^{county}}{Pop_{i,2000}^{county}} \times Pop_{i,2000}^{grid}$$

where, $Pop_{i,t}$ is improved population density at grid cell $i$ and year $t$; $Pop_{i,2000}^{county}$ and $Pop_{i,t}^{county}$ is the county-level population density at grid cell $i$ in 2000 and year $t$; $Pop_{i,2000}^{grid}$ is the gridded population density at grid cell $i$ in 2000 and year $t$.

[Figure]

**Figure R10**: Comparison between county-level and gridded population density data in 1850.

**Comment 25:** Section 2.3.2 – There needs to be more explanation here. You've basically summarized the entire actual allocation to the pixel level in one sentence. I certainly get that higher probability areas will likely have a higher proportion of a given LULC class, but it's all deterministic and it's all based solely on the probability surface? There's no stochasticity? With such a sparse description of methodology, it's also hard to see how this simple description of the methodology ends up with the aggregate totals from the allocation stage matching the quantitative estimates you established for each of the LULC classes.

**Response**: Thank you for this suggestion. We rewrite the description of spatial allocation strategies for generating fractional and Boolean type LULC data and make it as detailed as possible (section 2.3.2). We also add a random item when calculating the total transition probabilities (section 2.3.1). Please see the spatial allocation method described in the response to the general comments and Figure R4.

**Comment 26:** As noted in the main comments, I have other concerns about the allocation strategy.

**Response**: Thank you for this suggestion. We rewrite the spatial allocation strategy and make it as detailed as possible. Please see the spatial allocation method described in the response to the general comments and Figure R4.

**Comment 27:** Section 2.4 – Comparison to other LULC datasets isn't a validation, it's a consistency check. That's particularly true when every dataset has it's own

production methodologies, data sources, and thematic definitions, all of which makes even direct comparison problematic. Beyond that, you're comparing your results to some of the same datasets from which you parameterized your modeling, as noted in the overarching comments. Also note there aren't any details as to what methodologies you're actually using for "validation" in this very short, one-paragraph section.

**Response**: Thank you for this suggestion. We agree that the description or this step is a consistency check, rather than a data validation. Complete formal validation of model results was impractical because true, spatially explicit 'reference' data for historical LULC are difficult to obtain. In the historical LULC area reconstruction step, we assumed that the data used is reliable, which made it hard to conduct the validation. If we apply a rule to reconstruct the historical LULC area, we can use the census data to validate the accuracy of the prediction. For example, Sohl et al. (2016) used the LULC change data from the Trends project to reconstruct historical LULC proportions (demand) between 1938 and 1992, and compared the model results with census data, but such comparison still was a consistency check. Moreover, the definitional differences make it difficult to compare the newly developed dataset with other LULC products. We keep the LULC area comparison at the state level. Two new comparisons were conducted: Comparison between the newly developed cropland and USDA historical cropland area at the county level; Comparison between our reconstruction and NLCD developed land, cropland, pasture, and forest. Please see section 3.1.2 and 3.1.3 (Lines 304-338).

**Comment 28:** Section 3.1 – As noted previously, this isn't very useful for inferring confidence in your results, when you're using the same datasets to parameterize the model as you are to "validate" model results.

**Response**: Thank you for this suggestion. As our response in Comment 27, it is hard to validate the newly developed LULC data. We keep the LULC area comparison at the state level. Another two data comparisons were conducted: comparison between the newly developed cropland and USDA historical crop area at the county level; comparison between our reconstruction and NLCD developed land, cropland, pasture, and forest. Please see section 3.1.2 and 3.1.3 (Lines 304-338).

**Comment 29:** Line 273 (and throughout the results section) – If you're going to refer to a specific driving force of change, and, for example, point to a specific policy, you should name the policy and reference it (Immigration and Naturalization Act of 1965). While it certainly did change the nature of immigration to the country, you do give it too much focus as "the" causes of urban land increases after 1965. There's a lot more at play there than immigration policy.

**Response**: Thank you for this suggestion. We agree that urban land expansion is driven by multiple factors, while it is largely determined by population and economy growth. We rewrite the related sentences about urban land expansion. Please see Lines 353-359.

**Comment 30:** Lines 276-277 – You state "cropland area did not change significantly" from 1930 to present day. First, it's always problematic to use the term "significantly" in a journal paper, given the scientific meaning of the word. Secondly, I would argue there were "substantial" trends in agriculture after 1930, including some of those you mention (e.g., biofuel impacts).

**Response**: Thank you for this specific suggestion. We rewrite the related sentences, please see Line 360-361. What I want to express is that the change magnitude of the national total cropland area is not like the period of 1850-1920 (Figure R11). In fact, cropland was abandoned in the southeast US and expanded in the Great Plains.

[Figure]

**Figure R11**: Changes of national total cropland area derived from the newly developed LULC dataset during 1630-2020.

**Comment 31:** Figure 7 – On a national-scale map figure, it's difficult to see patterns of the individual land use transitions. Perhaps it would be augmented by a

complementary confusion matrix of changes or some other tabular data approach that
allows you to see (and easily quantify) transition types.

**Response**: Thank you for this suggestion. We add a table including the information of
major LULC conversions during 1630–1850, 1850–1920, 1920–2020, and 1630–
2020. Please see Table 3.

**Comment 32:** Section 3.4 – This isn't the most effective section to me. As noted in
the main comments, a major premise of the paper was to provide a "high resolution"
historical landscape reconstruction for the US. Much of the "regional" information
here is also discussed in the overall results above. I'd have much rather seen some real
examples (and preferably validation) of landscape pattern at finer scales, given the
focus on higher resolution with this paper.

**Response**: Thank you for this suggestion. Considering the differences in natural
environmental conditions and social-economic development, land use and land cover
change showed spatial heterogeneity in the CONUS during 1630–2020. The purpose
of section 3.4 is to give a general description of how LULC changes among regions.

To show the improvement of newly developed LULC data, we add several figures to
show LULC changes at a fine scale in the discussion section, please see Figure R12,
13, and 14.

[Figure]

**Figure R12:** Visual comparison between our cropland data and the History Database of Global Environment (HYDE), Yu and Lu (2017) cropland density (YLmap), and Zumkehr and Campbell (2013) historical fractional cropland areas (ZCmap) in four different sites (a-d). The locations of image center points are as follows: a. Ohio (83.05 °W, 40.17 °N), b. Georgia (83.58 °W, 32.77 °N), c. Arkansas (90.56 °W, 34.76 °N), d. Texas (100.92 °W, 32.81°N).

[Figure]

**Figure R13:** Visual comparison of our pasture data with History Database of Global Environment (HYDE), and Land Use Harmonization (LUH2) in four different sites (a-d). The locations of image center points are as follows: a. Iowa (93.64 °W, 42.03 °N), Virginia (78.72 °W, 37.96 °N), c. Illinois (90.07 °W, 38.68 °N), d. Arkansas (92.56 °W, 34.97 °N).

[Figure]

**Figure R14:** Visual comparison between our forest data and Land Use Harmonization (LUH2) in four different sites (a-d). The locations of image center points are as follows: a. Colorado (106.47 °W, 38.97 °N), Wisconsin (89.85 °W, 44.54 °N), c. Alabama (86.72 °W, 33.33 °N), d. New York (75.14 °W, 42.21 °N).

**References**:

Goldewijk, K. K., Beusen, A., Doelman, J., and Stehfest, E.: Anthropogenic land use estimates for the Holocene - HYDE 3.2, Earth Syst. Sci. Data, 9, 927-953, https://doi.org/10.5194/essd-9-927-2017, 2017.

Hurtt, G. C., Chini, L., Sahajpal, R., Frolking, S., Bodirsky, B. L., Calvin, K., Doelman, J. C., Fisk, J., Fujimori, S., Goldewijk, K. K., Hasegawa, T., Havlik, P., Heinimann, A., Humpenoder, F., Jungclaus, J., Kaplan, J. O., Kennedy, J., Krisztin, T., Lawrence, D., Lawrence, P., Ma, L., Mertz, O., Pongratz, J., Popp, A., Poulter, B., Riahi, K., Shevliakova, E., Stehfest, E., Thornton, P., Tubiello, F. N., van Vuuren, D. P., and Zhang, X.: Harmonization of global land use change and management for the period 850-2100 (LUH2) for CMIP6, Geosci. Model Dev., 13, 5425-5464, https://doi.org/10.5194/gmd-13-5425-2020, 2020.

Yu, Z. and Lu, C.: Historical cropland of the continental U.S. from 1850 to 2016, PANGAEA, https://doi.org/10.1594/PANGAEA.881801, 2017.

Zumkehr, A. and Campbell, J. E.: Historical U.S. Cropland areas and the potential for bioenergy production on abandoned croplands, Environ. Sci. Technol., 47, 3840-3847, https://doi.org/10.1021/es3033132, 2013.

**Comment 33:** Lines 338-339 – Agreed about the "per capita" approach.

**Response**: Thank you for this suggestion. We improve the urban land estimation method by using a changing urban land per capita. The HISDAC data is applied between 1810 and 2001 to reduce the bias in our estimation. The new method description can be found in the response to Comment 9.

**Comment 34:** Lines 339-341 – This doesn't serve as any kind of adequate validation or even consistency check between datasets. Showing a national-scale map and stating the patterns are "consistent" isn't valuable, and is very subjective at that scale.

**Response**: Thank you for this suggestion. The data validation/comparison or consistency check has been conducted in section 3.1 by comparing with NLCD data, agriculture census data, and state-level LULC area.

The national scale maps can give an overview of the spatial pattern of LULC in 1630, 1850, 1920, and 2010. We keep the national-scale map and add extra comparison figures at the fine scale to show the improvement in the newly developed LULC dataset. Please see Figure R12, R13, and R14.

**Comment 35:** Line 357-358 – Exactly why it's not very valuable to compare your model results to HYDE…those data were used to help establish the model parameters themselves.

**Response**: Thank you for this suggestion. The HYDE data was used to reconstruct historical cropland and pasture area. But not all the periods applied the HYDE data. Moreover, we used the trend of HYDE cropland/pasture per capita rather than the absolute value of LULC area. Thus, we compared the reconstructed historical LULC area and spatial pattern with HYDE.

**Comment 36:** Line 362 – Your product has higher spatial resolution than something like HYDE, but there's no quantitative analysis of that spatial pattern that proves the superior value of that higher native resolution.

**Response**: Thank you for this suggestion. We add regional scale figures to show the data improvement than the dataset with coarse resolution (Figure R12, R13, and R14). Moreover, the HYDE or LUH2 have higher cropland acreage compared to US-specific datasets, like Yu and Lu, and USDA census data. We fixed this problem and went back to 390 years ago.

**Comment 37:** Figure 9 – It is difficult to compare all of these datasets given the definitional differences between them, particularly for pasture and cropland.

**Response**: Thank you for this suggestion. We agree that it is hard to say which data is more accurate or reliable because of the definitional differences among them. But we can know whether the area of the reconstructed historical LULC dataset is in a reasonable range through data comparison. Meanwhile, the previous spatial LULC datasets are also a good reference to judge whether the reconstructed data has a reasonable spatial pattern.

**Comment 38:** Lines 382-383 – I'm not sure it's more "reliable", as sample-based, inventory approaches have flaws, just as satellite-based approaches have flaws. The bigger concern to me are the definitional differences, not the methodological differences.

**Response**: Thank you for this suggestion. We rewrite this sentence. The word 'reliable' may not be suitable to describe the FIA data, I would say it has better

consistency than other forest data for long-term study. But it is hard to say which forest data is more reliable because of the differences between definitions.

NLCD and Sohl et al. (2016) data define forest as the areas dominated by trees generally greater than 5 meters tall and greater than 20% of total vegetation cover, higher than that in our forest definition (forest cover greater than 10%). Thus, the forest area in this study was higher than the NLCD and Sohl et al. (2016) data.

In LUH2, the biomass density (BD) map is used to identify the potential forest (BD > 2 kg C m$^{-2}$) and non-forest at $0.25 \times 0.25$-degree resolution (Hurtt et al., 2020), which underestimates the forest in Rock Mountain and Northwest. NLCD is produced by using Landsat images and a comprehensive method and provides nationwide data on land use and land cover change at a 30 m resolution (Homer et al., 2020). Spatially, it can capture the forest distribution better than LUH2. The FIA data provides critical status and trend information through a system of annual resource inventory that covers both public and private forest lands across the United States (https://www.fs.usda.gov/research/inventory/FIA), and it can provide forest trend data back to 1630.

**References**:

Homer, C., Dewitz, J., Jin, S., Xian, G., Costello, C., Danielson, P., Gass, L., Funk, M., Wickham, J., Stehman, S., Auch, R., and Riitters, K.: Conterminous United States land cover change patterns 2001-2016 from the 2016 National Land Cover Database, ISPRS. J. Photogramm. Remote Sens., 162, 184-199, https://doi.org/10.1016/j.isprsjprs.2020.02.019, 2020.

Hurtt, G. C., Chini, L., Sahajpal, R., Frolking, S., Bodirsky, B. L., Calvin, K., Doelman, J. C., Fisk, J., Fujimori, S., Goldewijk, K. K., Hasegawa, T., Havlik, P., Heinimann, A., Humpenoder, F., Jungclaus, J., Kaplan, J. O., Kennedy, J., Krisztin, T., Lawrence, D., Lawrence, P., Ma, L., Mertz, O., Pongratz, J., Popp, A., Poulter, B., Riahi, K., Shevliakova, E., Stehfest, E., Thornton, P., Tubiello, F. N., van Vuuren, D. P., and Zhang, X.: Harmonization of global land use change and management for the period 850-2100 (LUH2) for CMIP6, Geosci. Model Dev., 13, 5425-5464, https://doi.org/10.5194/gmd-13-5425-2020, 2020.

Sohl, T., Reker, R., Bouchard, M., Sayler, K., Dornbierer, J., Wika, S., Quenzer, R., and Friesz, A.: Modeled historical land use and land cover for the conterminous United

States, J. Land Use Sci., 11, 476-499, https://doi.org/10.1080/1747423X.2016.1147619, 2016.

**Comment 39:** Section 4.2 doesn't add a lot to the paper for me, particularly since you've already tried to explain some driving forces in the previous paragraphs of the paper. I'd much rather have the drivers woven into the story of what's happening in your results, than as a separate section.

**Response**: Thank you for this suggestion. The driving forces of land use and land cover change are quite complex in the United States. Though some driving forces of LULC change have been mentioned in the *Results* part, we think a comprehensive analysis of the driving forces of LULC change is still needed. We reorganize this paragraph and add more discussions. Please see Section 4.2 (Lines 510-533).

**Comment 40:** Lines 435-436 – I think reconstruction of historical land use is limited more by reliable, consistent historical data than methodology. Machine learning methods aren't going to be that valuable for historical reconstruction given the paucity and inconsistency of historical data for training.

**Response**: Thank you for this suggestion. We agree that the most important step in the historical LULC reconstruction study is to collect reliable and consistent data. The spatial allocation algorithm also impacts the reconstructed landscape pattern.

**Comment 41:** Section 4.3 – Somewhere in here you absolutely need to highlight the difficulties with trying to harmonize data sets with different definitions, data sources, and methodologies.

**Response**: Thank you for this suggestion. We rewrite the section 4.3. In this section, we add the discussion about the difficulties of harmonizing data sets with different definitions, data sources, and methodologies. Please see Section 4.3 (Lines 535-567).

---

## Author Comment (AC5)

Thank you for the comments and suggestions. These comments were very helpful for revising and improving our paper. We have responded to the comments point by point.

**Reviewer #2:**

**General comment:**

The manuscript reconstructed land use and land cover history during 1630-2020 over the conterminous United States (CONUS). It gathered multiple sources of data including remote sensing-based land cover maps, national inventories and statistics data, meteorological fields, topographical data and others. The resulting dataset has an improved spatial resolution than other global datasets, potentially facilitating regional modeling work. However, the manuscript is not well organized. For example, in the method section, there are missing details about how various sources of input data were combined and adjusted to generate full time series of urban and crop land area. And also, the validation needs more justification on the choice of spatial scale and more discussion about what causes differences between the data from this work and other datasets. For example, this data shows a different trend in areas of urban and pasture in the past two decades compared to others. I have provided detailed comments below.

**Response**: Thank you for these suggestions. We revise the method for reconstructing historical urban and cropland area with more detail information, including the data used for difference period and harmonization method between different datasets. Please see the section 2.2 (Lines 105-185).

*Section 2.2.1*

In this study, we used the same definition for the developed land as NLCD for urban land. The developed land in NLCD includes four components: open space, low intensity developed land, medium intensity developed land, and high intensity developed land (Table 2). We used the NLCD developed land area during 2001–2019 as the urban land area baseline. Before 2001, we applied Historical Settlement Data Compilation for the United States (HISDAC-US) (Leyk et al., 2020; Uhl et al., 2021) as input to reconstruct the historical urban land area. The HISDAC-US built-up areas describes the built environment for most of the CONUS from 1810 to 2015 at 5-year temporal and 250 m

spatial resolution using built-up property records, locations, and intensity data (Leyk and Uhl, 2018; Uhl et al., 2021). Here, we assumed that the HISDAC built-up areas data could capture the trend of urban land development. Then, the historical urban land can be estimated as follows:

[revised manuscript text omitted]

For the data validation/comparison, we add the causes analysis for the differences between the datasets, please see section 3.1.1 (Lines 279-303). Considering the importance of cropland and the census data availability, we add the county level cropland area comparison between our data and USDA census data, please see section

3.1.2 (Lines 304-322). Moreover, we also compare the NLCD and our data at grid level, please see section 3.1.3 (Lines 323-338).

**References:**

Bigelow, D. P. and Borchers, A.: Major Uses of Land in the United States 2012, U.S. Department of Agriculture, Economic Research Service, 2017.

Goldewijk, K. K., Beusen, A., Doelman, J., and Stehfest, E.: Anthropogenic land use estimates for the Holocene - HYDE 3.2, Earth Syst. Sci. Data, 9, 927-953, https://doi.org/10.5194/essd-9-927-2017, 2017.

Homer, C., Dewitz, J., Jin, S., Xian, G., Costello, C., Danielson, P., Gass, L., Funk, M., Wickham, J., Stehman, S., Auch, R., and Riitters, K.: Conterminous United States land cover change patterns 2001-2016 from the 2016 National Land Cover Database, ISPRS. J. Photogramm. Remote Sens., 162, 184-199, https://doi.org/10.1016/j.isprsjprs.2020.02.019, 2020.

Leyk, S. and Uhl, J. H.: HISDAC-US, historical settlement data compilation for the conterminous United States over 200 years, Sci. Data, 5, 1-14, https://doi.org/10.1038/sdata.2018.175, 2018.

Leyk, S., Uhl, J. H., Connor, D. S., Braswell, A. E., Mietkiewicz, N., Balch, J. K., and Gutmann, M.: Two centuries of settlement and urban development in the United States, Sci. Adv., 6, https://doi.org/10.1126/sciadv.aba2937, 2020.

Liu, M. and Tian, H.: China's land cover and land use change from 1700 to 2005: Estimations from high-resolution satellite data and historical archives, Global Biogeochem. Cy., 24, GB3003, https://doi.org/10.1029/2009gb003687, 2010.

Uhl, J. H., Leyk, S., McShane, C. M., Braswell, A. E., Connor, D. S., and Balk, D.: Fine-grained, spatiotemporal datasets measuring 200 years of land development in the United States, Earth Syst. Sci. Data, 13, 119-153, https://doi.org/10.5194/essd-2020-217, 2021.

Zumkehr, A. and Campbell, J. E.: Historical U.S. Cropland areas and the potential for bioenergy production on abandoned croplands, Environ. Sci. Technol., 47, 3840-3847, https://doi.org/10.1021/es3033132, 2013.

**Specific comments**

**Comment 1:** P2, L55, LUH2 has a spatial resolution of 0.25-deg.

**Response**: Thank you for this suggestion. We revise it. Please see Line 60.

**Comment 2:** P2, l55, please provide references to the 'substantial uncertainties' statement.

**Response**: Thank you for this suggestion. We revise the word of "**substantial uncertainties**", and add references describing the spatial uncertainties of LULC data with coarse resolution.


Figure R1 shows the urban land per capita change derived from the newly developed urban land and HISDAC built-up areas during 1810–2010. The figure shows that the urban land per capita gradually increase over the past 200 years, which means that our previous data overestimated the urban land area in the early years.

[Figure]

**Figure R1**. Urban land per capita change between 1810 and 2020. The value indicated by the orange dot is the urban land per capita derived from HISDAC built-up areas; the value indicated by the black line is the urban land per capita derived from the newly developed urban land.

**Comment 6:** P6, L106-108, which dataset were these criteria applied to? CPHR, CAHA, or other?

**Response**: Thank you for this suggestion. The cropland harvested area from CAHA is used for historical cropland reconstruction.

**Comment 7:** P6, L112, I could not find the CPHR dataset in Table 1. Please make sure the product name and time period is consistent between the table and manuscript.

**Response**: Thank you for this suggestion. The cropland planted area in Table 1 is the CPHR data. The cropland planted area was used to establish the relationship with cropland harvested area.

**Referee 1** thinks that converting cropland harvested area with an unchanged linear equation is not reasonable and dangerous. So, we change the cropland reconstruction method, and the cropland planted area is not used in the new method. Please see section 2.2.2 (Lines 124-153).

**Comment 8:** P6, L117, 1975-2020, right? If not, please list this dataset clearly in Table 1. Please add more details about how the adjustment was done to make 'the inter-annual variation more reasonable'. For example, was this adjustment done at national scale then disaggregated to state scale?

**Response**: Thank you for this suggestion. The period of cropland planted area used before is 1975-2020. In the new method, the cropland planted area data is not used. In the revised method, we also conduct an adjustment to subtract the double-cropped area, and the national cropland harvested area was disaggregated to state-scale. Please see Lines 132-140.

**Comment 9:** P6, L125-126, what causes this difference? Due to the CAHA crop harvested area?

**Response**: Thank you for this suggestion. We assumed that there should be a large difference between the cropland harvest area between 1879 and 1889. However, we found that the cropland harvested area in South Dakota, Nebraska, and Kansas increased rapidly between 1879 and 1889 (Figure R2). Thus, we thought the cropland harvested area in 1879 is not correct before.

We rechecked the changes in cropland harvested area between 1879 and 1889 and total population (Figure R3) during 1850-1890. The rapidly increase of cropland area resulted from the population growth. we think the cropland harvested area should be right and can be used for cropland reconstruction.

[Figure]

**Figure R2**: Comparison between cropland harvested area in 1889 and 1879

[Figure]

**Figure R3**: Total population change in Nebraska, Kansas, and South Dakota between 1850 and 1890.

**Comment 10:** P6, L127, please justify why the cropland per capita was calculated at national level instead of state level like the urban area per capita. This could avoid any potential confusions about the following assumption you made.

**Response**: Thank you for this suggestion. Considering that HYDE is reconstructed based on the country level data, so the national level cropland per capita was used before. We agree that applying the state level cropland area per capita to estimate the total cropland area should be more reasonable. In the new method, the HYDE cropland per capita for each state was used in the harmonization process. Please see Lines 145-152.

**Comment 11:** P7, L130-134, was there a harmonization process applied to connect cropland area during 1630-1880 derived from the HYDE and cropland area during 1889-2020 derived from the CAHA? If not, please add time-series plots of derived cropland areas that show such harmonization is not needed.

**Response**: Thank you for this suggestion. There was a harmonization process to connect cropland area derived from HYDE and reconstructed historical cropland area. Please see Lines 145-152.

Because there was no available cropland census data at the state level before 1879, the HYDE cropland was used. We first estimated the cropland per capita by applying the trend of HYDE cropland per capita. Then, the total cropland area can be calculated by multiplying cropland per capita and total population. The data harmonization process can be expressed as follows:

$$HistCrop_{s,t} = (HistCrop\_p_{s,t+1} \times \frac{HYDE\_Crop\_p_{s,t}}{HYDE\_Crop\_p_{s,t+1}}) \times Pop_{s,t} \qquad (5)$$

where $HistCrop_{s,t}$ is the reconstructed cropland area of state $s$ in year t; $HistCrop\_p_{s,t+1}$ is the reconstructed cropland per capita of state s in year t+1; $HYDE\_Crop\_p_{s,t}$ and $HYDE\_Crop\_p_{s,t+1}$ are HYDE cropland per capita of state $s$ in year $t$ and $t$+1.

**Comment 12:** P7, L141-L143, please clarify how the pasture per capita during 1630–1982 was estimated from pasture per capita at the state level (NRI data-based) in 1982 and the national level (HYDE data-based).

**Response**: Thank you for the suggestions. Same as the cropland area estimation, we agree that applying the state level pasture per capita to estimate the total pasture area should be more reasonable. In the new method, the HYDE cropland per capita for each state was used in the harmonization process. Please see Lines 154-166.

Because there was no available pasture census data at the state level before 1982, the HYDE pasture was applied. We first estimated the pasture per capita by applying the trend of HYDE pasture per capita. Then, the total cropland area can be calculated by multiplying pasture per capita and total population. The data harmonization process can be expressed as follows:

$$HistPasture_{s,t} = (HistPasture\_p_{s,t+1} \times \frac{HYDE\_Pasture\_p_{s,t}}{HYDE\_Pasture\_p_{s,t+1}}) \times Pop_{s,t} \qquad (6)$$

where $HistPasture_{s,t}$ is the reconstructed pasture area of state $s$ in year $t$; $HistPasture\_p_{s,t+1}$ is pasture area per capita of state $s$ in year $t+1$; $HYDE\_Pasture\_p_{s,t}$ and $HYDE\_Pasture\_p_{s,t+1}$ are the HYDE pasture per capita of state $s$ in year $t$ and $t+1$.

**Comment 13:** P7, L157, "... then multiplied the forest area from USDA-FR to generate …". What is the time period of USDA-FR forest area used here? 1630?

**Response**: Thank you for this suggestion. We revise it. …then multiplied the forest area from USDA-FR in 1630.

**Comment 14:** P8, L170, why is there no scaling factor for TA_t_r (s) < TLA(s) ? How was the difference/residual between TA_t_r (s) and TLA(s) dealt in your following analysis? And also, what is the source of the state's total land area (TLA)?

**Response**: Thank you for this suggestion. The purpose of this section is to check whether the total area of urban, cropland, pasture, and forest is larger than the state land area. If the area is lower than the state land area, we will not modify the reconstructed area of each LULC class. Thus, there is no scaling factor for TA_t_r (s) < TLA(s). We don't do consider the difference/residual between TA_t_r (s) and TLA(s) dealt in the following analysis. In the updated version, we check the reconstructed area of each LULC class, the total area of urban, cropland, pasture, and forest at each state and each year stratify the condition "TA_t_r (s) < TLA(s)", so we remove this section.

For the state's total land area, we derive the area information from state boundary file. The file can be download in the following links:

https://www.nass.usda.gov/Publications/AgCensus/2017/index.php#highlights

**Comment 15:** P9, L182-184, please add references to how ANN can be used to solve the nonlinear geographical problems.

**Response**: Thank you for this suggestion. We add related references. Please see Lines 212-213.

**Comment 16:** P9, L184-185, please add more details about how land use probability was generated from ANN and NLCD. And also, NLCD is supposed to be an

independent variable used in the modeling, right? However, NLCD is a land cover data, how can it be useful for modeling of land use probability? Please keep in mind that land use and land cover are usually used interchangeably but are actually different terms. If you regard them as the same terms, why do you refer to it as land use probability instead of land use and land cover probability?

**Response**: Thank you for this suggestion. In this study, we used the ANN tool in Future Land Use Simulation (FLUS) software to estimate the probability of occurrence. The FLUS model is a land-use change simulation model that combines Cell Automata (CA) model and artificial neural networks (ANN) (Liu et al., 2017). The independent variables for the ANN model include elevation, slope, annual mean temperature, annual precipitation, annual maximum temperature (July), annual minimum temperature (January), crop productivity index, population density, distance to the city, distance to the road, distance to the railway, distance to the river, soil organic carbon, soil sand, soil clay. The purpose of ANN training is to establish the relationship between each LUCC type and the independent variables, and the NLCD Boolean type will be the dependent variable.

We agree that land use and land cover are usually used interchangeably but are different terms. In this study, we regard them as the same terms change the 'land use probability' as 'LULC probability'.

**References**:

Liu, X., Liang, X., Li, X., Xu, X., Ou, J., Chen, Y., Li, S., Wang, S., and Pei, F.: A future land use simulation model (FLUS) for simulating multiple land use scenarios by coupling human and natural effects, Landsc. Urban Plan., 168, 94-116, https://doi.org/10.1016/j.landurbplan.2017.09.019, 2017.

**Comment 17:** P9, L200, what are the difference between ES_weight_t and SE_weight_t in Eq. 4 and 6.

**Response**: Thank you for this suggestion. We revise it. Eq.4 should be SE_weight_t.

**Comment 18:** P9, L200, what are the values of t1 and t0 in Eq. 7? Are they the starting and ending year of each subperiod?

**Response**: Thank you for this suggestion. Yes, t0 and t1 are the starting and ending year of each subperiod.

**Comment 19:** P10, L216, what does 'Boolean type' mean? Categorical type like NLCD that each grid (i.e, 30 m) has a single land use and land cover type?
**Response**: Thank you for this suggestion. Yes, categorical type.

**Comment 20:** P10, L217-218, please elaborate more on "the total number of potential pixels or the land use demand was determined based on the reconstruction results in Section 2.2. Then, the area difference of land use type k between the target and current map was calculated." The resulting data from section 2.2 is the state level total area of urban, crop, pasture and forest, right? So, such 'area difference' is at the state level, right?
**Response**: Thank you for this suggestion. We implement the spatial allocation algorithm at state level, so the resulting data from section 2.2 is the state level total area of urban, crop, pasture and forest and the 'area difference' is also at the state level. We rewrite the spatial allocation process for both fractional LULC type and category type, please see Lines 235-266.

**Comment 21:** P10, L228-236, the LUH2 used for validation, but it was not mentioned in section 2.1 and Table 1.
**Response**: Thank you for this suggestion. We revised the material and methods section and add the description of LUH2. Please see Table A2.

**Comment 22:** P10 L229, please justify why the validation against NLCD was at state-level. As you highlighted that your land use data is at 1 km, a finer spatial resolution than most other data, so the validation at 1 km will be more informative about the value of this dataset. A good agreement on state-level total land use area does not necessarily indicate the spatial allocation of total area to 1 km is good as well.
**Response**: Thank you for this suggestion. The lack of actual spatial explicit reference data made a complete formal validation impractical. Though the LULC definitions in this study are different from other LULC datasets, data comparison is a way to access the accuracy of the reconstructed LULC area and spatial pattern. Thus, we conducted

[revised manuscript text omitted]

**Comment 23:** P11, Figure 4. Such comparisons to different datasets are very important and informative. However, I would suggest more discussion on the 'outlier' states in figure 4a and 4b. For example, please dig into which states your estimates of urban land area is lower than HISDAC? And what are the potential causes?

**Response**: Thank you for this suggestion. We revised this section and added more analysis about the differences between multiple datasets, please see Lines 279-296.

**Comment 24:** P14, L290-L300, is the transition in figure 7 the gross transition or net transition? Please clarify this somewhere because their magnitude and impacts on climate and carbon modeling are quite different. For example, 30% deforestation and 30% reforestation have zero transition on forest, but the biophysical effects could not be ignored.

**Response**: Thank you for this suggestion. We agree that the effects of LULC gross transition and net transition are quite different. Figure 7 shows the LULC net transition, that is the LULC difference in 1630, 1850, 1920 and 2020.

**Comment 25:** P17, Figure 9d, the black line stops at near 1920.

**Response**: Thank you for this suggestion. We revise the Figure 9.

[Figure]

**Revised Figure 9**: Comparison with other datasets for the conterminous United States: urban land (a); cropland (b); pasture (c); forest (d). NLCD: National Land Cover Database; HYDE: History Database of the Global Environment; HISDAC: Historical Settlement Data Compilation; ERS: Economic Research Service; YLmap: Yu and Lu (2018) cropland density; ZCmap: Zumkehr and Campbell (2013) historical fractional cropland areas; LUH2: Land Use Harmonization; FATD: Forest Area Trend Data; USDA-FR: USDA Forest Resources of the United States of 2017.

**Response**: Thank you for this suggestion. We estimated the historical urban land area by multiplying the urban land per capita (average between 2001 and 2016) and state total population. The area difference between NLCD and our estimation was induced by the increase rate difference between urban land and total population. In the new method, the NLCD developed land area during 2001-2019 was set as the baseline. Considering the possible underestimation as you said in Comment 5, we revise the urban land estimation method by applying the HISDAC built-up area between 1810 and 2001. The new estimation method can be found in the response of Comment 5.

For the pasture, we compared the state level NRI and NLCD data. The definitions of pasture in NRI and NLCD are different. In NLCD, pasture/hay areas is land of grasses, legumes, or grass-legume mixtures planted for livestock grazing or the production of seed or hay crops, typically on a perennial cycle. But pasture includes land that has a vegetative cover of grasses, legumes, and/or forbs, regardless of whether it is being grazed by livestock in NRI.

**Comment 27:** P20, L382-383, please explain why inventory-based data is more reliable than the satellite-based forest (NLCD) and biomass density-based forest (LUH2). Forests could have different definitions by various sources, and forest areas are subject to the definition, thus I could see which definition/data source is more reliable than others.

**Response**: Thank you for this suggestion. Exactly, it's hard to say which forest data is more reliable because of the differences between forest definitions (Please see Table R1). In LUH2, the biomass density (BD) map is used to identify the potential forest

(BD > 2 kg C m$^{-2}$) and non-forest at 0.25° × 0.25° resolution (Hurtt et al., 2020), which underestimates the forest in the Rocky Mountain region and Northwest. NLCD is produced by using Landsat images and a comprehensive method and provides nationwide data on land cover and land cover change at a 30 m resolution. So, it can provide a more accurate spatial pattern than the LUH2. The FIA data provides critical status and trend information through a system of annual resource inventory that covers both public and private forest lands across the United States (https://www.fs.usda.gov/research/inventory/FIA), and it can provide a forest trend data back to 1630. Thus, we think that the forest area from FIA is more credible and consistent for long-term LULC change study.

**Table R1.** Forest definitions from different data sources.

| Data source | Definition |
| --- | --- |
| FIA | Land at least 10 percent stocked by forest trees of any size, or formerly having such tree cover, and not currently developed for non-forest uses, with a minimum area classification of 1 acre (https://www.fia.fs.fed.us/tools-data/maps/2007/descr/yfor_land.php). |
| NLCD | Areas dominated by trees generally greater than 5 meters tall, and greater than 20% of total vegetation cover (https://www.mrlc.gov/data/legends/national-land-cover-database-class-legend-and-description). |
| LUH2 | Forest was defined using a single tree canopy cover threshold to match the global forest extent provided by the FAO FRA report (Hurtt et al., 2020). |

**References**:

Hurtt, G. C., Chini, L., Sahajpal, R., Frolking, S., Bodirsky, B. L., Calvin, K., Doelman, J. C., Fisk, J., Fujimori, S., Goldewijk, K. K., Hasegawa, T., Havlik, P., Heinimann, A., Humpenoder, F., Jungclaus, J., Kaplan, J. O., Kennedy, J., Krisztin, T., Lawrence, D., Lawrence, P., Ma, L., Mertz, O., Pongratz, J., Popp, A., Poulter, B., Riahi, K., Shevliakova, E., Stehfest, E., Thornton, P., Tubiello, F. N., van Vuuren, D. P., and Zhang, X.: Harmonization of global land use change and management for the period 850-2100 (LUH2) for CMIP6, Geosci. Model Dev., 13, 5425-5464, https://doi.org/10.5194/gmd-13-5425-2020, 2020.

---

## Author Comment (AC6)

Thank you for the comments and suggestions. These comments were very helpful for revising and improving our paper. We have responded to the comments point by point.

**Reviewer #3:**

**General comments:**

This manuscript describes the development and details of a land-use dataset for the United States for the years 1630-2020. The dataset differs from other land-use datasets in that it is a high-resolution product, for almost 400 years of the historical period. The dataset combines multiple different input datasets, for different time-periods, and in different formats/spatial resolutions and reconstructs the historical areas of cropland, pasture, urban land, and forests annually at 1km x 1km spatial resolution, for the CONUS. The results show expansion of cropland and urban areas, with associated losses of natural vegetation. Comparison with other datasets show many areas of qualitative agreement, with some interesting differences for some time periods and land-use types.

Overall, the manuscript is mostly well-written and organized. It includes some useful information about the dataset development process, and an analysis of the resulting products. The dataset will be useful to modelers working in the areas of climate and ecosystems to better understand the high-resolution impacts of LCLUC in the CONUS over a long historical period. A few areas for improvement include:

**Specific comments:**

**Comment 1**: Although other alternative datasets are mentioned and compared with the new dataset, it would be helpful to know what advantages those other datasets might have (if any) over the new dataset (e.g. for HYDE an even longer time period is used, and for some datasets there could be additional data layers beyond the ones provided in this dataset, etc).

**Response**: Thank you for this suggestion. We add the related discussion in section 4.3, please see Lines 558-567.

The newly developed LULC dataset reconstructed the LULC history with more LULC types than ZCmap and YLmap and has higher spatial resolution than HYDE and LUH2. Our LULC data emphasizes the accuracy of area change resulting from LULC conversion rather than the changes in LULC structure or attributes. For example, forest

management (e.g., wood harvest and thinning) results in the forest cover decreases and ecosystem function change, but the LULC type is unchanged. HYDE and LUH2 not only have a more extended cover period, but also provide more sub-types and LULC attributes. HYDE classified cropland into rain-fed rice, irrigated rice, rain-fed other crops, and irrigated other crops (Goldewijk et al., 2017). LUH2 divides cropland into C3 crops and C4 crops and includes the wood harvest (traditional fuelwood, commercial biofuels, and industrial roundwood) and primary/secondary forest age (Hurtt et al., 2020). In the future, the LULC sub-types (e.g., tree species, crop types) and attributes (e.g., forest age, management intensity) through collecting from agricultural census data and forest inventory data can be incorporated into our dataset (Thompson et al., 2013; Chen et al., 2017; Crossley et al., 2021).

**References**:

Chen, G., Pan, S., Hayes, D. J., and Tian, H.: Spatial and temporal patterns of plantation forests in the United States since the 1930s: an annual and gridded data set for regional Earth system modelling, Earth Syst. Sci. Data, 9, 545-556, https://doi.org/10.5194/essd-9-545-2017, 2017.

Crossley, M. S., Burke, K. D., Schoville, S. D., and Radeloff, V. C.: Recent collapse of crop belts and declining diversity of US agriculture since 1840, Glob. Change Biol., 27, 151-164, https://doi.org/10.1111/gcb.15396, 2021.

Goldewijk, K. K., Beusen, A., Doelman, J., and Stehfest, E.: Anthropogenic land use estimates for the Holocene - HYDE 3.2, Earth Syst. Sci. Data, 9, 927-953, https://doi.org/10.5194/essd-9-927-2017, 2017.

Hurtt, G. C., Chini, L., Sahajpal, R., Frolking, S., Bodirsky, B. L., Calvin, K., Doelman, J. C., Fisk, J., Fujimori, S., Goldewijk, K. K., Hasegawa, T., Havlik, P., Heinimann, A., Humpenoder, F., Jungclaus, J., Kaplan, J. O., Kennedy, J., Krisztin, T., Lawrence, D., Lawrence, P., Ma, L., Mertz, O., Pongratz, J., Popp, A., Poulter, B., Riahi, K., Shevliakova, E., Stehfest, E., Thornton, P., Tubiello, F. N., van Vuuren, D. P., and Zhang, X.: Harmonization of global land use change and management for the period 850-2100 (LUH2) for CMIP6, Geosci. Model Dev., 13, 5425-5464, https://doi.org/10.5194/gmd-13-5425-2020, 2020.

Thompson, J. R., Carpenter, D. N., Cogbill, C. V., and Foster, D. R.: Four Centuries of Change in Northeastern United States Forests, Plos One, 8, e72540, https://doi.org/10.1371/journal.pone.0072540, 2013.

**Comment 2**: There are different versions of HYDE3.2 – it would be good to know which one was used in this manuscript.

**Response**: Thank you for this suggestion. The HYDE3.2 baseline version was used, and we add the information in the Table 1.

**Comment 3**: Does the pasture category in the dataset include natural grasslands, as well as managed grasslands and rangelands?

**Response**: Thank you for this suggestion. In this study, pasture is defined as a land cover/use category of land managed primarily for the production of introduced forage plants for livestock grazing, consistent with the National Resource Inventory. So, it doesn't include natural grasslands and rangelands.

**Comment 4**: I also had a bit of confusion about the forest category in the dataset – is it primarily about land that is being used as a forest (regardless of the numbers or ages of trees)? Or is it based more on forest land cover? This distinction between land use and land cover could be discussed a bit more to help with this. There are several places in the manuscript where the authors state that forest area decreased due to wood harvest or fuelwood extraction, but if that did not result in a conversion to another land-use type, then the forest area would not be changed (even if the land cover changed).

**Response**: Thank you for this suggestion. In this study, we use the forest definition from FIA. Forest is the land at least 10 percent stocked by forest trees of any size, or formerly having such tree cover, with a minimum area classification of 1 acre (https://www.fia.fs.fed.us/tools-data/maps/2007/descr/yfor_land.php). The newly developed forest dataset doesn't include the tree numbers or age information, and it is land used as forest.

Moreover, we agree your opinion about the wood harvest or other management activities may not change the land use type. If the forest land is cleared, the forest land will be converted to another LULC type. We revise the related description in the manuscript.

**Comment 5:** I found the color scale on figures 5 and 7 quite difficult to read to distinguish between the various land-use colors.

**Response**: Thank you for this suggestion. For figure 5, the color of pasture and grassland makes people difficult to read. We change the colors and update the figure to make it easy to read. Please see Figure R1 (Figure 8 in the revised manuscript).

[Figure]

**Figure R1:** Spatial and temporal patterns of land use and land cover in the conterminous United States during 1630-2020.

In Figure 7, there are 12 types of LULC conversion to show and it is hard to assign a suitable color, so we add a table (Table 3) to show the LULC conversion area in four periods (1630-1850, 1850-1920, 1920-2020, and 1630-2020).

**Comment 6:** Overall, I think a discussion of the differences between land-use and land-cover and how that is represented in this dataset would be a helpful addition. Also, some more discussion of how this product differs from the technical details of other products and in what ways that is useful and in what ways other products might have some advantages, along with how those differences in underlying details are driving differences in the qualitative dataset results.

**Response**: Thank you for this suggestion. In this study, our LULC data emphasizes the accuracy of area change resulting from LULC conversion rather than the changes in LULC structure or attributes. For example, forest management (e.g., wood harvest and thinning) results in the forest cover decreases and ecosystem function change, but the LULC type and area is unchanged.

For the technical details, it should include the LULC area reconstruction strategy and spatial allocation strategy. We discussed the LULC probability calculation and spatial allocation algorithm difference between this study and other land use and land cover simulation model. In fact, the key to the spatial allocation algorithm is what ways you choose to allocate the LULC area. Some LULC simulation models allocate the LULC demand (net change) at a LULC base map and generate a new LULC map in the prediction year. But this algorithm will underestimate the gross LULC change area whatever it is used to generate fractional or Boolean type data. It is also not suitable for long-term LULC simulation, because they assumed the LULC probability or suitability surface is stable. In this study, because the contemporary LULC probability pattern is not representative for the early period, we need to modify the probability to make it close to the historical LULC pattern. Therefore, we used the spatial allocation algorithm in this study and generate a map for each year. But this strategy ignores the linkages of landscape in the neighboring two years. Please see Lines 546-567.

---

## Author Response (AR2)

**Response to Reviewer #4**

Thank you for the comments and suggestions. These comments were very helpful for revising and improving our paper. We have responded to the comments point by point.

**General comments**:

As the three reviewers indicated, the manuscript reconstructed the history of land use and land cover over the conterminous United States (CONUS) at annual time scale and 1 km x 1 km spatial resolution in the past 390 years (1630-2020). Obviously, this is an important task. The high spatial and temporal LULC dataset is crucial for understanding and predicting the dynamics of coupled natura-human system across CONUS. I have reviewed the revised manuscript as well as the point by point responses to the comments by three reviewers. Overall, the authors have done a good job in addressing these comments. They have revised the manuscript by following the suggestions and comments closely. Having said that, I also want to point out the authors could address the comment about uncertainty associated with the new datasets by reviewer #1 better. I agree with the reviewer about a major need for the manuscript is recognition of the differences between source datasets, and the uncertainties it introduces into the modeling. I think the authors could address this comment better in revising the manuscript.

I have another comment regarding the presentation of the result. Overall, the flow and organization of this manuscript is pretty good. However, right now the first heading under Results is 3.1 Comparison with other datasets. It seems to me this is not the best. I'd like the authors to present the high-resolution and long-term dataset first. Tell us some unique features of this dataset. After that, then compare this dataset with other datasets.

**Response:** Thank you for the comments and suggestions. According to your suggestions, we add the uncertainty analysis by comparing the multiple datasets in their overlap period and reorganized the ***Results*** section.

**Comment 1:** Having said that, I also want to point out the authors could address the comment about uncertainty associated with the new datasets by reviewer #1 better. I agree with the reviewer about a major need for the manuscript is recognition of the

differences between source datasets, and the uncertainties it introduces into the modeling. I think the authors could address this comment better in revising the manuscript.

**Response:** Thanks for this suggestion. In this study, four major land use and cover types (urban, cropland, pasture, and forest) were reconstructed by integrating multisource datasets. Due to the differences among the datasets, some uncertainties would be introduced to the model in the data integration process. Therefore, we added the related data comparisons in the supplementary materials to evaluate and quantify the uncertainties. Please see the following description or supplementary materials. We also add the main results of the data comparisons in the *Discussion* section. Please see Lines 541-555.

**Urban land**

In the *Result* and *Discussion* section, we compared the Historical Settlement Data Compilation for the United States (HISDAC-US) built-up area and the newly developed urban land. The results showed that the urban land area derived from the HISDAC data was higher than that from our data. It is because the HISDAC data is rebuilt using the detailed property records and have a relatively coarse resolution. For example, the national total urban land area from this study is about 73% of that from HISDAC data between 2001 and 2015.

Considering the differences in the urban land area, we applied the annual change rate rather than the absolute value of HISDAC data as the input to reconstruct the historical urban land area for 1810-2001. We assumed that the HISDAC data could accurately capture the urban land expansion trends. To quantify the uncertainties, we calculated the relative difference in area change rate ($HISDAC_{t1}/HISDAC_{t2}$) between our reconstruction and HISDAC data in the overlap period (2001-2015), which can be expressed as follows:

$$RD_{urban,t} = \left| \frac{HistUrban\_CR_t - HISDAC\_CR_t}{HistUrban\_CR_t} \right| \times 100\% \qquad (1)$$

Where $RD_{urban,t}$ refers the relative difference in area change rate between the HISDAC data and the newly developed urban land data; $HistUrban\_CR_t$ and $HISDAC\_CR_t$ are the area change rate derived from the newly developed urban land and HISDAC data.

The mean relative difference in area change rate between the two datasets at the national level is 3.83% during 2001-2015 (Figure R1a). Because there are only four overlap time points (2001, 2005, 2010, and 2015), we further calculated the state-level relative difference (Figure R1b). The results show that the mean relative difference of the 48 states in 2001-2005, 2005-2010, 2010-2015, and 2001-2015 are 3.34±1.90%, 1.75±1.65%, 0.71±0.97%, and 1.93±1.89%, respectively (Figure R1b). Thus, the uncertainty induced by data difference should be little.

[Figure]

**Figure R1.** (a) National level relative difference in area change rate between the newly developed urban land data and HISDAC data during 2001-2005, 2005-2010, and 2010-2015. (b) State-level mean relative difference in area change rate between the newly developed urban land data and HISDAC data during 2001-2005, 2005-2010, 2010-2015, and 2001-2015.

**Cropland**

Four datasets, including the USDA Economic Research Service (ERS) cropland harvested area, USDA Census of Agriculture Historical Archive (CAHA), HYDE3.2 cropland, and total population, were used to reconstruct the historical cropland area. For 1910-2020, we used the ERS cropland harvested area (national level) to subtract the double-cropped area and optimize the interannual variations of CAHA cropland harvested area data. To quantify the uncertainties, we calculated the relative difference between the newly developed cropland area and the CAHA cropland harvested area, which can be expressed as:

$$RD_{crop,t} = \left|\frac{HistCrop_t - CHA_t}{HistCrop_t}\right| \times 100\% \tag{2}$$

Where $RD_{crop,t}$ refers the relative difference in cropland area between the newly developed cropland data and CAHA cropland harvested area data in year $t$;

$HistCrop_t$ is the cropland area derived from the newly developed cropland data in year $t$; $CHA_t$ is the CAHA cropland harvested area in year $t$.

We found that the mean relative difference in cropland harvested area between the two datasets is 2.23±1.18%, but the data between the 1960s and the 1980s had relatively large differences ranged from 1.81% to 6.02% (Figure R2a). Therefore, the uncertainty induced by cropland area adjustment is little.

During 1879-1909, we adjusted the CAHA cropland harvested area based on the reconstructed cropland area between 1910-2020. The mean relative difference (1.02± 0.19%) between the CAHA cropland harvested area and the newly developed cropland (Figure R2b), indicating that little uncertainty was introduced to the model.

[Figure]

**Figure R2.** Relative differences in cropland area between the newly developed cropland data and the CAHA cropland harvested area data during 1919-2017 (a) and 1879-1909 (b).

For the period before 1879, we integrated the newly developed cropland data (1879-2020) and HYDE3.2 cropland data (1630-1879) to reconstruct the historical cropland area. However, the cropland definitions of this study and HYDE3.2 cropland data are different (Table S5), resulting in the uncertainties to the reconstruction.

Considering the cropland area and definition differences, we used the HYDE3.2 cropland per capita change rate rather than the absolute value of cropland area between 1630 and 1879. To quantify the uncertainties, we calculated the relative difference in cropland per capita change rate ($HistCrop_{p,t1}/HistCrop_{p,t2}$) between the newly developed cropland data and HYDE3.2 cropland data in the overlap period (1880-2017), which can be expressed as follows:

$$RD_{crop\_p,t} = \left| \frac{HistCrop\_CR_{p,t} - HYDE\_crop\_CR_{p,t}}{HistCrop\_CR_{p,t}} \right| \times 100\% \qquad (3)$$

Where $RD_{crop\_p,t}$ refers the relative difference of cropland per capita change rate between the newly developed cropland data and HYDE3.2 cropland data; $HistCrop\_CR_{p,t}$ and $HYDE\_crop\_CR_{p,t}$ are the cropland per capita change rate derived from the newly developed cropland data and HYDE3.2 cropland data in year $t$, respectively.

Compared with HYDE3.2 cropland data, the newly developed cropland data showed significant interannual variations during 1880-2017 (Figure R3a). The mean relative difference in cropland per capita change rate between HYDE3.2 cropland data and the newly developed cropland data during 1880-2017 is $2.10 \pm 2.82\%$, and the relative difference values in most of the years were lower than 5% (Figure R3b).

[Figure]

**Figure R3.** Cropland per capita relative difference between the newly developed data and HYDE3.2 cropland data during 1880-2017.

We further calculated the relative difference of state-level cropland per capita change between HYDE3.2 cropland data and the newly developed cropland data during 2001-2017 (Figure R4). The results showed that the two datasets matched well with the mean relative difference value of $1.21 \pm 1.45\%$ (2001-2017). And the relative difference values in most states are lower than 3%, except Colorado (4.17%), New Mexico (4.83%), and Rhode Island (3.35%) (Figure R4).

[Figure]

**Figure R4.** Mean relative difference of state-level cropland per capita change derived from HYDE3.2 cropland and the newly developed cropland land data during 2001-2017.

**Pasture**

In this study, three datasets (National Resources Inventory (NRI) pasture data, HYDE3.2 pasture, and total population data) were used for the historical pasture reconstruction. For the year before 1982, we integrated the newly developed pasture data (1982-2017) and HYDE3.2 pasture data (1630-1982) to reconstruct the historical pasture area. However, the pasture definitions of this study and HYDE3.2 pasture data are different (Table S6), resulting in uncertainties to the reconstruction.

Considering the pasture area and definition differences, we used the change rate of HYDE3.2 pasture per capita ($HYDE\_Pasture\_p_{t1}/HYDE\_Pasture\_p_{t2}$) rather than the absolute value of pasture area between 1630 and 1982. To quantify the uncertainties, we calculated the relative difference in pasture per capita change rate between our data and HYDE3.2 pasture data, which can be expressed as follows:

$$RD_{pasture\_p,t} = \left| \frac{HistPasture\_CR_{p,t} - HYDE\_Pasture\_CR_{p,t}}{HistPasture\_CR_{p,t}} \right| \times 100\% \tag{4}$$

Where $RD_{pasture\_p,t}$ refers the relative difference in pasture per capita change rate between the newly developed pasture data and HYDE3.2 pasture data; $HistPasture\_CR_{p,t}$ and $HYDE\_Pasture\_CR_{p,t}$ are the pasture per capita change rate of the newly developed pasture and HYDE3.2 pasture in year *t*, respectively.

We calculated the pasture per capita change rate from HYDE3.2 pasture data and the newly developed pasture data in the overlap period (1982-2017) (Figure R5). The results showed that mean change rates in pasture per capita were 1.01±0.03 (HYDE3.2) and 1.02±0.02 (This study), respectively. The mean relative difference in pasture per capita change rate between HYDE3.2 pasture data and the newly developed pasture data is 4.89±1.94%.

[Figure]

**Figure R5:** Pasture per capita change rate of HYDE3.2 pasture data and the newly developed pasture data during 1982-2017.

We also calculated the relative difference of state-level pasture per capita change between HYDE3.2 pasture data and the newly developed pasture data during 1982-2017 (Figure R6). The results showed that the mean relative difference of 48 states was 6.51±6.62%, and the relative difference values of all the states were lower than 8% in the seven sub-period.

[Figure]

**Figure R6:** Mean relative difference of state-level pasture per capita change rate between HYDE3.2 pasture data and the newly developed pasture data during 1982-2017.

For the forest, we integrated two datasets (USDA and FATD) to generate the historical forest land during 1630-2020. In the overlap period, the forest area from the two datasets is the same. We didn't calculate uncertainties for the forest area.

**Comment 2:** I have another comment regarding the presentation of the result. Overall, the flow and organization of this manuscript is pretty good. However, right now the first heading under Results is 3.1 Comparison with other datasets. It seems to me this is not the best. I'd like the authors to present the high-resolution and long-term dataset first. Tell us some unique features of this dataset. After that, then compare this dataset with other datasets.

**Response**: Thank you for this suggestion. We reorganize of the ***Results*** section. We move the section 3.1 to the last part of ***Results*** (3.4). The previous section 3.2., 3.3, and 3.4 were revised as section 3.1, 3.2, and 3.3, respectively.

**Specific comments:**

**Comment 1:** The title of this manuscript can be further improved. I'd hope the title includes the word of "reconstruction" or "reconstruct" to reflect the task of this manuscript.

**Response**: Thank you for this suggestion. We revised the title for this manuscript. The new title for this manuscript is as follows:

***Four-century history of land transformation by humans in the United States (1630-2020): Reconstructing annual and 1-km grid data for the HIStory of LAND changes (HISLAND-US)***

**Comment 2:** Line 249-251. The last sentence of this paragraph is a little bit wordy and need to be cleared up.

**Response**: Thank you for this suggestion. We revised the sentence and make it clear. Please see Lines 250-251.

**Comment 3:** Line 269. The word "access" need to be replaced by the word "assess" instead. Similar errors need to be corrected elsewhere.

**Response**: Thank you for this suggestion. We check the similar errors and revised.